# New Guarantees for Learning Revenue Maximizing Menus of Lotteries and Two-Part Tariffs

**Maria-Florina Balcan**                                          *ninamf@cs.cmu.edu*
*Carnegie Mellon University*

**Hedyeh Beyhaghi**                                              *hedyeh@cmu.edu*
*Carnegie Mellon University*

**Reviewed on OpenReview:** *https://openreview.net/forum?id=mhawjZcmrJ*

## Abstract

We advance a recently flourishing line of work at the intersection of learning theory and computational economics by studying the learnability of two classes of mechanisms prominent in economics, namely *menus of lotteries* and *two-part tariffs*. The former is a family of randomized mechanisms designed for selling multiple items, known to achieve revenue beyond deterministic mechanisms, while the latter is designed for selling multiple units (copies) of a single item with applications in real-world scenarios such as car or bike-sharing services. We focus on learning high-revenue mechanisms of this form from buyer valuation data in both distributional settings, where we have access to buyers' valuation samples up-front, and the more challenging and less-studied online settings, where buyers arrive one-at-a-time and no distributional assumption is made about their values. We provide a suite of results with regard to these two families of mechanisms. We provide the first online learning algorithms for menus of lotteries and two-part tariffs with strong regret-bound guarantees. Since the space of parameters is infinite and the revenue functions have discontinuities, the known techniques do not readily apply. However, we are able to provide a reduction to online learning over a finite number of *experts*, in our case, a finite number of parameters. Furthermore, in the limited buyers type case, we show a reduction to online linear optimization, which allows us to obtain no-regret guarantees by presenting buyers with menus that correspond to a barycentric spanner. In addition, we provide algorithms with improved running times over prior work for the distributional settings. Finally, we demonstrate how techniques from the recent literature in data-driven algorithm design are insufficient for our studied problems.

## 1 Introduction

**Overview.** In recent years, a growing body of work has emerged in the field of machine learning for pricing and mechanism design problems. These problems involve selling items to buyers with the objective of maximizing revenue. The majority of the existing research has primarily concentrated on *distributional settings*, i.e., when the buyers' values for the items are drawn from an unknown distribution. Less attention has been paid to the more challenging case of *online setting*, where buyers arrive one by one and no distributional assumption about buyers' values is considered. In this case, the previous literature has mostly focused on simple mechanisms such as posted pricing or, more generally, mechanisms that sell the items separately (Blum et al., 2004; Kleinberg and Leighton, 2003; Blum and Hartline, 2005; Balcan and Blum, 2006; Bubeck et al., 2017; Cesa-Bianchi et al., 2014; Balcan et al., 2018b; 2020a). We advance this line of work by studying the learnability of two prominent classes of mechanisms, both represented as menus providing the buyers a list of allocation and payment options to choose from, namely menus of two-part tariffs and lotteries. These mechanisms go beyond selling the items separately, resulting in potentially higher revenue guarantees with applications to modern real-world scenarios. We provide a collection of results for these mechanisms while

discovering technical surprises compared to prior work in data-driven algorithm and mechanism design. Our results include the first online learning guarantees for menus of two-part tariffs and lotteries and improved guarantees for distributional learning. In the process, we establish a data-independent discretization method, despite the drastic failure of this technique in problems with a similar utility function (Balcan et al., 2017; 2018a; 2023a;b). In addition, we demonstrate inadequacy of recently developed techniques in data-driven algorithm design for our settings. In particular, for the first time, we provide evidence for the failure of the *dispersion* property (Balcan et al., 2018b; 2020a)—a sufficient condition to provide a no-regret algorithm under the smooth distributional assumption, which is widely applied to parametric algorithm and mechanism design problems—for a specific problem (menus of lotteries).

**Problem Setup.** The first class we study is *menus of two-part tariffs* (Lewis, 1941), used for selling multiple units (i.e., copies) of a single item. In this family of mechanisms, the buyer is presented with a list (menu) of *two-part tariffs*, where *tariff* $i$ is a pair consisting of an up-front fee, $p_1^{(i)}$, and a per-unit fee, $p_2^{(i)}$. If the buyer wishes to buy $k \geq 1$ units of tariff $i$, she pays in total $p_1^{(i)} + kp_2^{(i)}$, and if she does not want to buy anything, she does not pay anything. The buyer has the freedom to select any of the tariffs. In particular, the cost for purchasing $k \geq 1$ units is the minimum cost among all the tariffs, i.e., $\min_i(p_1^{(i)} + kp_2^{(i)})$. Various products in the real world are sold via menus of two-part tariffs; for example, car or bike-sharing services and delivery service subscriptions.

The second class we study is the *menus of lotteries* for selling multiple items. In this context, the buyer is presented with a list (menu) of *lotteries*, where *lottery* $i$ is defined as a pair consisting of a vector of probabilities for allocating each item, $\phi^{(i)}$, and a price, $p^{(i)}$. If the buyer wishes to choose lottery $i$, she receives each item $j$ with probability $\phi^{(i)}[j]$ and pays $p^{(i)}$. Menus of lotteries are a crucial family of mechanisms because (1) this family captures all possible mechanisms, including the optimal one (Dasgupta et al., 1979; Guesnerie and Oddou, 1981), and (2) menus of lotteries achieve revenue beyond other well-studied families of mechanisms such as posted pricing and, more generally, any deterministic mechanism (Briest et al., 2010; Hart and Nisan, 2019).

We study menus of two-part tariffs and lotteries in the context of parameter optimization, where the objective function (revenue) depends on parameter vectors. In menus of two-part tariffs, the parameters determining the mechanisms are the up-front fees and per-unit fees for each tariff, while for menus of lotteries, the allocation probability vectors and the prices for the lotteries determine the mechanism. In the parameter space, each point corresponds to a mechanism. A common approach in learning algorithms involves considering the objective function for a fixed buyer's valuation (Balcan et al., 2017; 2018c;b). In our context, the mechanism designer faces a utility-maximizing buyer, who, given the parameters determining the menu, chooses the entry, i.e., a lottery or a two-part tariff, in the menu that maximizes her utility. Therefore, the revenue function at any parameter vector is equal to the payment corresponding to the entry selected by the buyer.

## 1.1 Our Contributions

We study the learnability of menus of two-part tariffs and lotteries in both online and distributional settings. We advance the state-of-the-art in several aspects.

**Technical challenges, Structural Properties, and a Revenue Preserving Cover.** "Discretization" is a natural technique in data-driven algorithm design. In this approach, a finite set of parameter vectors, each representing a menu in the parameter space, are selected, and the algorithms optimize over that set. The smaller the set, the better the generalization guarantees will be in the distributional setting, and the better the regret guarantees will be in the online setting, with respect to the best menu in the set. In our setting, a proper data-independent discretization scheme would guarantee that independent of the buyer's valuation, this set always contains a nearly optimal menu. More specifically, for any arbitrary parameter vector representing a menu, a menu in the set should generate almost as much revenue, independent of the buyer's valuation. However, due to sharp discontinuities of revenue in the parameter space, devising such a discretization can be challenging. For instance, consider a menu with two high-utility entries for a buyer such that these entries have similar utility for the buyer but very different prices (e.g., one with high allocation and high price, the other with low allocation and low price). Minor changes in the parameters of

these entries; e.g., rounding the parameters down to multiples of $\epsilon$, may alter their utility order, causing the buyer to switch between them, resulting in an arbitrary loss in revenue.

By extracting structural properties for menus of **two-part tariffs**, we develop a novel discretization method that identifies a finite set of menus that approximate the revenue of any arbitrary limited-length menu (Theorem 1). At a high level, in finding a corresponding menu with approximate revenue guarantee, the options (tariff and quantity pairs) with higher prices need to experience a more significant decrease in price (compared to the lower price ones) so that no buyer switches from a high-price to a low-price option. In menus of **lotteries**, we extend the discretization of menus of lotteries developed by Dughmi et al. (2014) (Theorem 27). Our extension is three-fold: we remove the lower bound assumption on value distribution, support additive valuations, and provide improved regret bounds and running times when the size of the menu is limited. In both settings (two-part tariffs lotteries), our discretization is data-independent; e.g., the set of discretized menus consists of all menus with parameters that are multiples of $\varepsilon$ or powers of $(1-\varepsilon)$. The novelty of the result, however, lies in the analysis, which illustrates despite the challenges discussed above, for each arbitrary menu and valuation, this set contains a corresponding approximately revenue-preserving menu.

**Online Learning (adversarial inputs and smooth distributional assumptions).** For menus of **two-part tariffs**, we provide the first no-regret online learning algorithms under adversarial (worst-case) inputs and also smooth distributional assumptions. For the full information setting, both settings lead to similar regret terms; however, the comparison of their running time depends on the support of the distribution and the maximum number of units available (Theorems 7 and 10). In the bandit setting, again, the regrets of both settings are similar. However, the comparison between the efficiencies of the algorithms depends on the smoothness factor of the distributions (Theorems 8 and 11). Furthermore, we provide the first no-regret algorithm for a *semi-bandit* setting (Theorem 12) with a polynomial running time in the number of discontinuities in the parameter space. This setting lies between the full-information and bandit settings, and the learner observes the revenue function for a set of menus containing the menu used. For menus of **lotteries**, we provide the first no-regret online learning algorithms under adversarial inputs (Theorems 28 to 30). In addition, we provide evidence that menus of lotteries may not satisfy *dispersion* (Balcan et al., 2018b; 2020a)—a sufficient condition to provide a no-regret algorithm under smooth distributional assumption—without assuming extra structures about the optimal solution (Theorem 33). Menus of lotteries are the first family of mechanisms for which there is evidence of a potential failure of the dispersion property.

**Distributional Learning.** We also provide novel distributional learning algorithms for menus of two-part tariffs and lotteries. Our algorithms choose several menus in a data-independent way (via data-independent discretization) and then select the best of them based on the data. In the context of **two-part tariffs**, our algorithm is much simpler than prior ones for the same problem, yet it enjoys improved worst-case runtime guarantees compared to them (Balcan et al., 2018c; 2020b) when the length of the menu is more than one (Theorem 26). We note that for other data-driven algorithm design problems, such as data-driven clustering and data-driven learning to branch, it was proven that algorithms that use data-independent discretization could perform very poorly (Balcan et al., 2017; 2018a; 2023a). Thus, by contrast, our work shows the power of data-independent discretization for data-driven mechanism design and algorithm design more generally. In the context of **lotteries**, compared to the previous distributional learning results for fixed-length menus (Balcan et al., 2018c), our algorithm requires similar sample complexity; however, it has an efficient implementation (Theorems 34 and 57).

**Limited Buyer Types.** For limited buyer types, we provide improved regret bounds for both the full-information and partial-information (bandit) settings for both menus of two-part tariffs and lotteries (Theorems 24, 25, 31 and 32). The high-level idea is as follows. Consider the revenue function in the parameter space for a fixed buyer. The parameter space is partitioned into regions where, within each region, the buyer selects the same option in the menu, e.g., the same lottery, resulting in a continuous revenue function. Discontinuity occurs across regions. For limited-type buyers, by superimposing the revenue functions for all types, the parameter space divides into more (albeit still a limited number of) regions. Regardless of the buyer type at hand, the revenue function is continuous within each region and in our case, linear. Therefore, it is sufficient to only consider the corner points as potential parameter vectors that maximize the revenue.

We show that in the full information case, running the weighted majority algorithm on the set of menus corresponding to the regions' corner points results in sublinear regret.

In the partial information setting, we show a reduction to online linear optimization, allowing us to obtain no-regret guarantees by presenting buyers with menus corresponding to a barycentric spanner. Our reduction is inspired by Balcan et al. (2015); however, we apply the reduction in the different contexts of pricing schemes. In the partial information setting, in each round, we only observe the revenue of the current menu. To estimate the revenue from all the menus efficiently, or in other words, to find an *unbiased estimator* with a *bounded range*, we employ the notion of *barycentric spanners* in online learning introduced by Awerbuch and Kleinberg (2008). By utilizing this concept, we provide algorithms with a regret bound that is sublinear in the number of timesteps and polynomial in other parameters. This is the first time that the barycentric spanner notion has been applied to an auction design setting.

## 1.2 Summary of Contributions

First, we overview the results related to **menus of two-part tariffs**.

- By extracting structural properties, we develop a novel discretization method that identifies a finite set of menus that approximate the revenue of any arbitrary menu, including the optimum for any valuation. This allows the development of new no-regret online learning algorithms as well as improved distributional learning algorithms (see the two bullet points below).

- We provide the first no-regret online learning algorithms under adversarial inputs, smooth distributional assumptions, and limited buyer-type assumptions (under full information, bandit setting, and semi-bandit setting).

- We also provide a novel distributional learning algorithm for menus of two-part tariffs. Our algorithm chooses several menus of two-part tariffs in a data-independent way (via data-independent discretization) and then selects the best of them based on data. This is much simpler than previous algorithms (Balcan et al., 2018c; 2020b) for the same problem, yet it enjoys improved runtime guarantees in the worst-case scenario when the length of the menu is more than one.

- For limited buyer types, we provide improved regret bounds for both the full-information and bandit settings. We show a reduction to online linear optimization, which allows us to obtain no-regret guarantees by presenting buyers with menus that correspond to a barycentric spanner.

Next, we overview our results related to **menus of lotteries**.

- We extend the discretization of menus of lotteries developed by Dughmi et al. (2014). Our extension is three-fold: we remove the lower bound assumption on value distribution, support additive valuations, and provide improved regret bounds and running times when the size of the menu is limited.

- We provide the first no-regret online learning algorithms under adversarial inputs.

- Compared to the previous distributional learning results for fixed-length menus Balcan et al. (2018c), our algorithm requires similar sample complexity; however, it has an efficient implementation.

- We provide evidence that menus of lotteries may not satisfy dispersion—a sufficient condition to provide a no-regret algorithm under smooth distributional assumption—without assuming extra structures about the optimal solution. Menus of lotteries are the first family of mechanisms where there is evidence for potential failure of the dispersion property.

- For limited buyer types, we provide improved regret bounds for both the full-information and bandit settings. We show a reduction to online linear optimization, which allows us to obtain no-regret guarantees by presenting buyers with menus that correspond to a barycentric spanner.

### 1.3 Related Work

Studying learnability of classes of mechanisms for the revenue maximization objective has been of great interest in recent years (Alon et al., 2017a; Cole and Roughgarden, 2014; Devanur et al., 2016; Elkind, 2007; Gonczarowski and Nisan, 2017; Guo et al., 2019; Roughgarden and Schrijvers, 2016). These mechanisms have been studied mostly in a distributional setting, where buyers' values are drawn from an unknown distribution, and the online setting, where there is no distributional assumption on the buyers' values, has been explored less.[1] In the distributional setting, various mechanism classes, including posted-price mechanisms, second-price auctions with reserves, menus of two-part tariffs, and menus of lotteries, are known to be learnable (Morgenstern and Roughgarden, 2015; 2016; Balcan et al., 2016; 2018c; 2021a; Dughmi et al., 2014; Gonczarowski and Weinberg, 2021; Mohri and Medina, 2016; Syrgkanis, 2017; Dütting et al., 2019). In the online setting, under adversarial input (Blum et al., 2004; Kleinberg and Leighton, 2003; Blum and Hartline, 2005; Balcan and Blum, 2006; Roughgarden and Wang, 2016; Bubeck et al., 2017), and also under stochastic input (Cesa-Bianchi et al., 2014; Balcan et al., 2018b; 2020a) mostly simple mechanisms such as posted pricing and second-price auction are considered where both mechanisms sell the items separately. An exception is Roughgarden and Wang (2016) who study Vickrey-Clarke-Groves (VCG) mechanism with multiple reserves; however, the algorithms provided are not no-regret in the classic sense but are bounded-regret compared to a constant approximation of the optimal solution.

Two of the prominent approaches used for developing distributional results are pseudo-dimension-based and discretization-based. In the first approach, despite the discontinuity present in the utility of buyers as a function of the parameters used in the mechanism, it is shown that the pseudo-dimension of the family is bounded by using smoothness assumptions on the distribution. This approach applies to all the mechanisms mentioned above. In the discretization approach, a finite set of parameters is identified such that limiting the search space to this set is approximately optimal. This approach has been used for a limited number of mechanisms, such as item-pricing for combinatorial auctions for unrestricted supply (Balcan et al., 2008) and menus of lotteries in a limited setting (Dughmi et al., 2014). In the online setting, Balcan et al. (2018b) and Balcan et al. (2020a) introduce *dispersion* as a sufficient condition for online learnability of families of mechanisms. They show several classes of mechanisms, such as posted-price mechanisms and second-price auctions with reserves, satisfy dispersion and, therefore, establish strong regret bounds for online learning. Discretization-based techniques in online learning scenarios have been used for the simple cases of item-pricing (Blum et al., 2004) and the second-price auctions (Cesa-Bianchi et al., 2014).

**Two-Part Tariffs.** Two-part tariff pricing schemes were first introduced by Lewis (1941) and later analyzed by Oi (1971). Menus of two-part tariffs have been studied recently in the context of distributional learning (Balcan et al., 2018c; 2020b; 2022b). A recent work (Balcan et al., 2022b) provides improved running time bounds over Balcan et al. (2020b) for distributional learning of two-part tariffs in the case where the number of pieces with continuous sum of utility functions $u(x_i, \cdot)$ across all problem instances is small (as defined in Section 3.2.2 the utility function $u(x_i, .)$ measures the performance of our two-part tariff mechanisms on a fixed problem instance $x_i$ as a function of its parameters). However, for the case where the menu length is strictly greater than 1, Balcan et al. (2022b)'s approach does not lead to improved running time over Balcan et al. (2020b) for worst-case instances. So, for worst-case instances and menu length $> 1$, our approach for distributional learning improves over previously best-known results.

**Menus of Lotteries.** Menus of lotteries capture all possible mechanisms, including the optimal one, for selling items to buyers. The Taxation Principle (Dasgupta et al., 1979; Guesnerie and Oddou, 1981) asserts that any mechanism for a single buyer can be represented as a menu of lotteries, where the buyer selects their favorite lottery (that is, the one that maximizes the buyer's expected value for the randomized allocation minus the price paid). Furthermore, menus of lotteries achieve revenue beyond other well-studied families of mechanisms such as posted pricing and, more generally, any deterministic mechanism. For a correlated buyer (a buyer whose values for items are correlated), even in the simple cases where the buyer is additive (their value for a bundle of items is the sum of the value for individualized items) or unit-demand (their value for a bundle of items is the maximum value for an item in the bundle), the gap between optimal

---

[1]Some online learning algorithms, including those proved via the dispersion method, explained later, still make distributional assumptions; however, unlike the distributional learning setting, the draws are not necessarily from identical distributions.

randomized mechanism (lotteries) and item-pricing is infinite (Briest et al., 2010; Hart and Nisan, 2019). Daskalakis et al. (2014) show that even for an independent additive buyer (the values for the items are independent), lotteries (randomized mechanisms) are necessary and provide strictly more revenue compared to any deterministic mechanism, including pricing mechanisms.

**Failure of data-independent discretization-based learning.** Discretization is a natural approach for designing algorithms to tune parameters (e.g., prices for menus of two-part tariffs and allocation probabilities and prices for menus of lotteries) and is commonly used in applied fields such as applied machine learning. However, recent work has shown that in tuning parameters of algorithms for solving discrete combinatorial problems, discretization in the context of data-driven algorithm design does not always work if discretization is done in a data-independent way. For the case of tuning parameters for linkage-based algorithms, Balcan et al. (2017) showed that for several natural parameterized families of clustering procedures, for any data-independent discretization, there exists an infinite family of clustering instances such that any of the discrete parameters will output a clustering that is an $\Omega(n)$ factor worse than the optimal parameter, where $n$ is the input size. Here, the quality of clustering can be defined according to several well-known objectives, including $k$-median, $k$-means, and $k$-center. Balcan et al. (2018a; 2023a) show that for the data-driven problem of learning to branch for solving mixed integer linear programs (MILPs), data-independent discretization will not work either. More specifically, for any discretization of the parameter space $[0, 1]$, there exists an infinite family of distributions over MILP problem instances such that for any parameter in the discretization, the expected tree size is exponential in the input parameter. Yet, there exists an infinite number of parameters such that the tree size is just a constant (with probability 1). Remarkably, we show that in our context, even data-independent discretization works.

**Dispersion and Online Data-Driven Algorithm Design.** Dispersion is a recently-developed notion for families of algorithmic and mechanism design problems and serves as a sufficient condition for the existence of bounded-regret online learning algorithms (Balcan et al., 2018b; 2020a; Balcan, 2020) and differentially private distributional learning algorithms (Balcan et al., 2018b). Generally speaking, this condition bounds the concentration of discontinuities of the objective function in any small regions in the parameter space. Dispersion-based techniques have been established successfully for a variety of algorithms (Balcan and Sharma, 2021; Balcan et al., 2021b; 2022a), among which is tuning parameters in combinatorial problems, such as clustering problems discussed above (Balcan et al., 2018b). For menus of two-part tariffs, we show that the dispersion condition is satisfied, immediately implying no-regret online learning algorithms and differentially private algorithms for distributional learning. Surprisingly, we present evidence that dispersion might not apply to menus of lotteries. In particular, we show in menus of lotteries the objective function might have sharp discontinuities concentrated in a small region. This structural property is in stark contrast with menus of two-part tariffs and other mechanism and algorithm families satisfying dispersion. Despite this evidence, we show that a simple discretization-based approach leads to no-regret online learning algorithms for menus of lotteries.

**Sample Complexity for Menus of Lotteries.** The sample complexity for menus of lotteries has been studied under two different assumptions: independence of valuation across items, as studied by Gonczarowski and Weinberg (2021) and correlated valuation across items, as studied by Dughmi et al. (2014); Brustle et al. (2020). By assuming independence simultaneously among the buyers and the items, a significant improvement over the sample complexity is possible (Gonczarowski and Weinberg, 2021). However, when the values for the items are possibly correlated, Dughmi et al. show a lower bound on the sample complexity, verifying an exponential gap on the dependence in the number of items compared to Gonczarowski and Weinberg. Brustle et al. (2020) study a setting between the two extremes of arbitrary correlation and independence where they assume structured dependence across items, generalizing the results of Gonczarowski and Weinberg (2021) and improving the sample complexity over Dughmi et al. (2014) for special cases of correlation. Similar to Dughmi et al. and in contrast with Gonczarowski and Weinberg, we do not assume independence (or structured dependence) across items and only assume independence among the buyers.

## 2 Model and Preliminaries

We consider selling items to a single buyer for the revenue objective through parameterized families of mechanisms. In this paper, the family of mechanisms is either the set of menus of two-part tariffs or lotteries. To put our notations in context, in this section, we focus on menus of two-part tariffs as our running example. The discussed settings also hold for menus of lotteries – we defer the discussion related to menus of lotteries to Section 4.

Menus of two-part tariffs are used for selling multiple units (i.e., copies) of a single item through a list of up-front and per-unit fee pairs that the buyer can choose from. Menu $M = \left\{ \left( p_1^{(1)}, p_2^{(1)} \right), \ldots, \left( p_1^{(\ell)}, p_2^{(\ell)} \right) \right\} \subseteq \mathbb{R}^{2\ell}$, is a length-$\ell$ menu of two-part tariffs. Each menu $M$ is parameterized by $\boldsymbol{\rho}$ which in this case is $2\ell$-dimensional and contains all $p_1^{(j)}$ and $p_2^{(j)}$ where all $p_1^{(j)}, p_2^{(j)} \in [0, H]$. $p_1^{(j)}$ and $p_2^{(j)}$ are called the up-front fee (price) and per-unit fee (price) of tariff $j$, respectively. We denote a buyer's valuations for all $1, 2, \ldots, K$ units by $\boldsymbol{v} = (v(1), \ldots, v(K))$, where the values are nonnegative, monotonically increasing, belong to $[0, H]$, and $v(0) = 0$. Under the tariff $j$ denoted by $\left( p_1^{(j)}, p_2^{(j)} \right)$ and the number of units $k \in \{1, \ldots, K\}$ that the buyer selects, she receives $k$ units of the item and pays $p_1^{(j)} + k p_2^{(j)}$. The buyer's utility is her value for the number of units bought $v(k)$ less the payment. Each buyer has the option of buying their utility-maximizing tariff and number of units. In other words, the buyer will buy $k$ units using tariff $j$ that maximizes $v(k) - p_1^{(j)} - k p_2^{(j)}$ or does not buy and does not pay anything.

Let $\mathcal{M}$ be an infinite set of mechanisms parameterized by a set $\mathcal{C} \subseteq \mathbb{R}^d$. In this paper, $\mathcal{M}$ is either the set of two-part tariff menus or lottery menus. Consider the case where $\mathcal{M}$ is the set of two-part tariff menus for selling multiple units of a single item to a buyer with value $\boldsymbol{v}$ while the menu corresponds to parameter $\boldsymbol{\rho} \in \mathcal{C}$. Next, let $\Pi$ be a set of problem instances for $\mathcal{M}$, such as a set of buyer valuations $\boldsymbol{v}$, and let $u : \Pi \times \mathcal{C} \to [0, H]$ be a utility function where $u(x, \boldsymbol{\rho})$ measures the performance of the mechanism with parameters $\boldsymbol{\rho}$ on problem instance $x \in \Pi$. In our case, $u(x, \boldsymbol{\rho})$ is the revenue of the mechanism (a menu of two-part tariffs or lotteries) with parameters $\boldsymbol{\rho}$ on input $x$. For example, for the menus of two-part tariffs, $\mathcal{M}$ is the set of possible menus $M$ and since each menu is $2\ell$-dimensional with each dimension in $[0, H]$, $\mathcal{C} = [0, H]^{2\ell} \subseteq \mathbb{R}^{2\ell}$. $\Pi$ is the set of buyer valuations $\boldsymbol{v}$ and $u : \Pi \times \mathcal{C} \to [0, H]$ be a utility function where $u(\boldsymbol{v}, \boldsymbol{\rho})$ measures the revenue of the menu with parameters $\boldsymbol{\rho}$ on buyer valuations $\boldsymbol{v} \in \Pi$.

**Online Setting.** In this setting, a sequence of functions $u_1, \ldots, u_T : \mathcal{C} \to [0, H]$ arrive one by one. Unlike $u$, $u_t$ only takes parameter $\boldsymbol{\rho_t}$ as the input and is defined as $u_t(\boldsymbol{\rho_t}) := u(\boldsymbol{\rho_t}, x_t)$, where $x_t$ is the problem instance at timestep $t$. At the time $t$, the no-regret learning algorithm chooses a parameter vector $\boldsymbol{\rho_t}$ and then either observes the function $u_t$ in the full information setting, the scalar $u_t(\boldsymbol{\rho_t})$ in the bandit setting, or $u_t(\boldsymbol{\rho_t})$ for a set of $\boldsymbol{\rho}$ in the semi-bandit setting. The goal is to minimize the expected regret, $\mathbb{E}[\max_{\boldsymbol{\rho} \in \mathcal{C}} \sum u_t(\boldsymbol{\rho}) - u_t(\boldsymbol{\rho_t})]$. We study the online setting both under adversarial input, where $u_t()$ are selected adversarially, and under smoothed distribution inputs which assume more structure. The expectation in the regret formula is taken over the randomness of the algorithm in the adversarial setting and over the randomness of the algorithm and distribution of buyers in the smoothed distributional setting.

**Distributional Setting.** In the distributional setting, the algorithm receives samples from an unknown distribution $\mathcal{D}$ over problem instances $\Pi$. The goal is to find a parameter vector $\hat{\boldsymbol{\rho}}$ that nearly maximizes the expected utility, i.e., $\max_{\boldsymbol{\rho} \in \mathcal{C}} \mathbb{E}_{x \sim D}[u(x, \boldsymbol{\rho})]$ similar to statistical learning theory (Vapnik, 1998) or PAC learning (Valiant, 1984).

## 3 Menus of Two-Part Tariffs

In this section, we consider $M = \left\{ \left( p_1^{(1)}, p_2^{(1)} \right), \ldots, \left( p_1^{(\ell)}, p_2^{(\ell)} \right) \right\} \subseteq \mathbb{R}^{2\ell}$ as a length-$\ell$ menu of two-part tariffs. See Section 2 for a detailed description.

### 3.1 Discretization Procedure

This section shows a discretization procedure for the menus of two-part tariffs. Given any menu and value $0 < \alpha < 1$, we provide an alternate menu such that all the price elements, $p_1^{(i)}$ and $p_2^{(i)}$ for all $i$, are multiples of $\alpha$ and the alternate menu provides nearly as much revenue as the given menu up to a term that depends on $\alpha$. The main result of this section is summarized in the following statement.

**Theorem 1.** *Given a menu of two-part tariffs $M$ and parameter $0 < \alpha < 1$, Algorithm 1 outputs menu $M'$ whose revenue is at least the revenue of $M$ minus $2K\alpha\ell$, for any buyer's valuation. Furthermore, for all $i$, all $p_1^{(i)}$ and $p_2^{(i)}$ are multiples of $\alpha$. The set of potential outcomes constitutes a space with at most $\min\{(H/\alpha)^{2\ell}, 2^{H^2/\alpha^2}\}$ menus, where $H$ is the maximum value for any number of units.*

Correctness of the rounding procedure (Algorithm 1) as in the proof of Theorem 1 implies that the set of menus whose prices, i.e., $p_1^{(i)}, p_2^{(i)}$, are multiples of $\alpha$ constitute (an almost) revenue-preserving set of menus.

---

**Algorithm 1:** (Almost) revenue preserving rounding for menus of two-part tariffs

---

**Input:** Menu $M$, discretization parameter $\alpha$.
1: Let $M'$ be the menu of *Pareto frontier tariffs* in $M$, derived by one by one deleting tariffs $i$ for which there exists tariff $j \neq i$ such that $p_1^{(i)} \geq p_1^{(j)}$ and $p_2^{(i)} \geq p_2^{(j)}$.
2: Reindex the tariffs in $M'$ in increasing order of $p_1$ (and hence, decreasing order of $p_2$).
3: For each tariff $i$, decrease $p_1^{(i)}$ and $p_2^{(i)}$ by $(i-1)\alpha$.       ▷ The revenue preserving step.
4: Round down all $p_1^{(i)}$ and $p_2^{(i)}$ to the closest multiple of $\alpha$.
5: Remove the duplicate tariffs.
**Output:** Menu $M'$.

---

**Proof idea of Theorem 1 and intuition behind Algorithm 1.** At a high level, for finding corresponding menus through Algorithm 1, the options (tariff quantity pairs) with higher prices need to experience a larger decrease in price so that no buyer switches from a high-price to a low-price option. The main structural ideas deriving the algorithm and the proof of the revenue guarantee are as follows: (i) for a fixed number of units $k$ to be purchased, the utility-maximizing tariff is the same across all the buyer's valuations; namely, the tariff that has the smallest overall price (upfront price plus $k$ times per-unit price), and (ii) as the number of units to be purchased increases, the per-unit price of the utility-maximizing tariff decreases. The main idea of the rounding algorithm is decreasing the corresponding prices of tariffs with lower per-unit fees by a larger amount (Line 3). By doing so, for each buyer, the total price of buying more units decreases more than the total price of buying fewer units. This step ensures that the buyer does not switch from purchasing more units to fewer units after the rounding. This property is sufficient for the revenue guarantees. The other steps of the algorithm delete redundant tariffs (Lines 1 and 5) and ensure the final prices are multiples of $\alpha$ (Line 4). The theorem provides two upper bounds for the size of the discretized space. By Line 4, all the prices are multiples of $\alpha$. Therefore, the $2\ell$ price components in a length-$\ell$ menu each have $H/\alpha$ options. This gives the first bound. On the other hand, if we consider a single tariff, each of the up-front fee and the per-unit fee has $H/\alpha$ possibilities, therefore, the total number of possible unique tariffs are $H^2/\alpha^2$. Each of these possible tariffs may or may not be on the menu, giving the second bound. The full proof is provided below.

**Remark.** Our rounding scheme (Algorithm 1) is only described for the purpose of the proof to argue that the multiples of $\alpha$ provide (an almost) revenue-preserving set of menus. Algorithmically, we only need to enumerate the multiples of $\alpha$.

#### 3.1.1 Proof of Theorem 1

Before providing the proof of the discretization procedure, we provide intuition as to why discretization is a nontrivial procedure for menus of two-part tariffs. For this family of mechanisms, standard procedures, such as rounding down the prices to multiples of $\alpha$, may result in arbitrary revenue loss because the price

parameters of each tariff decrease by different amounts affecting unpredictable changes in utilities of selecting each tariff and number of units. It would be possible that the utility-maximizing choice for a buyer switches from a higher-price tariff and more units (that originally has slightly higher utility for the buyer) to a low-price tariff and fewer units (that originally has slightly lower utility for the buyer) after a simple rounding.

Now, we provide structural results that enable us to design a discretization procedure. Given a menu of two-part tariffs, the following definition deletes the dominated tariffs (independent of the valuation).

**Definition 2** (Pareto frontier tariffs). *Given menu $M$ with distinct tariffs, the Pareto frontier of $M'$ is derived by deleting all tariffs $i$ for which there exists a tariff $j \neq i$ such that $p_1^{(j)} \leq p_1^{(i)}$ and $p_2^{(j)} \leq p_2^{(i)}$.*

**Lemma 3.** *Given a menu of tariffs, a user only selects a tariff in the Pareto frontier.*

**Lemma 4.** *Sorting the tariffs in the Pareto frontier in increasing order of $p_1$ is equivalent to sorting them in decreasing order of $p_2$.*

**Lemma 5.** *For any fixed number of units $k$, the highest utility tariff in $M$ is $\arg\min p_1^{(i)} + k p_2^{(i)}$. This is independent of the buyers' values.*

The following lemma states that as we increase the number of units the utility-maximizing tariff has higher $p_1$ and lower $p_2$.

**Lemma 6.** *Let $M'$ be the menu of Pareto frontier tariffs derived from menu $M$. Suppose the tariffs in $M'$ are reindexed in increasing order of $p_1$. Consider the index of the utility-maximizing tariff for each number of units. This index is increasing as a function of the number of units.*

*Proof of Theorem 1.* First, we reason about the length of the outcome menu. Let $\ell$ and $\ell'$ be the length of the original menu and outcome menu, respectively. First, note that $\ell'$ is also the length of the menu after rounding down $p_1^{(i)}$ and $p_2^{(i)}$ to their closest multiples of $\alpha$. Observe that $\ell'$ is at most $\ell$ (because we never add extra tariffs) and also at most $H^2/\alpha^2$ because there are $H/\alpha$ distinct options for each $p_1$ and $p_2$. Therefore, $\ell' \leq \min\{\ell, H^2/\alpha^2\}$.

Then, we reason about the maximum loss in revenue. First, note that for any fixed tariff and number of units, the total price decreases by at most $2K\ell'\alpha$. We only need to show that the buyer does not switch from buying more units to fewer. Switching in the opposite order does not decrease the revenue more than $2K\ell'\alpha$. The reason is that the total price of each tariff is an increasing function as the number of units. Therefore, the minimum total price is increasing as a function of the number of units. Next, we prove that a buyer never switches from buying more units to less. We show two cases: switching between tariffs and staying with the same tariff. In the first case, by Lemma 6, this means that that a buyer never switches from a tariff with higher $p_1$ (lower $p_2$) to a lower $p_1$ (higher $p_2$). Since in the discretization procedure, the price of tariffs with higher $p_1$ decreases more than lower $p_1$, the lower $p_1$ tariffs do not become utility-maximizing if they were not before. In the second case, by the rounding procedure, the total price of more units in the same tariff always decreases more; therefore, the lower number of units never becomes utility maximizing. Therefore, we conclude the payment of each tariff and therefore the revenue decreases at most by $2K\ell'\alpha$. Thus, $\text{Rev}(M') \geq \text{Rev}(M) - 2K\alpha\ell$.

Finally, we find the total number of possible menus. Also, after the discretization all $p_1^{(i)}$ and $p_2^{(i)}$ are multiples of $\alpha$. Therefore, when restricted to length-$\ell$ menus, there are $H/\alpha$ choices for each $2\ell$ parameter of the menu, making an upper bound of $(H/\alpha)^{2\ell}$. On the other hand, there are at most $H^2/\alpha^2$ possible tariffs, and each one of them may appear or not in the menu. Therefore, the number of menus is also bounded by $2^{H^2/\alpha^2}$. □

**Technical contribution.** The establishment of data-driven discretization (and the subsequent online learning and distributional learning algorithms) are in contrast with previous findings. For other data-driven algorithm design problems, such as data-driven clustering and data-driven learning to branch that share a similar piecewise structure in the utility functions, it has been proven that algorithms that use data-independent discretization could perform very poorly (Balcan et al., 2017; 2018a; 2023a). Thus, by contrast, our work shows the power of data-independent discretization for data-driven mechanism design and algorithm design more generally.

### 3.2 Online Learning

We provide bounded-regret online learning algorithms in full and partial information settings. Sections 3.2.1 to 3.2.3 provide online algorithms under adversarial input, under smooth distributions, and for limited type buyers, respectively. No online learning algorithms have been known previously for menus of two-part tariffs.

#### 3.2.1 Online Learning Under Adversarial Inputs

The main statements are Theorems 7 and 8 which provide regret guarantees for the full-information case and partial-information case, respectively. Using the discretization in Section 3.1, we show a reduction to a finite number of experts and run standard learning algorithms (weighted majority and Exp3) over the menus in the discretized set. Similar ideas were used in previous papers, for example Blum et al. (2004); Balcan et al. (2018b).

**Full Information.** In the full information setting, the seller sees the revenue generated for all the possible menus. To design an online algorithm in this case, we use a variant of the weighted majority algorithm by Auer et al. (1995). The experts in our case are the discretized menus from the previous section, denoted in the algorithm by set $X = m_1, \ldots, m_n$. Furthermore, $\boldsymbol{v}_t$ is the valuation of the buyer are time $t$ and $\text{Rev}_k(\boldsymbol{v}_1, \ldots, \boldsymbol{v}_t)$ is the cumulative revenue of menu $m_k$ for the buyers until time step $t$.

---

**Algorithm 2:** Full-information (Weighted majority on discretized menus)

---

**Input:** Set of menus (experts) $X = m_1, \ldots, m_n$, learning rate $\beta \in (0, 1]$.
1: **Initialize:** For each menu $m_k$, initialize $\text{Rev}_k() = 0$, $w_k(0) = 1$
2: **for** *buyer* $t = 1, \ldots, T$ **do**
  Select menu at time $t$ to be $m_k$ with probability $\pi_k[t] = \frac{w_k(t-1)}{\sum_{j=1}^n w_j(t-1)}$ ;
  Observe valuation of buyer $t$ as $\boldsymbol{v}_t$ ;
  For each menu $m_k$, update $\text{Rev}_k(\boldsymbol{v}_1, \ldots, \boldsymbol{v}_t)$ and $w_k(t) = (1 + \beta)^{\text{Rev}_k(\boldsymbol{v}_1, \boldsymbol{v}_2, \ldots, \boldsymbol{v}_t)/H}$;

---

**Theorem 7.** *In the full information case for length-$\ell$ menus of two-part tariffs, running Algorithm 2 over discretized set of menus specified in Theorem 1 for $\alpha = \beta = 1/\sqrt{T}$ has regret bounded by $\tilde{O}\left(\ell(K + H \ln H)\sqrt{T}\right)$, and running time $O(T\ell K \min\{H^{2\ell}T^\ell, 2^{H^2 T}\})$.*

The proof follows by combining the guarantees of the discretization procedure (Theorem 1) and previously known results (specifically Auer et al. (1995), Theorem 3.2) and is deferred to Appendix A.

**Partial Information (Bandit Setting).** In the partial information setting, the seller does not see the outcome for all the possible menus and only observes the outcome of the menu used (the tariff and number of units chosen by the buyer). To design an online algorithm in this case, we use a version of the Exp3 algorithm in Auer et al. (1995). This variant of the Exp3 algorithm contains the weighted majority algorithm (Algorithm 2) as a subroutine. At each step, we mix the probability distribution $\pi$, used by the weighted majority algorithm, with the uniform distribution to obtain a modified probability distribution $\overline{\pi}$, which is then used to select a menu from our discretized set. Following the tariff and the number of units chosen by buyer $t$, we use the price paid (the gain from the chosen menu) to formulate a simulated gain vector, which is then used to update the weights maintained by the weighted majority algorithm.

**Theorem 8.** *In the partial information case for length-$\ell$ menus of two-part tariffs, running Algorithm 3 over discretized set of menus in Theorem 1 for $\alpha = T^{-1/(2(1+\ell))}$, $\beta = \gamma = T^{-1/(4(1+\ell))}$ has regret bound $\tilde{O}\left(T^{1-\frac{1}{2(1+\ell)}}\ell(K + H^{2\ell+1})\right)$, and running time $O(T \min\{\min\{H^{2\ell}T^\ell, 2^{H^2 T}\}, 2^{H^2 T}\})$.*

The proof follows by combining the guarantees of the discretization procedure (Theorem 1) and previously known results (specifically (Auer et al., 1995), Theorem 4.1) and is deferred to Appendix A.

---

**Algorithm 3:** Partial-information (Exp3 on discretized menus)

---

**Input:** Set of menus (experts) $X = m_1, \ldots, m_n$, learning rate $\beta \in (0, 1]$, parameter $\gamma \in (0, 1]$.

1: **Initialize:** For each menu $m_k$, initialize $\mathrm{Rev}_k() = 0$, $w_k(0) = 1$
2: **for** *buyer* $t = 1, \ldots, T$ **do**

Select menu at time $t$ to be $m_k$ with probability $\overline{\pi}_k(t) = (1 - \gamma)\pi_k(t) + \gamma/n$ where
$\pi_k[t] = \frac{w_k(t-1)}{\sum_{j=1}^{n} w_j(t-1)}$ ;

For the selected menu $k^*$, set $g_{k^*}(t)$ to be the price paid by buyer $t$ (i.e., $g_{k^*}(t)$ is equal to $p_1^j + kp_2^j$, where $j$ and $k$ are the tariff index and quantity chosen by buyer $t$). Set $\overline{g}_{k^*}(t) = \frac{\gamma}{n}\frac{g_{k^*}(t)}{\overline{\pi}_{k^*}(t)}$;

For all other menus $k$, set $\overline{g}_k(t) = 0$; For all menus $k$, update $\mathrm{Rev}_k(t) = \mathrm{Rev}_k(t-1) + \overline{g}_k(t)$ and
$w_k(t) = (1 + \beta)^{\mathrm{Rev}_k(t)/H}$;

---

### 3.2.2 Online Learning Under Smooth Distributions

Recent papers studying online learning of mechanisms, e.g., Balcan et al. (2018b; 2020a), studied the problem in a restricted setting, where at each point in time, instead of a worst-case value, the value is drawn from a bounded-density distribution. This assumption is in the same spirit as the "smoothed analysis" paradigm of Spielman and Teng (Spielman and Teng, 2004) and is used in similar contexts in papers, including Cohen-Addad and Kanade (2017); Gupta and Roughgarden (2017). Specifically, we assume the buyers' valuations come from $\kappa$-*bounded* distributions, where the density function is bounded at all points by $\kappa$. This assumption has proved to be sufficient for a few classes of mechanisms, including posted-pricing and second-price mechanisms, to establish *dispersion*. At a high level, dispersion ensures that the number of discontinuities in a small ball in the parameter space is limited with high probability and is a sufficient condition for bounded-regret online algorithms. We prove that menus of two-part tariffs satisfy dispersion and use it to derive bounded-regret algorithms for full-information, bandit, and semi-bandit settings. The main difference between the algorithms used in this section compared to the adversarial input setting in Section 3.2.1 is that we previously needed to go through a careful data-independent discretization step (Section 3.1) to reduce the problem to a finite number of experts. However, under smooth distributions, the assumed properties of the distribution influence the set of experts chosen.

We provide the main results in this setting, followed by a discussion of the key ideas behind the algorithms and proofs. After establishing the dispersion constraint for menus of two-part tariffs, it is sufficient to employ previously known algorithms designed for dispersed settings to achieve no-regret guarantees. The primary purpose of this section is to compare the regret guarantees from the recently developed online learning technique of dispersion and the discretization approach discussed in the previous section. The formal definition of dispersion and technical descriptions of the algorithms and proofs are deferred to the appendix. The main results are as follows[2]:

**Definition 9.** *[$\kappa$-bounded] A density function $f : \mathbb{R} \to \mathbb{R}$ corresponds to a $\kappa$-bounded distribution if* $\max\{f(x)\} \leq \kappa$.

**Theorem 10.** *Let $u_1, \ldots, u_T : \mathcal{C} \to [0, H]$ be the revenue functions of two-part tariff menus such that $u_t(\boldsymbol{\rho})$ denotes the revenue of a mechanism associated with menu parameters $\boldsymbol{\rho}$ for the buyer arriving at time $t$. Let the samples of buyers' values be drawn from $\mathcal{S} \sim \mathcal{D}^{(1)} \times \cdots \times \mathcal{D}^{(T)}$. Suppose $v(k) \in [0, H]$ for any number of units $k \in [K]$. Also, suppose that for each distribution $\mathcal{D}^{(t)}$, and every pair of number of units $k$ and $k'$, $v(k)$ and $v(k')$ have a $\kappa$-bounded joint distribution. An efficient implementation of the exponentially weighted forecaster with $\lambda = \sqrt{2\ell \ln(2H^2\kappa\sqrt{T})/T}/H$ (Algorithm 4) has expected regret bounded by $\tilde{O}((H\ell^2 K^2\sqrt{\log \kappa} + 1/(H\kappa))\sqrt{T})$ and runs in time $\tilde{O}((T+1)^{poly(\ell, K)}poly(\ell, \sqrt{T}) + KT\sqrt{T})$.*

**Theorem 11.** *Let $u_1, \ldots, u_T : \mathcal{C} \to [0, H]$ be the revenue functions of two-part tariff menus such that $u_t(\boldsymbol{\rho})$ denotes the revenue of a mechanism associated with menu parameters $\boldsymbol{\rho}$ for the buyer arriving at time $t$. Let*

---

[2]The regret term in the semi-bandit algorithm (Theorem 12) is smaller than the full-information algorithm (Theorem 10) since different notions of dispersion are used. Also, the stated running time of both algorithms are the same; however, this is in the worst case, and the semi-bandit algorithm potentially performs fewer computations.

*the samples of buyers' values be drawn from $\mathcal{S} \sim \mathcal{D}^{(1)} \times \cdots \times \mathcal{D}^{(T)}$. Suppose $v(k) \in [0, H]$ for any number of units $k \in [K]$. Also, suppose that for each distribution $\mathcal{D}^{(t)}$, and every pair of number of units $k$ and $k'$, $v(k)$ and $v(k')$ have a $\kappa$-bounded joint distribution. There is a bandit-feedback online optimization algorithm with expected regret $\tilde{O}\left(T^{(2\ell+1)/(2\ell+2)}\left(H^2 K \sqrt{\ell} \kappa^{d/2} \sqrt{\log \kappa}\right) + 1/H\kappa + H\ell^2 K^2\right)$. The per-round running time is $O(H^{4\ell} \kappa^{2\ell} T^\ell)$.*

**Theorem 12.** *Suppose the buyers' values are drawn from $\mathcal{D}^{(1)} \times \cdots \times \mathcal{D}^{(T)}$, where each $\mathcal{D}^{(t)}$ is $\kappa$-bounded for $\kappa = \tilde{o}(T)$. Then, running the continuous Exp3-SET algorithm (Algorithm 7) for menus of two-part tariffs under semi-bandit feedback has expected regret bounded by $\tilde{O}(H\sqrt{\ell T})$. An efficient implementation has the same regret bound and running time $\tilde{O}((T+1)^{poly(\ell,K)} poly(\ell, \sqrt{T}) + KT\sqrt{T})$.*

**Smoothed Distributional Assumptions.** In an online setting under smoothed distributions, the algorithm receives samples $\mathcal{S} \sim \mathcal{D}^T$, where $\mathcal{D}$ is an arbitrary distribution over problem instances $\Pi$ (which in our case is the buyer valuations). The goal is to find $\hat{\boldsymbol{\rho}}$ that nearly maximizes $\sum_{\boldsymbol{v} \in \mathcal{S}} u(\boldsymbol{v}, \boldsymbol{\rho})$. In this setting, the goal is to find a value $\boldsymbol{\rho}$ that is nearly optimal in hindsight over a stream $\boldsymbol{v}_1, \ldots, \boldsymbol{v}_T$ of instances, or equivalently, over a stream $u_1 = u(\boldsymbol{v}_1, \cdot), \ldots, u_T = u(\boldsymbol{v}_T, \cdot)$ of functions. Each $\boldsymbol{v}_t$ is drawn from a distribution $\mathcal{D}^{(t)}$, which may be adversarial. Therefore, $\{\boldsymbol{v}_1, \ldots, \boldsymbol{v}_T\} \sim \mathcal{D}^{(1)} \times \cdots \times \mathcal{D}^{(T)}$.

**Dispersion.** Let $u_1, \ldots, u_T$ be a set of functions mapping a set $\mathcal{C} \subseteq \mathbb{R}^d$ to $[0, H]$. In this paper, we study the mechanism selection setting, given a collection of problem instances $\boldsymbol{v}_1, \ldots, \boldsymbol{v}_T \in \Pi$ and a utility function $u : \Pi \times \mathcal{C} \to [0, H]$, each function $u_i(\cdot)$ might equal the function $u(\boldsymbol{v}_i, \cdot)$, measuring a mechanism's performance on a fixed problem instance as a function of its parameters. Informally, dispersion is a constraint on the functions $u_1, \ldots, u_T$ that guarantees although each function $u_i$ may have discontinuities, they do not concentrate in a small region of space. We study two definitions of dispersion previously introduced in algorithm and mechanism selection problems. We show that menus of two-part tariffs satisfy both definitions; $(w, k)$-dispersion (Definition 15) and $\beta$-dispersion (Definition 38). Then, we use the first to establish online learning results for full-information and bandit settings and the second for the semi-bandit setting.

In order to prove menus of two-part tariffs satisfy dispersion under smoothed assumptions, we show this family of mechanisms satisfies certain structural properties. Balcan et al. (2018c) show in two-part tariff menus, for each function $u_i$, the parameter space $\mathcal{C}$ is partitioned into sets $\mathcal{P}_1, \ldots, \mathcal{P}_n$ such that $u_i$ is $L$-Lipschitz on each piece, but $u_i$ may have discontinuities at the boundaries between pieces.[3] We refine this structural property and show that multi-sets of parallel hyperplanes, corresponding to the stream of buyer valuations, partition the parameter space $\mathcal{C}$ into convex polytopes with bounded-degree linear utility functions inside each polytope. Later, we show this property is sufficient for proving dispersion and employing the related algorithms.

**Partitioning of parameter space to convex regions with linear utilities (Balcan et al., 2018c).** Consider the sequence of buyers valuations $\bar{b}$. At each time step, a buyer is presented with a menu, and based on the menu and their valuation, they select the tariff index and number of units that maximize their utility. Formally, given menu $\boldsymbol{\rho}$, buyer $i$ with valuation $\boldsymbol{b}_i$ selects option $(j, k)$, where $j$ is the tariff index and $k$ is the number of units if this option produces more utility for the buyer than any other options. Concretely,

$$b_i(k) - \mathbb{I}\{k \geq 1\} \left(p_1^{(j)}(\boldsymbol{\rho}) + k p_2^{(j)}(\boldsymbol{\rho})\right) \geq b_i(k') - \mathbb{I}\{k' \geq 1\} \left(p_1^{(j')}(\boldsymbol{\rho}) + k' p_2^{(j')}(\boldsymbol{\rho})\right) \quad \forall j', k' \qquad (1)$$

where $p_1^{(j)}(\boldsymbol{\rho})$ and $p_2^{(j)}(\boldsymbol{\rho})$ are the up-front fee and per-unit fee of tariff $j$ in menu $\boldsymbol{\rho}$. The above inequalities identify a convex polytope of parameter vectors (menus $\boldsymbol{\rho}$) with hyperplane boundaries. Since the tariff index and the number of units that $\boldsymbol{b}_i$ selects are fixed in the region, the revenue, $\mathbb{I}\{k \geq 1\} \left(p_1^{(j)}(\boldsymbol{\rho}) + k p_2^{(j)}(\boldsymbol{\rho})\right)$, is continuous and more specifically linear in the region (formally proved in Lemma 20). Following the same argument for the buyers in the sequence, the parameter space for each buyer is partitioned into convex polytopes where the revenue for the buyer's valuation is linear inside the polytopes. By superimposing these partitionings, since the intersections of convex regions are also convex, and the sum of linear functions (here revenues) is linear, the parameter space, $\mathcal{C}$ is partitioned into convex regions such that the cumulative revenue

---

[3]This previously-known structural result suffices for the techniques used in the setting with the limited number of buyers (Section 3.2.3 and appendix A.1.3); however, we need a refined statement for proving dispersion.

for the sequence is linear in each region. Inside each region, the utility-maximizing choice of each buyer is fixed; therefore, each region is associated with a *mapping* from buyer valuations to their corresponding utility-maximizing tariff index and number of units. We may use the mapping, formally defined in Section 3.2.3, to denote the region, e.g., region $\mathcal{P}_\mu$ corresponding to mapping $\mu$, or simply use cardinal indices for the regions $\mathcal{P}_1, \mathcal{P}_2, \ldots$.

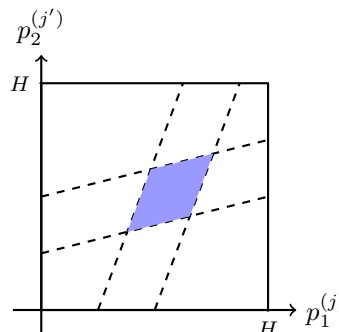

Figure 1: The figure is an abstraction of the regions for parameter space of two-part tariffs drawn in two dimensions for illustration. The coordinates are the up-front and per-unit fees for the tariff indices. The dashed hyperplanes correspond to a buyer valuation having the same utility through two pairs of tariff indices and the number of units; see Equation (1). The colored region area is defined by hyperplane boundaries. Inside each such region, any buyer valuation selects a fixed tariff index and the number of units, resulting in a linear cumulative revenue function.

**Lemma 13.** *Consider the sequence of buyer valuations $\overline{v}$ arrived until time $t$. For menus of two-part tariffs, the parameter space $\mathcal{C}$ is partitioned into convex polytopes, $\mathcal{P}_1, \ldots, \mathcal{P}_n$ by multisets of parallel hyperplanes, such that the utility function at each time step inside each region $\mathcal{P}_j$ is a linear function satisfying $(K + 1)$-Lipschitz continuity.*

*Proof.* Part of the proof that identifies the regions with linear utilities has been shown previously in Balcan et al. (2018c), Lemma 3.15. We reiterate that part for completeness and also prove the extra structural properties, i.e., parallel hyperplanes and $(K + 1)$-Lipschitz continuity. Consider the set of menus for which the buyer with valuation $\boldsymbol{v}^{(i)}$ arriving at time $i$ selects the tariff index $j$ and the number of units $k$. The buyer selects this option for menu $\boldsymbol{\rho}$ if it produces more utility for the buyer than any other option. Formally,

$$v^{(i)}(k) - \mathbb{I}\{k \geq 1\} \left( p_1^{(j)}(\boldsymbol{\rho}) + k p_2^{(j)}(\boldsymbol{\rho}) \right) \geq v^{(i)}(k') - \mathbb{I}\{k' \geq 1\} \left( p_1^{(j')}(\boldsymbol{\rho}) + k' p_2^{(j')}(\boldsymbol{\rho}) \right). \quad \forall j', k' \qquad (2)$$

The above inequalities identify a convex polytope of parameter vectors (menus $\boldsymbol{\rho}$) with hyperplane boundaries. Considering all the possible selections $(j, k)$ (the tariff index and the number of units), the parameter space for $\boldsymbol{v}^{(i)}$ is partitioned into convex polytopes where inside each polytope the payment of $\boldsymbol{v}^{(i)}$ is linear; i.e., $\mathbb{I}\{k \geq 1\} \left( p_1^{(j)}(\boldsymbol{\rho}) + k p_2^{(j)}(\boldsymbol{\rho}) \right)$. Considering the same analysis for all the buyers' valuations in the sequence, for each buyer, the parameter space is partitioned into convex polytopes where inside each polytope, the revenue function is linear and $(K + 1)$-Lipschitz. Since convex polytopes are closed under intersection, superimposing the partitions for $i = 1, \ldots, t$ results in polytopes with the properties in the statement.

For a fixed valuation vector $\boldsymbol{v}^{(i)}$, the discontinuities in the utility function are defined by at most $\ell^2 K^2$ hyperplanes: $v^{(i)}(k) - \mathbb{I}\{k \geq 1\} \left( p_1^{(j)}(\boldsymbol{\rho}) + k p_2^{(j)}(\boldsymbol{\rho}) \right) = v^{(i)}(k') - \mathbb{I}\{k' \geq 1\} \left( p_1^{(j')}(\boldsymbol{\rho}) + k' p_2^{(j')}(\boldsymbol{\rho}) \right)$. Let $\Psi_{\boldsymbol{v}}$ be the multi-set union of all these hyperplanes. Consider a set $\mathcal{S} = \left\{ \boldsymbol{v}^{(1)}, \ldots, \boldsymbol{v}^{(t)} \right\}$ with corresponding multi-sets $\Psi_{\boldsymbol{v}^{(1)}}, \ldots, \Psi_{\boldsymbol{v}^{(t)}}$ of hyperplanes. We now partition the multi-set union of $\Psi_{\boldsymbol{v}^{(1)}}, \ldots, \Psi_{\boldsymbol{v}^{(t)}}$ into at most $\ell^2 K^2$ multi-sets $\mathcal{B}_{j,k,j',k'}$ for all $j, j' \in [\ell]$ and $k, k' \in [K]$ and $i \in [t]$ such that for each $\mathcal{B}_{j,k,j',k'}$, the hyperplanes in $\mathcal{B}_{j,k,j',k'}$ are parallel with probability 1 over the draw of $\mathcal{S}$. To this end, define a single multi-set $\mathcal{B}_{j,k,j',k'}$ to consist of the hyperplanes

$$\{v^{(1)}(k) - \mathbb{I}\{k \geq 1\} \left( p_1^{(j)}(\boldsymbol{\rho}) + k p_2^{(j)}(\boldsymbol{\rho}) \right) = v^{(1)}(k') - \mathbb{I}\{k' \geq 1\} \left( p_1^{(j')}(\boldsymbol{\rho}) + k' p_2^{(j')}(\boldsymbol{\rho}) \right),$$

$$v^{(2)}(k) - \mathbb{I}\{k \geq 1\} \left( p_1^{(j)}(\boldsymbol{\rho}) + k p_2^{(j)}(\boldsymbol{\rho}) \right) = v^{(2)}(k') - \mathbb{I}\{k' \geq 1\} \left( p_1^{(j')}(\boldsymbol{\rho}) + k' p_2^{(j')}(\boldsymbol{\rho}) \right),$$

$$\ldots,$$

$$v^{(t)}(k) - \mathbb{I}\{k \geq 1\} \left( p_1^{(j)}(\boldsymbol{\rho}) + k p_2^{(j)}(\boldsymbol{\rho}) \right) = v^{(t)}(k') - \mathbb{I}\{k' \geq 1\} \left( p_1^{(j')}(\boldsymbol{\rho}) + k' p_2^{(j')}(\boldsymbol{\rho}) \right)\};$$

where the only variables are coordinates of $\boldsymbol{\rho}$. The hyperplanes inside each multi-set are parallel and the utility of the regions defined by the hyperplanes are linear and $K + 1$-Lipschitz.[4] $\qquad\square$

Next, we establish an upper bound on the number of regions with continuous (linear) regions.

**Lemma 14.** *The partitioning of the parameter space for menus of two-part tariffs explained in Lemma 13 after $T$ rounds results in $O((T + 1)^{\ell^2 K^2})$ regions, with linear cumulative utility function inside each region.*

*Proof.* Lemma 13 identifies multi-sets $\mathcal{B}_{j,k,j',k'}$ of size $T$ for each $j, k, j', k'$ such that the hyperplanes inside the multi-sets are parallel. Therefore, each multi-set divides the parameter space into $T + 1$ parts. Thus, each region with continuous utility can be defined as the intersection at most $\ell^2 K^2$ parts, where each part corresponds to a distinct multi-set. This results in at most $O((T + 1)^{\ell^2 K^2})$ such regions. $\qquad\square$

**Dispersion for menus of two-part tariffs.** We provide intuition as to why menus of two-part tariffs for bounded density distributions satisfy dispersion; that is, the discontinuities in the revenue function do not concentrate with high probability. To prove this, we focus on Equation (1) for fixed values of $j, k, j', k'$, i.e., pairs of tariffs and units, and for all $\boldsymbol{b}_i \in \bar{b}$. The equalities for all of these equations are met at parallel hyperplanes because, for each $\boldsymbol{\rho}$ and fixed pairs of tariffs and units, other parameters, i.e., $k, k', p_1^{(j)}, p_2^{(j)}, p_1^{(j')}, p_2^{(j')}$ are fixed, and the equations are only different in $\boldsymbol{b}_i$. Assuming independence of distributions among buyers and $\kappa$-bounded joint distributions over $b_i(k)$ and $b_i(k')$, with high probability the intersection of multisets of parallel hyperplanes, defined by Equation (1) do not concentrate, implying dispersion.

We first provide the formal definition of $(w, k)$-dispersion. Recall that $\Pi$ is a set of instances, $\mathcal{C} \subset \mathbb{R}^d$ is a parameter space, and $u$ is an abstract utility function. We use the $l_2$ distance and let $B(\boldsymbol{\rho}, r) = \{\boldsymbol{\rho}' \in \mathbb{R}^d : \|\boldsymbol{\rho} - \boldsymbol{\rho}'\|_2 \le r\}$ denote a ball of radius $r$ centered at $\boldsymbol{\rho}$. We use this notion of dispersion to derive our full-information and bandit setting results.

**Definition 15** ((Balcan et al., 2018b), $(w, k)$-dispersion). *Let $u_1, \ldots, u_T : \mathcal{C} \to [0, H]$ be a collection of functions where $u_i$ is piecewise Lipschitz over a partition $\mathcal{P}_i$ of $\mathcal{C}$. We say that $\mathcal{P}_i$ splits a set $A$ if $A$ intersects with at least two sets in $\mathcal{P}_i$. The collection of functions is $(w, k)$-dispersed if every ball of radius $w$ is split by at most $k$ of the partitions $\mathcal{P}_1, \ldots, \mathcal{P}_T$. More generally, the functions are $(w, k)$-dispersed at a maximizer if there exists a point $\boldsymbol{\rho}^* \in \operatorname{argmax}_{\boldsymbol{\rho} \in \mathcal{C}} \sum_{i=1}^T u_i(\boldsymbol{\rho})$ such that the ball $B(\boldsymbol{\rho}^*, w)$ is split by at most $k$ of the partitions $\mathcal{P}_1, \ldots, \mathcal{P}_T$.*

We now prove menus of two-part tariffs satisfy $(w, k)$-dispersion and use it to derive no-regret online learning results for full-information and bandit settings.

**Proposition 16.** *Suppose that $u(\boldsymbol{v}, \boldsymbol{\rho})$ is the revenue of the two-part tariff menu mechanism with prices $\boldsymbol{\rho}$ and buyer's values $\boldsymbol{v}$. With probability at least $1 - \zeta$ over the draw $\mathcal{S} \sim \mathcal{D}^{(1)} \times \cdots \times \mathcal{D}^{(T)}$ for any $\alpha \ge 1/2$ the following statement holds:*

*Suppose $v(k) \in [0, H]$ for any number of units $k \in [K]$. Also, suppose that for each distribution $\mathcal{D}^{(t)}$, and every pair of number of units $k$ and $k'$, $v(k)$ and $v(k')$ have a $\kappa$-bounded joint distribution. Then $u$ is*

$$\left( \frac{1}{2H\kappa T^{1-\alpha}}, O\left( \ell^2 K^2 T^\alpha \sqrt{\ln \frac{\ell K}{\zeta}} \right) \right) \text{-dispersed}$$

*with respect to $\mathcal{S}$.*

*Proof.* Lemma 13 gives multisets of parallel hyperplanes that partition the parameter space into regions with $K + 1$-Lipschitz continuous utility functions. Since the samples are drawn independently from $\kappa$-bounded distributions with support $[0, H]$, the offsets of the hyperplanes in each multiset $\mathcal{B}_{j,k,j',k'}$ are independent random variables with $H\kappa$-bounded distributions. Furthermore, the number of multisets is at most $\ell^2 K^2$. Using these properties, Theorem 32 of Balcan et al. (2018b) gives the statement. $\qquad\square$

---

[4]Partitioning of the parameter space by parallel multisets of hyperplanes has been established before for other families of mechanism design such as posted pricing (Balcan et al., 2018b). We extend this idea to the more complicated case of two-part tariffs.

After establishing dispersion and showing that the parameter space is partitioned into convex regions with cumulative linear utility inside each region, the no-regret guarantees and their performances are implied by prior results.

---

**Algorithm 4:** Full-information online learning of two-part tariffs under smoothed distributional assumptions (Adapted to two-part tariffs from (Balcan et al., 2018b), Algorithm 4)

---

**Input:** $\lambda \in (0, 1/H]$, $\eta, \zeta \in (0, 1)$.

1: Set $u_0(\cdot) = 0$ (to be the constant 0 function over $\mathcal{C}$).
2: **for** $buyer\ t = 1, 2, \ldots, T$ **do**

    Present menu $\boldsymbol{\rho}_t$ sampled with probability approximately proportional to $e^{g(\boldsymbol{\rho}_t)}$ to the buyer, where where, $g(\cdot) = \lambda \sum_{s=0}^{t-1} u_s(\cdot)$. (Use Algorithm 6, with approximation parameter $\eta/4$ and confidence parameter $\zeta/T$).;

    Observe the revenue for all the potential menus as function $u_t(\cdot)$. Receive payment $u_t(\boldsymbol{\rho}_t) = \mathbb{I}\{k \geq 1\}(p_1^{(i)}(\boldsymbol{\rho}_t) + kp_2^{(i)}(\boldsymbol{\rho}_t))$, where $i$ and $k$ are the tariff index and the number of units chosen by buyer $t$ respectively given menu $\boldsymbol{\rho}_t$.

---

**Overview of Algorithms.** We provide high-level ideas for the full-information, bandit, and semi-bandit setting algorithms used for Theorems 10 to 12, respectively. Generic forms of these algorithms were devised by Balcan et al. (2018b; 2020a) for dispersed families of algorithms. The full information algorithm considers the cumulative revenue function up until the time $t-1$ over the parameter space and samples the menu to present at time $t$ proportional to an exponential function of its cumulative revenue. In order to have an efficient implementation, they use techniques from high-dimensional geometry and approximately sample menu $\boldsymbol{\rho}_t$. Let $\mathcal{P}_1, \ldots, \mathcal{P}_n$ be the partition of $\mathcal{C}$ until time $t$. The algorithm picks $\mathcal{P}_i$ with probability approximately proportional to the region's cumulative weight and outputs a sample from the conditional distribution of menus in $\mathcal{P}_i$. The bandit-setting algorithm considers a grid over the parameter space, whose granularity depends on the dispersion parameters, and runs the Exp3 algorithm over menus corresponding to the grid. The semi-bandit setting algorithm is a continuous version of the Exp3-SET algorithm of Alon et al. (2017b). At each time step, the algorithm learns the revenue function (only) inside the region $\mathcal{P}_i$ that the presented menu belongs to and updates the menu weights for the next round accordingly.

**Comparison to the results in Section 3.2.1 .** Although the discretization-based algorithms work under adversarial inputs and are more general, they provide similar regret bounds and even improved running times in some cases. In the full information case, the dependence on the regret bound in parameter $T$ is similar in both algorithms. In running time, the discretization-based algorithm suffers worse dependence in $H$, but enjoys better dependence in $T$ and $K$ (the maximum number of units) compared to the dispersion-based algorithm. In the bandit setting, similarly, the regret bounds are similar in their dependence on $T$, while the running-time comparison depends on the value of $\kappa$ (maximum density under smoothness assumption) such that lower-density distributions may result in better running times.

**Comparison to prior work.** For menus of two-part tariffs, it has been shown in Balcan et al. (2018b)that based on the values observed from users until time $t$, the parameter space is partitioned into convex regions with hyperplane boundaries such that the utility inside each region satisfies Lipschitz continuity. We give a more refined characterization by showing that (1) the utility function inside each region is linear, and (2) the boundary hyperplanes constitute a multiset of parallel hyperplanes. Properties (1) and (2) are important for establishing dispersion and obtaining no-regret online learning algorithms under smooth distributional assumptions, as in Theorems 10 and 11. After establishing dispersion, we use previously developed results, i.e., regret bounds for dispersed settings, from prior work (e.g., Theorem 1 in Balcan et al. (2018b) for full information and Theorem 3 in (Balcan et al., 2018b) for bandit setting). The algorithms for full-information, semi-bandit, and bandit settings were previously developed in a general format (Balcan et al., 2018b; 2020a) for any problem setting satisfying dispersion property. We adapt those algorithms to our settings in Algorithms 4, 6 and 7.

### 3.2.3 Limited Buyer Types

In this section, we assume that there are a finite number of known buyer types. This information provides extra structures compared to the general setting considered previously. In particular, now the mechanism designer is aware of where the potential discontinuities happen as a function of the parameter space. We provide algorithms with bounded regrets both for the full information and partial information settings specific to limited types. These algorithms improve the regret bounds significantly when the number of buyer types is small. This section is inspired by Balcan et al. (2015) and includes similar algorithms and notations.

Balcan et al. (2015) study a security games setting, in which at each time step, the *defender* has a mixed strategy (a probability distribution) for protecting the *attack targets*. Knowing this mixed strategy, the attacker selects a target to attack, which maximizes the attacker's utility (depending on the attacker's type). Considering the target selected by each attacker type as a function of the defender's mixed strategy, the mixed strategy space is partitioned into regions where the action of each attacker type is fixed throughout each region. This is very similar to our setting, where the parameter space is partitioned into regions, where inside each region, each buyer type selects a fixed tariff index and the number of units (see the discussion on partitioning the parameter space in Section 3.2.2). Balcan et al. use the linear structure of the utility function inside each region to develop a no-regret full-information algorithm. In the partial information setting, other than the linearity of utility functions, they use the dependence of an agent's (in their case, attacker, and in our case, buyer) actions across different regions and identify a limited number of mixed strategies (corresponding to menus in our case) such that observing the agent's response to them suffice to estimate the utility of other strategies. We use similar machinery in both the full and partial information settings. However, the source of linearity of the utility is different across the two settings. In the security games context, the attacker's action corresponds to a fixed coordinate axis in the parameter space, and the utility is defined as a fixed linear function of that coordinate. In our setting, however, the utility depends on multiple coordinates, and its formula depends on the buyer's choice. Nevertheless, we show the cumulative utility is a linear function of coordinates (See Lemma 20). For completeness and to make the paper self-contained, we include a full description of the algorithms and techniques adapted to our setting and using our terminology.

In this setting, we utilize the knowledge of the potential buyer types to design a limited number of menus and optimize over this set. In contrast to the previous section, where the valuations were realized after the arrival of the buyers, here, we have access to all potential buyer types up-front, but similarly, as discussed in Section 3.2.2, the piecewise linear structure of the utility for the buyers partition the parameter space such that each part has linear cumulative utility (Balcan et al., 2018c). This partitioning is equivalent to dividing the parameter space into convex regions such that in each region, there is a fixed *mapping* from the buyer types to the menu options that each buyer selects. We show that in each region, we need to consider only a limited number of menus, namely the extreme points.

Consider $\boldsymbol{v}_1, \ldots, \boldsymbol{v}_V$ as the set of all potential buyer valuations. $V$ denotes the number of buyer types. In order to define the behavior of buyers in each region, we need to define a concept called *menu options*, which determines the buyers' choices.

**Definition 17** (menu option for menus of two-part tariffs, $\mathcal{O}$)**.** *A pair $(j, k)$, where $j$ is the tariff index $1, \ldots, \ell$, and $k$ is the number of units $0, 1, \ldots, K$ is a menu option. We denote the set of all menu options as $\mathcal{O}$. This set identifies all potential actions of a buyer when presented with a menu.*

**Definition 18** (mapping $\mu$, feasible mappings, $\mathcal{P}_\mu$)**.** *A mapping $\mu$ is a function from buyer types, $\boldsymbol{v}_1, \ldots, \boldsymbol{v}_V$ to menu options $(j, k)$, where $j$ and $k$ are the tariff index and the number of units assigned to the buyer type respectively. Mapping $\mu$ is feasible if there is a menu corresponding to the mapping, i.e., a menu that, if presented to the buyers, each buyer selects their corresponding option in the mapping as their utility maximizing option. $\mathcal{P}_\mu$ denotes the region of the parameter space corresponding to $\mu$, i.e., the set of menus inducing mapping $\mu$.*

Using the discussion in Section 3.2.2, the parameter space is partitioned to convex polytopes, each with a linear utility function for any sequence of buyer types. We reiterate this result in Lemmas 19 and 20, adapting the statements to the limited buyer type setting and corresponding notations.

**Lemma 19.** *For each feasible mapping $\mu$, as defined in Definition 18, $\mathcal{P}_\mu$ is a convex polytope with hyperplane boundaries.*

*Proof.* The statement is a corollary of Lemma 13. For a fixed buyer type $\boldsymbol{i}$ and option $(j, k)$, let $\mathcal{P}_{(j,k)}^{(i)}$ be the set of all parameter vectors $\boldsymbol{\rho}$ corresponding to the length-$\ell$ menus that buyer type $i$ selects option $(j, k)$. The buyer selects option $(j, k)$ for menu $\boldsymbol{\rho}$ if this option produces more utility for the buyer than any other option. Formally,

$$v_i(k) - \mathbb{I}\{k \geq 1\} \left( p_1^{(j)}(\boldsymbol{\rho}) + k p_2^{(j)}(\boldsymbol{\rho}) \right) \geq v_i(k') - \mathbb{I}\{k' \geq 1\} \left( p_1^{(j')}(\boldsymbol{\rho}) + k' p_2^{(j')}(\boldsymbol{\rho}) \right). \quad \forall j', k'$$

The above inequalities identify a convex polytope of parameter vectors (menus $\boldsymbol{\rho}$) with hyperplane boundaries. $\mathcal{P}_\mu$ is the intersection of $\mathcal{P}_{\mu(i)}^{(i)}$ for $i = 1, \ldots, V$. Therefore, $\mathcal{P}_\mu$ is also a convex region with hyperplane boundaries. $\square$

**Lemma 20.** *For each feasible mapping $\mu$ and any sequence of buyer valuations $\overline{b}$ the cumulative utility, $\sum_i u(\boldsymbol{b}_i, \boldsymbol{\rho})$, is linear in $\mathcal{P}_\mu$.*

*Proof.* Before presenting the proof, we point out the difference between the proof of linearity in Balcan et al. (2015) and in this lemma. In Balcan et al. (2015), in each region, the attacker (corresponding to buyer in our case) chooses a target. There is a one-to-one correspondence between targets and coordinate indices of the parameter space. The utility is defined as a fixed linear function of the corresponding coordinate; immediately implying its linearity in the parameter space in each region. In our setting, however, the utility depends on multiple coordinates and its formula depends on the buyer's choice.

The proof builds on Lemma 13. We show that for any buyer valuation $\boldsymbol{v}_i$ in the sequence, $u(\boldsymbol{v}_i, \rho)$ is linear in the region. Proving this claim is sufficient for concluding the statement. Let $(j, k) = \mu(\boldsymbol{v}_i)$, i.e., $j$ is the tariff index and $k$ is number of units that buyer valuation $\boldsymbol{v}_i$ selects under $\mu$. Therefore, the utility for the mechanism designer for menu $\boldsymbol{\rho} \in \mathcal{P}_\mu$ is $\mathbb{I}\{k \geq 1\} \left( p_1^{(j)}(\boldsymbol{\rho}) + k p_2^{(j)}(\boldsymbol{\rho}) \right)$. Both $p_1^{(j)}(\boldsymbol{\rho})$ and $p_2^{(j)}(\boldsymbol{\rho})$ grow linearly as a function of $\boldsymbol{\rho}$. Therefore, since the option that each buyer valuation selects (the tariff index and the number of units) is fixed inside $\mathcal{P}_\mu$, the utility is also linear. $\square$

After establishing the partitioning of parameter space into convex polytopes with linear utilities, for optimization purposes, it seems enough only to consider menus corresponding to the extreme points. This intuition is accurate conditioned on a small tweak. Depending on the tie-breaking rule of buyers among menu options producing the same utility, the polytopes $\mathcal{P}_\mu$ may not be closed. Therefore, depending on the tie-breaking rule, we consider a menu in proximity to the extreme point but inside the polytope.

**Definition 21** ($\mathcal{E}$, extended set of extreme points (Balcan et al., 2015))**.** *For a given $\varepsilon > 0$, set $\mathcal{E}$ is the set of menus as follows: for any $\mu$ and any $\boldsymbol{\rho}$ that is an extreme point of the closure of $\mathcal{P}_\mu$, if $\boldsymbol{\rho} \in \mathcal{P}_\mu$, then $\boldsymbol{\rho} \in \mathcal{E}$, otherwise, there exists $\boldsymbol{\rho}' \in \mathcal{E}$ such that $\boldsymbol{\rho}' \in \mathcal{P}_\mu$ and $||\boldsymbol{\rho} - \boldsymbol{\rho}'||_1 \leq \varepsilon$. From now on, we may refer to $\mathcal{E}$ as the extreme points.*

**Lemma 22.** *The number of extreme points, $|\mathcal{E}|$ is at most $(V\ell^2 K^2/4)^{2\ell}$.*

*Proof.* Length-$\ell$ menus of two-part tariffs occupy a $2\ell$-dimensional parameter space. In each $d$-dimensional space, an extreme point is the intersection of $d$ linearly independent hyperplanes. The total number of hyperplanes defining the regions is $\mathcal{H} = V\binom{\ell}{2}\binom{K}{2}$, where for each buyer type compares the utility of any pair of options, i.e., the number of units $0, \ldots, K$ and tariff indices $1, \ldots, \ell$. Out of these hyperplanes, we need $2\ell$ of them to intersect to form an extreme point. Therefore, the number of extreme points is at most $\binom{\mathcal{H}}{2\ell}$, implying the statement. $\square$

The following lemma bounds the loss in utility where the set of menus is limited to the extreme points $\mathcal{E}$. The proof is similar to Balcan et al. (2015); however, the loss depends on the problem-specific utility functions.

**Lemma 23.** *Let $\mathcal{E}$ be as defined in Definition 21, then for any sequence of buyer valuations $\bar{b} = \boldsymbol{b}_1, \ldots, \boldsymbol{b}_T$, and $\boldsymbol{\rho}^*$ as the optimal menu in the hindsight:*

$$max_{\boldsymbol{\rho} \in \mathcal{E}} \sum_{t=1}^{T} u(\boldsymbol{b}_t, \boldsymbol{\rho}) \geq \sum_{t=1}^{T} u(\boldsymbol{b}_t, \boldsymbol{\rho}^*) - 2K\varepsilon T.$$

*Proof.* The proof consists of a few simple steps: (i) since the mappings partition the space into regions with a fixed mapping, there exists a mapping $\mu$ such that $\boldsymbol{\rho}^* \in P_\mu$, (ii) the revenue of the buyer valuation sequence is linear in $P_\mu$ as shown in Lemma 20, (iii) the closure of $P_\mu$ is a convex polytope whose extreme points contain the maximizers of the linear function $\sum_{\boldsymbol{b}_i \in \bar{b}} u(\boldsymbol{b}_i, \boldsymbol{\rho})$, (iv) one of the maximizers has cumulative utility at least as $\boldsymbol{\rho^*}$, (v) the parameter vectors in $\varepsilon$ proximity of the extreme point inside $\mathcal{P}_\mu$ approximately preserve the revenue of the extreme points, (vi) since by definition of $\mathcal{E}$ the $L1$ distance of each member to an extreme point is at most $\varepsilon$, there is at most $\varepsilon$ distance in the upfront fee and per-unit fee for any tariffs, resulting in the bound in the statement. $\square$

**Full Information.** We first provide an algorithm for the full information case specific to the finite number of buyers. The main result of this section is provided below. The algorithm to achieve this regret guarantee is a weighted majority algorithm (Algorithm 2) on the set of menus corresponding to the extreme points $\mathcal{E}$.

**Theorem 24.** *In the full information case for length-$\ell$ menus of two-part tariffs, when there are $V$ types of buyers, running Algorithm 2 over the set of menus corresponding to set $\mathcal{E}$ for $\beta = 1/\sqrt{T}$ has regret bounded by $\tilde{O}(H\ell\sqrt{T}\ln(V\ell K))$.*

The proof follows from Lemma 23 and the guarantee of weighted majority algorithm and is deferred to the appendix.

**Partial Information (bandit).** In the partial information setting, in each time step $t$, we present the arriving buyer a menu and only observe the option selected by the buyer (e.g., the tariff and the number of units) in the presented menu. A natural approach in this setting is running the EXP3 algorithm and using the weighted majority algorithm for the full information case as a subroutine. However, this approach leads to a regret bound that is exponential in the size of the menu (this result is presented formally in Appendix A). An alternative to this approach is estimating the revenue of other menus, more technically finding an *unbiased estimator* with *bounded range* for the revenue of all the menus, and then running the full information algorithm with the estimates, as introduced by Awerbuch and Mansour (2003). We take the latter approach and find the estimates by employing the notion of *barycentric spanners* (Awerbuch and Kleinberg, 2008). A barycentric spanner is a basis in a vector space such that any vector can be represented as a linear combination of basis vectors with bounded coefficients. By utilizing this concept, we provide algorithms with a regret bound that is sublinear in the number of timesteps and polynomial in other parameters. Similar ideas were employed in Balcan et al. (2015).

There are two main ideas deriving our bounded-regret algorithm. The first is a reduction from the partial information case to the full information case assuming Oracle access to *proper estimates* of utilities for all the menus, and the second is deriving these estimates. The first idea was introduced by Awerbuch and Mansour (2003), and we directly use an inspired theorem by Balcan et al. (2015) that suits our setting more accurately. For the second, we also use similar machinery to Balcan et al. (2015).

We first show how to estimate the utility of any menu by only using the response of the buyers to a limited number of menus. In doing so, we take advantage of the dependence between responses of the buyers for different menus to obtain estimates for unused menus. In order to estimate the expected revenue of each menu over a time interval, it is sufficient to estimate the probability of selection of each option in the menu (tariff index and number of units) by the buyers. Since the price of each option is determined by the menu, we can infer the expected revenue using these probabilities. Note that the option that each buyer type selects is fixed throughout each region. Balcan et al. (2015) use the dependence between these probabilities across regions to find a limited set of menus that infer the estimates. An analogous argument to theirs in our setting is as follows. Let $\mathcal{I}$ be the set of length-$V$ indicator vectors that, for each region $\mathcal{P}_\mu$ and each option $(j, k)$, indicate the (maximal set of) buyer types that select the option $(j, k)$ given menus in $\mathcal{P}_\mu$. The algorithm

presents the menus corresponding to the barycentric spanner of $\mathcal{I}$ to buyers at random times and records whether the buyer selects the corresponding option. We show the utility of each menu can be represented as a linear function of its corresponding vectors in $I$ and, therefore, a linear function of the barycentric spanner vectors of $\mathcal{I}$. This is enough to derive the estimates.

Now, we describe the overall structure of the algorithm. The algorithm operates in time blocks, with each block consisting of exploitation and exploration time steps. The exploration time steps are selected uniformly at random within the block and are limited in number. In an exploitation step, the menu used is the output of the full information algorithm, employing unbiased estimators from the previous time block. These menus are always the extreme points $\mathcal{E}$. During exploration time steps, the menus corresponding to the barycentric spanner are used. At the end of each time block, the algorithm refines the estimators of all corner points using the information gathered in the exploration phases. The uniform random selection of time steps ensures that under any arbitrary sequence of valuations, the values observed in exploration time steps are selected uniformly at random, and thus, the estimator is unbiased (the expected value of the utility estimator for each menu is equal to the utility of that menu). A detailed description and proof of the theorem are provided in the appendix.

**Theorem 25.** *In the partial information (bandit) case for length-$\ell$ menus of two-part tariffs, when there are $V$ different types of buyers, there is an algorithm with regret bound of $\tilde{O}(T^{2/3}\ell(HKV)^{1/3}\log^{1/3}(V\ell K))$.*

**Technical contribution.** Although the general structure of the algorithm is similar to Balcan et al. (2015), the problem settings are quite different, and whether similar ideas could work in both settings is not apparent. We are able to adapt the ideas to provide no-regret algorithms for menus of two-part tariffs. This adaptation requires establishing new properties and definitions for our problem settings.

### 3.3 Distributional Learning for Two-Part Tariffs

We present distributional learning results for menus of two-part tariffs. The learning algorithm simply considers all menus in the discretized set specified by Theorem 1 and outputs the empirical revenue-maximizing menu given the samples. More specifically, for each menu in the discretized set, the algorithm computes the cumulative revenue achieved from the samples and outputs the menu with the maximum cumulative revenue. The revenue from each sample (buyer) for a fixed menu is the total payment corresponding to the buyer's utility maximizing option (tariff index and the number of units). This approach has a major difference with the previous line of work, e.g., (Balcan et al., 2018c; 2020b; 2022b), that did not use a discretization and optimized over the infinite parameter space.

**Theorem 26.** *In the distributional setting, for length-$\ell$ menus of two-part tariffs, there exists a learning algorithm with sample complexity $\frac{H^2}{2\varepsilon^2}(2\ell\ln\left(\frac{2KH\ell}{\varepsilon}\right) + \ln\left(2/\delta\right))$, and running time $\frac{H^2}{2\varepsilon^2}\left(2\ell\ln\left(\frac{2KH\ell}{\varepsilon}\right) + \ln\left(2/\delta\right)\right)K\ell\left(\frac{2HK\ell}{\varepsilon}\right)^{2\ell}$.*

**Remark.** For menus of length larger than one, i.e., $\ell > 1$, Theorem 26 provides much simpler algorithm and its running time is roughly the square root of the running time of the previous result (Balcan et al., 2020b; 2022b) in the worst case in terms of parameters $H$, $K$, and $1/\varepsilon$. Under extra structural assumptions, (Balcan et al., 2022b) may result in better running times (see Appendix B for more details). Furthermore, in the real-world applications of menus of two-part tariffs, the length of the menu is often a small number; for example, there are a limited number of gym membership or delivery subscription options. Therefore, the exponential dependence on the length of the menu might not be a significant issue in such settings.

**Technical comparison to prior work.** For both menus of lotteries and two-part tariffs, distributional learning results were presented before (Balcan et al., 2018c; 2020b). Our discretization-based techniques lead to improvements over the previously best-known algorithms. Our algorithms choose several menus in a data-independent way (via data-independent discretization) and then select the best of them based on the data (empirical risk minimization over a cover); however, the prior algorithms optimize over the infinite space based on the sampled data (empirical risk minimization over the entire space utilizing geometric structure of utility functions). In the context of two-part tariffs, our algorithm is much simpler than prior ones for the same problem, yet it enjoys improved worst-case runtime guarantees compared to them Balcan et al. (2018c; 2020b) when the length of the menu is more than one (Theorem 26).

## 4 Menus of Lotteries

Consider selling $m$ items to a buyer. A set $M = \left\{ \left( \boldsymbol{\phi}^{(0)}, p^{(0)} \right), \left( \boldsymbol{\phi}^{(1)}, p^{(1)} \right), \ldots, \left( \boldsymbol{\phi}^{(\ell)}, p^{(\ell)} \right) \right\} \subseteq \mathbb{R}^m \times \mathbb{R}$, where $\boldsymbol{\phi}^{(0)} = \mathbf{0}$ and $p^{(0)} = 0$ is a length-$\ell$ menu of lotteries. Each $\boldsymbol{\phi}^{(j)}$ is a vector of length $m$. Under the lottery $\left( \boldsymbol{\phi}^{(j)}, p^{(j)} \right)$, a buyer receives each item $i$ with probability $\phi^{(j)}[i]$ and pays a price of $p^{(j)}$. The buyer's expected utility for the lottery $\left( \boldsymbol{\phi}^{(j)}, p^{(j)} \right)$ is their expected value for the lottery less their payment. We consider additive and unit-demand buyers. For additive buyers, their value for lottery $j$ is $\sum_{i=1}^{m} v(\boldsymbol{e}_i) \cdot \phi^{(j)}[i]$, where $v(\boldsymbol{e}_i)$ is their value for item $i$. The buyer's expected utility is $\sum_{i=1}^{m} v(\boldsymbol{e}_i) \cdot \phi^{(j)}[i] - p^{(j)}$. Note that for additive buyers, due to linearity of expectation, it does not matter whether the allocations of the items in a lottery, are independent or correlated. For unit-demand buyers, without loss of generality, we only consider lotteries such that $\sum_{i=1}^{m} \phi^{(j)}[i] \leq 1$. Under this constraint, for each lottery $j$, the allocations of the items are dependent, and the buyer never receives more than one item. In this case, the utility for lottery $j$ has the same expression as for additive buyers. Presented with a menu of lotteries, the buyer selects a utility-maximizing lottery $\left( \boldsymbol{\phi}^{(j^*)}, p^{(j^*)} \right)$ and the mechanism achieves revenue $p^{(j^*)}$.

Putting the problem formulation in the context of Section 2, $\mathcal{M}$ is the set of all menus of lotteries, each parameterized by $\boldsymbol{\rho}$ which in this case contains all $\boldsymbol{\phi}^{(j)}$ and $p^{(j)}$, where each $\phi^{(j)}[i] \in [0, 1]$ and $p^{(j)}$ is $\in [0, mH]$ for the additive setting (and $\in [0, H]$ for the unit-demand setting). $\Pi$ is the set of buyer valuations and $u : \Pi \times \mathcal{C} \to [0, mH]$ be a utility function where $u(\boldsymbol{v}, \boldsymbol{\rho})$ measures the revenue of the menu with parameters $\boldsymbol{\rho}$ on buyer valuations $\boldsymbol{v} \in \Pi$.

### 4.1 Discretization procedure

In this section, we introduce a rounding procedure for menus of lotteries. In this procedure, given any vector of parameters (representing a menu) with arbitrary coordinates, we find a transformation to another vector that has two properties; first, the revenue of the output is nearly as high as the original menu for *any* valuation; secondly, the coordinates corresponding to allocation probabilities and prices belong to a finite set of values. This rounding procedure performed on all possible menus results in a final set of outcomes. We perform the learning algorithms over this finite set.

**Theorem 27.** *Given a menu of lotteries $M$ and parameters $0 < \alpha < 1$, $0 < \delta < 1$, and $K$, an arbitrary natural number, Algorithm 5 outputs menu $M'$ such that $\mathrm{Rev}(M') \geq \mathrm{Rev}(M)(1-\delta)(1-\alpha)^K - (2K+1)\alpha - mH(1-\delta)^K$. The set of possible allocation probabilities is $\{0, (1-\alpha)^{K'}, (1-\alpha)^{K'-1}, \ldots (1-\alpha)^0 = 1\}$, where $K' = \lfloor 1/\alpha \ln(Hm/\alpha) \rfloor$ and the set of possible prices is $\{0, Hm\alpha, 2Hm\alpha, \ldots Hm\}$. This constitutes a space with at most $O\left( (1/\alpha^{\ell m + \ell})(\ln(Hm/\alpha))^{lm} \right)$ discrete points, when limiting to length-$\ell$ menus and $O\left( 2^{(1/\alpha^{m+1})(\ln(Hm/\alpha))^m} \right)$ discrete points for arbitrary-length menus.*

**Overview of Algorithm 5.** The algorithm consists of three main steps, and its logic is similar to that of Dughmi et al. (2014). In step 1, we divide the lotteries in the menu exceeding a minimum price into $K$ levels based on their price (and remove the ones below the minimum). The division in prices is proportional to powers of $(1-\delta)$ with a higher level $k$ having a higher price, compared to a lower level $k' < k$. Step 2 rounds down the allocation probability coordinates to a finite set. By multiplying $\boldsymbol{\phi}$ by $(1-\alpha)^{K-k}$ and then rounding to integer powers of $(1-\alpha)$, the allocation probabilities of lower-price levels decrease by a larger factor, making lower-price levels less desirable. Step 3 rounds down the prices, first by multiplying all prices by the same factor, $(1-\alpha)^K$, then by rounding to multiples of $\alpha$ and finally by subtracting $2k\alpha$, which results in more subtraction of price for originally higher-price entries. The main insight behind nearly preserving the revenue of the original menu (and circumventing the issue with simple rounding) is that prices of the more expensive lotteries (higher-price level) are decreased more than the lower-price ones, while their allocation decreases by a lower factor. This ensures that no buyer *with any valuation*, switches from a higher-price level to a lower-price, after the rounding.

Before providing the proof of the discretization step, we note that this procedure for menus of lotteries needs extra care and the common rounding of the parameters may result in arbitrarily lower revenue. For example, if there are two lotteries with a similar utility for the buyer but a large difference in prices, minor changes

---

**Algorithm 5:** (Almost) revenue preserving rounding for menus of lotteries

---

**Input:** Menu of lotteries $M$ with entries of pairs $(\boldsymbol{\phi}, p)$, $K \in \mathbb{N}$, and $\alpha$ such that $0 < \alpha < 1$.
**Step 1:** Partition the entries $(\boldsymbol{\phi}, p)$ of the menu $M$ into levels, where each level $k$, for $k = 1, \ldots, K$, contains all entries whose price is in the range $mH(1-\delta)^{K-k+1} < p \leq mH(1-\delta)^{K-k}$.
For every entry $(\boldsymbol{\phi}, p)$ in level $k$, put an entry $(\boldsymbol{\phi}', p')$ in $M'$ where $\boldsymbol{\phi}'$ is the outcome of step 2 and $p'$ is obtained by step 3.
**Step 2:** multiply $\boldsymbol{\phi}$ by $(1-\alpha)^{K-k}$, and round down all allocation probabilities to the set of zero and all integer powers of $(1-\alpha)$ in the range $[\frac{\alpha}{Hm}, 1]$.
**Step 3:** First, multiplying $p$ by a factor of $(1-\alpha)^K$, then rounding $p$ down to an integer multiple of $\alpha$, and then subtracting $2k\alpha$.
**Output:** $M'$: the modified menu.

---

in the probability of allocations or the prices may make the user switch from the high-price lottery to the low-price one. What follows is a concrete example of why standard rounding procedures fail.

**Example 1.** *Consider a menu of three lotteries.*

| alloc. prob. | price | utility | alloc. prob. | price | utility |
|---|---|---|---|---|---|
| 0 | 0 | 0 | 0 | 0 | 0 |
| 0.26 | 0.24 | -0.084 | 0.25 | 0.125 | 0.025 |
| 0.95 | 0.52 | 0.05 | 0.5 | 0.5 | -0.2 |

| alloc. prob. | price | utility | alloc. prob. | price | utility |
|---|---|---|---|---|---|
| 0 | 0 | 0 | 0 | 0 | 0 |
| 0.5 | 0.125 | 0.175 | 0.5 | 0.25 | 0.05 |
| 1 | 0.5 | 0.1 | 1 | 1 | -0.4 |

*Consider the buyer that has value $0.6$ for the item. The first table shows the original menu. With this menu the buyer's highest utility option is the last lottery that causes the highest revenue, i.e., $\mathrm{Rev} = 0.52$. The following tables show the new menus after rounding down the allocation probabilities and prices, rounding up allocation probabilities and rounding down prices, and rounding up allocation probabilities and prices (all to powers of $1/2$), respectively. All these transformations result in the highest utility lottery changing to the middle lottery which causes smaller revenue.*

*Proof of Theorem 27.* Most of this proof is identical to that of Dughmi et al. (2014). Note that in the algorithm, the original entries in a menu are divided into levels $k = 1, \ldots, K$ such that $k = 1$ is the lowest-price level and $k = K$ is the highest price one. First, we show that if a buyer's utility-maximizing lottery is in level $k$ given $M$, their utility-maximizing lottery in $M'$ is never in a lower-price level $k' < k$. Intuitively, the reason is that the lotteries with lower-level prices have their allocation reduced more and their prices reduced less than the ones in higher levels. More formally, let $(x, p)$ be at level $k$ and $(y, q)$ at level $k' < k$. Also, let $(x', p')$ and $(y', q')$ be the transformed lotteries in the output of the algorithm. Than, $p' - q' < ((1-\alpha)^K p - 2k\alpha) - ((1-\alpha)^K q - 2k'\alpha - \alpha) \leq (1-\alpha)^K(p - q) - \alpha$, and for every valuation $v$, $x' \cdot v - y' \cdot v > ((1-\alpha)^{K-k+1} x \cdot v - \alpha) - (1-\alpha)^{K-k'} y \cdot v \geq (1-\alpha)^K(x \cdot v - y \cdot v) - \alpha$. Now, consider an arbitrary valuation $v$ that has higher utility choosing $(x, p)$ than $y, q$. Therefore $x \cdot v - p \geq y \cdot v - q$, and therefore $p - q \leq x \cdot v - y \cdot v$. Combining this inequality with the ones above implies $x' \cdot v - p' \geq y' \cdot v - q'$.

Secondly, we compute an upper bound on the loss incurred. Suppose the original utility-maximizing lottery was $(x, p)$ in $M$. Also, suppose in $M'$, the utility-maximizing lottery is $(y', q')$ which is the transformation of $(y, q)$. The first scenario is when $p \geq mH(1-\delta)^K$. Note that in this case, $q$ may be smaller by a factor $(1-\delta)$ than $p$, then to obtain $q'$ we first lost a multiplicative factor of $(1-\alpha)^K$ and then an additive factor of at most $(2K+1)\alpha$ (including the rounding). Thus $q' \geq (1-\delta)(1-\alpha)^K p - (2K+1)\alpha$. In the second case where $p < mH(1-\delta)^K$, the loss is at most $mH(1-\delta)^K$. Therefore, in any case, $q' \geq (1-\delta)(1-\alpha)^K p - (2K+1)\alpha - mH(1-\delta)^K$.

Thirdly, the set of possible prices is $\{0, Hm\alpha, 2Hm\alpha, \ldots Hm\}$ which is of size $1/\alpha$ and the set of possible allocation probabilities is $\{0, (1-\alpha)^{K'}, (1-\alpha)^{K'-1}, \ldots (1-\alpha)^0 = 1\}$, for $K' = \lfloor 1/\alpha \ln{(Hm/\alpha)} \rfloor$ which is of size $1/\alpha \ln(Hm/\alpha)$. In the $\ell$-length menus, there are $\ell$ prices and $m\ell$ allocation probabilities in total. In the unlimited-length menus, we consider the possibility that each potential lottery (each distinct vector of parameters) belongs to the lottery or not. This analysis gives us the final size of the discrete points. □

**Technical contribution.** Our discretization scheme (Algorithm 5) extends that of Dughmi et al. (2014) in the following aspects: (i) We remove the lower bound assumption on value distribution: Dughmi et al. (2014) assume values belong to $[1, H]$, and we extend the discretization scheme to work when there is no lower bound on value distributions; i.e., values are in $[0, H]$. (ii) Supporting additive valuations: The original discretization in Dughmi et al. (2014) works for unit-demand valuations. (iii) We also modify the algorithm to support limited-length menus. As a consequence, we are able to provide improved regret bounds and running times when the size of the menu is limited. These extensions are done by small modifications to the algorithm and expand the scope of the application of the scheme.

## 4.2 Online Learning

We provide bounded-regret online learning algorithms in full and partial information settings for fixed and arbitrary-length menus of lotteries. The setting considered is as follows. In each round, a new buyer arrives, and a length-$\ell$ lottery menu is presented to the buyer. The buyer selects her utility-maximizing lottery $j$ and pays $p^{(j)}$. The mechanism achieves revenue $p^{(j)}$. Missing proofs and explicit descriptions of the algorithms are deferred to Appendix B.

In the full information setting, the seller sees the revenue generated for all the possible menus. Similar to the previous section, we run Algorithm 2 (a weighted majority algorithm) over the discretized set as the outcome of Algorithm 5 and derive the following results for the length-$\ell$ menus and arbitrary length menus.

**Theorem 28.** *In the full information case for length-$\ell$ menus of lotteries, running Algorithm 2 over the discretized set of menus specified in Theorem 27 for $\alpha = T^{-1}$, $\beta = T^{-0.5}$, $K = T^{0.5}$, and $\delta = T^{-0.5}$ has regret $\tilde{O}(m^2 H \ell \sqrt{T})$.*

**Theorem 29.** *In the full information case for arbitrary length menus of lotteries, running Algorithm 2 on menus specified in Theorem 27 for $\alpha = T^{-1/(2m+2)}$, $\beta = T^{-1/(m+1)}$, $K = T^{1/(m+1)}$, and $\delta = T^{-1/(m+1)}$ has regret $\tilde{O}(mHT^{1-1/(2m+4)} \ln^m{(mHT)})$.*

In the partial information setting, the seller only observes the revenue generated for the menu at hand. Similar to the previous section, we run Algorithm 3 (EXP3 algorithm) over the discretized set as the outcome of Algorithm 5 and derive the following result for length $\ell$ menus.

**Theorem 30.** *In the partial information case for length-$\ell$ menus of lotteries, running Algorithm 3 over discretized set of menus in Theorem 27 for $\alpha = T^{-1/(\ell m+2)}$, $\beta = \gamma = T^{-1/(4\ell m+8)}$, $K = T^{1/(2\ell m+4)}$, and $\delta = T^{-1/(2\ell m+4)}$ has regret $\tilde{O}(m^2 H \ell T^{1-1/(2\ell m+4)} \ln^{\ell m+1}{(mHT)})$.*

For the case with $V$ buyer types, we use similar machinery to Section 3.2.3 to derive bounded regret algorithms in the full and partial information settings. The discussion of how to adapt to the lotteries setting is deferred to the appendix. the partial information case.

**Theorem 31.** *In the full information case for length-$\ell$ menus of lotteries, when there are $V$ types of buyers, there is an algorithm with regret bound of $O(m^2 H \ell \sqrt{T} \ln{(V\ell)})$.*

**Theorem 32.** *In the partial information (bandit) case for length-$\ell$ menus of lotteries, when there are $V$ different types of buyers, there is an algorithm with regret bound of $O(T^{2/3}(\ell m)^{4/3}(HV)^{1/3} \log^{1/3}{(V\ell)})$.*

**Remark.** The above results hold under adversarial input. Unlike menus of two-part tariffs (and many other families of algorithms and mechanisms discussed in Balcan et al. (2018b; 2020a)), for menus of lotteries, we provide evidence that *dispersion*, a sufficient condition for online learning under smooth distributions, may not hold. A formal result is stated as Theorem 33.

### 4.2.1  Failure of Dispersion for menus of lotteries

In this section, we prove that without making extra assumptions about optimal menus of lotteries, both definitions of dispersion (Definitions 15 and 38) fail. In particular, we show that the failure of both conditions happens if the optimal menu (maximizer) has two lotteries close to each other (similar coordinates) and satisfies some other properties. Example 2 illustrates a setting where there are lotteries with arbitrarily close coordinates in the optimal menu.

**Theorem 33.** *Let the maximizer $\rho^*$ have the following properties, where $\phi_{\rho^*}^{(1)}, p_{\rho^*}^{(1)}, \phi_{\rho^*}^{(2)}, p_{\rho^*}^{(2)}$ are the coordinates of $\rho^*$, respectively illustrating the probability of allocating item one in lottery 1, the price of lottery 1, the probability of allocating item one in lottery 2, the price of lottery 2, and the allocation probability for other items are the same across these lotteries.*

1. *$p_{\rho^*}^{(1)} - p_{\rho^*}^{(2)} = (L + 1/2)\varepsilon$, where $L$ is the Lipschitz parameter.*

2. *$\phi_{\rho^*}^{(1)} - \phi_{\rho^*}^{(2)} = (L + 1)\varepsilon/c + \varepsilon/2$.*

3. *$c$ is a constant such that $c \leq H$.*

*In this case, for every $\kappa$-bounded distribution whose density is also lower-bounded by $1/\kappa$, the conditions of Definitions 15 and 38, are violated. In particular, in Definition 15, the probability of a hyperplane crossing the $\varepsilon$-radius ball centered at the maximizer is a constant depending on $c$; and in Definition 38, there exists a pair of points such that the expected number of times that their loss function difference violates the Lipschitz condition for any Lipschitz constant $L' = L/2$ is a constant depending on $c$.*

*Proof.* We first show why Definition 15 fails. Consider a ball of radius $\varepsilon$ centered at the maximizer $\rho^*$. Let this ball be $B$. We show that the probability of a hyperplane crossing $B$ is constant. Consider a point $\rho \in B$. We first find the probability density of hyperplanes going through $\rho$. Then, we integrate it to find the probability of crossing the ball. The following equation shows for what value of $v$ (the value for the item), the hyperplane goes through $\rho$.

$$v\phi_\rho^{(1)} - p_\rho^{(1)} = v\phi_\rho^{(2)} - p_\rho^{(2)}$$
$$v = \frac{p_\rho^{(1)} - p_\rho^{(2)}}{\phi_\rho^{(1)} - \phi_\rho^{(2)}}$$

Let $v_B^{\min}$ and $v_B^{\max}$ be the minimum value of $v$ for which the hyperplane crosses the ball (i.e., there is $\rho \in B$ such that $v_B^{\min} = \frac{p_\rho^{(1)} - p_\rho^{(2)}}{\phi_\rho^{(1)} - \phi_\rho^{(2)}}$), and the maximum value respectively. The probability that the hyperplane crosses the ball is $\int_{v_B^{\min}}^{v_B^{\max}} f(v)dv$, where $f(v)$ is the density function of the value for the item.

We consider the following points. These points are all in $\varepsilon$ proximity of $\rho^*$, therefore, fall in a ball of radius $\varepsilon$ centered at $\rho^*$. Consider points with $p^{(2)} = p_{\rho^*}^{(2)}$ and $\phi^{(2)} = \phi_{\rho^*}^{(2)}$. Let $p^{(1)}$ be in $[p_{\rho^*}^{(2)} + L\varepsilon, p_{\rho^*}^{(2)} + (L+1)\varepsilon]$. Let $\phi^{(1)}$ be in $[\phi_{\rho^*}^{(2)} + (L+1)\varepsilon/c, \phi_{\rho^*}^{(2)} + (L+1)\varepsilon/c + \varepsilon]$.

With the above construction, the numerator ranges from $L\varepsilon$ to $(L+1)\varepsilon$, and the denominator ranges from $(L+1)\varepsilon/c$ to $(L+1)\varepsilon/c + \varepsilon$. Therefore, $v_B^{\min} = \frac{Lc}{L+c+1}$ and $v_B^{\max} = c$. For $\kappa$-bounded distribution with support $[0, 1]$, $\int_{v_B^{\min}}^{v_B^{\max}} f(v)dv$ is at least

$$\frac{c - \frac{Lc}{L+c+1}}{\kappa} = \frac{\frac{c(c+1)}{L+c+1}}{\kappa};$$

which is constant for a constant $c$.

Now, we show that Definition 38 fails. To do so, we still consider pair of points $\rho$ and $\rho'$ which correspond to $v_B^{\min}$ and $v_B^{\max}$, respectively. If we consider the line segment connecting $\rho$ and $\rho'$, the probability of the hyperplane crossing these two points is still $\int_{v_B^{\min}}^{v_B^{\min}} f(v)dv$ which again for $\kappa$-bounded distribution with support $[0,1]$ whose density is also lower-bounded by $1/\kappa$, $\int_{v_B^{\min}}^{v_B^{\max}} f(v)dv$ is at least

$$\frac{c - \frac{Lc}{L+c+1}}{\kappa} = \frac{\frac{c(c+1)}{L+c+1}}{\kappa};$$

which is constant for a constant $c$. Note that $|p_\rho^{(1)} - p_{\rho'}^{(2)}| \geq L\varepsilon$ and $|p_\rho^{(2)} - p_{\rho'}^{(1)}| \geq L\varepsilon$ which implies anytime the hyperplane crosses between $\rho$ and $\rho'$, the difference in the loss, $|\ell_t(\rho) - \ell_t(\rho')|$ is at least $L\varepsilon$. Also, the Euclidean distance between $\rho$ and $\rho'$ is less than $2\varepsilon$. Therefore, the Lipschitz condition for constant $L' = L/2$ is violated a constant fraction of times in expectation. □

The following example shows that in the optimal menu of lotteries, lottery pairs can be arbitrarily close to each other.

**Example 2** ((Daskalakis et al., 2014)). *Consider the case of two items, when the buyer's value for each item is drawn i.i.d. from the distribution supported on $[0,1]$ with density function $f(x) = 2(1-x)$. Daskalakis et al. prove for this example that the unique (up to differences of measure zero) optimal mechanism has uncountable menu complexity. That is, the number of distinct options available for the buyer to purchase is uncountable. They show that the optimal mechanism contains the following four kinds of options: (a) the buyer can receive item one with probability $1$, and item two with probability $\frac{2}{(4-5x)^2}$ paying the price $\frac{2-3x}{4-5x} + \frac{2x}{(4-5x)^2}$, for any $x \in [0, \approx .0618)$, (b) the buyer can receive item two with probability $1$, and item one with probability $\frac{2}{(4-5x)^2}$ paying the price $\frac{2-3x}{4-5x} + \frac{2x}{(4-5x)^2}$, for any $x \in [0, \approx .0618)$, (c) the buyer can receive both items and pay $\approx .5535$, and (d) the buyer can receive neither item and pay nothing.*

**Technical contribution compared to prior work.** Dispersion property has been shown to hold for various algorithm and mechanism design problems (Balcan et al., 2018b; 2020a; Balcan and Sharma, 2021; Balcan et al., 2022a). This section illustrates the first evidence for the failure of the dispersion property.

### 4.3 Distributional Learning

In the distributional setting, we have sample access to buyers' valuations. The value of the buyer for item $i$ is drawn from distribution $D_i$ with support $[0, H]^m$; we do not assume independence among items. Similar to the distributional learning algorithm for menus of two-part tariffs, the algorithm simply considers all menus in the discretized set specified by Theorem 27 and outputs the empirical revenue-maximizing menu given the samples. The revenue from each sample (buyer) for a fixed menu is the payment corresponding to the buyer's utility-maximizing lottery in the menu.

**Theorem 34.** *For length-$\ell$ menus of lotteries, there is a discretization-based distributional learning algorithm with sample complexity $\tilde{O}\left(m^2 H^2/\varepsilon^2(\ell m + \ln(2/\delta))\right)$, and running time $\tilde{O}\left(\left(2m^2 H^2/\varepsilon^2\right)^{\ell m + \ell + 1} \ell(\ell m + \ln(2/\delta)) \ln^{\ell m}(mH/\varepsilon \ln(mH/\varepsilon))\right)$.*

**Remark.** For the limited menu length, the sample complexity of Theorem 34 is roughly the same as Balcan et al. (2018c), but the advantage is that we provide an efficient algorithm when $m$ and $\ell$ are constant. The analysis for arbitrary-length menus is provided in the appendix as Theorem 57. The sample complexity and running time provided are similar to that of Dughmi et al. (2014), however, Theorem 57 works for a more general setting. Dughmi et al. (2014) provide a lower bound on the sample complexity, verifying an exponential dependence on the number of items.

**Technical contribution compared to prior work.** Similar to menus of two-part tariffs, our algorithms choose several menus in a data-independent way (via data-independent discretization) and then select the best of them based on the data (empirical risk minimization over a cover); however, the prior algorithms (Balcan et al., 2018c; 2020b) optimize over the infinite space based on the sampled data (empirical risk minimization over the entire space utilizing geometric structure of utility functions). In the context of lotteries,

compared to the previous distributional learning results for fixed-length menus (Balcan et al., 2018c), our algorithm requires similar sample complexity; however, it has an efficient implementation. For arbitrary-length menus, our algorithm provides similar sample complexity and running time compared to (Dughmi et al., 2014); however, it works for a slightly more general setting.

## 5 Discussion

This paper contributes to both learning theory and mechanism design by studying prominent families of mechanisms from a learning perspective. Our work is focused on learning menu mechanisms that go beyond selling the items separately. Menus of lotteries provide a list of randomized allocations and their corresponding prices to the buyers and are specifically advantageous for selling multiple items. Menus of two-part tariffs, on the other hand, are employed for selling multiple units (copies) of an item by presenting a list of up-front fees and per-unit fees to the buyer. The two families of mechanisms are pricing schemes commonly studied for revenue maximization in the sale of goods. Both are presented as lists (menus) of options from which potential buyers can choose. From a structural perspective, both problems involve a utility function (in this case, revenue) that is piecewise linear in the parameter space. Our findings suggest that similar techniques can be applied to both problems.

We provide a suite of results with regard to these two families of mechanisms. By leveraging the structure of menus of two-part tariffs and lotteries, we provide a revenue-preserving reduction to a finite number of menus (discretization). Using this approach, we provide the first online learning algorithms for menus of lotteries and two-part tariffs with strong regret-bound guarantees and propose algorithms with significantly improved running times over prior work for the distributional settings. When there is a limited number of buyer types, we provide a reduction to online linear optimization, which enables us to obtain no-regret guarantees by presenting buyers with menus that correspond to a barycentric spanner. Finally, for the first time, we provide evidence of the failure of the "dispersion" property (Balcan et al., 2018b; 2020a)—a sufficient condition to provide a no-regret algorithm under smooth distributional assumption, which is widely applied to parametric algorithm and mechanism design problems—for a specific problem (menus of lotteries).

**Discretization versus Dispersion.** The majority of the paper focuses on online learning of these families of mechanisms. Two of the commonly used techniques for this setting are (the more traditional) discretization-based and (the recently developed) dispersion-based techniques. Menus of lotteries and two-part tariffs are examples of parametric algorithm or mechanism design, where the objective function, here revenue, has sharp discontinuities in the parameter space, and the standard procedures, such as rounding down the parameters to multiples of $\varepsilon$, may result in arbitrary revenue loss. A discretization scheme means that there exists a grid in the parameter space such that for any arbitrary parameter vector, there is a corresponding parameter vector in proximity over the grid generating similar revenue. However, finding the corresponding parameter vector (the direction to move from the original parameter vector in the space) needs taking extra care, and moving in an arbitrary direction may cause a large revenue loss. In contrast to the discretization scheme, another method developed for proving online learnability of parameterized algorithms, called *dispersion* (Balcan et al., 2018b; 2020a), asserts that under smoothness assumptions moving in a small ball of parameter vectors, does not face sharp discontinuities with high probability. This means that with high probability, moving in any direction preserves similar revenue. Nevertheless, we show evidence that the dispersion may not hold for menus of lotteries (Theorem 33) and while dispersion holds for menus of two-part tariffs (Propositions 16 and 39), it heavily uses the smoothness assumption. In conclusion, although a small but arbitrary modification may change the revenue drastically when starting from a parameter vector, in designing our discretization scheme, we show a specific direction such that a small modification along that direction preserves the revenue. See Theorems 1 and 27.

**Lower bound for regret terms.** In the full information case, the dependence of the regret bounds on $T$ is tight according to Nisan et al. (2007), Theorem 4.8. In the bandit setting, our dependence on $T$ matches a lower bound provided by Kleinberg et al. (2008) for general globally Lipschitz functions even though the utility functions in our case are only piecewise Lipschitz (not globally Lipschitz). The construction in Kleinberg et al. (2008) does not immediately imply a lower bound for our case since, since on one hand, learning piecewise Lipschitz functions is harder than globally Lipschitz ones, and on the other hand, in our

case, the utility functions have more structure beyond Lipschitzness in each piece. It is a nontrivial open question if the dependence is tight for our case. Finding a lower bound for the dependence on the other parameters is an interesting open problem. Similar dependence appears both in our discretization-based and dispersion-based algorithms. The question of whether such dependencies can be avoided motivated us to study more structured settings, such as the limited buyer type setting. In the limited buyers' type case, where we utilize the knowledge of the potential buyer types and interdependence of utilities across experts, the dependence is improved.

**Computational Efficiency.** Our discretization-based learning approach has the strength of not relying on any extra assumptions about the data and results in no-regret learning algorithms in the online learning setting without any extra assumptions. However, the drawback of this approach is that the algorithms may not be computationally efficient. Concerning known efficient algorithms, for the problem of menus of two-part tariff even in the simpler problem of distributional learning, prior results were not computationally efficient either (Balcan et al., 2020b), and our discretization-based algorithms improve upon those and satisfy the best computational guarantees in the worst case. Exploring computational complexity and the existence of more efficient algorithms is an interesting open direction. We have also taken steps to explore the possibility of more efficient algorithms by adding extra structure, e.g., smooth distributional assumption or limited buyer type assumption, to our settings. Establishing the "dispersion" property under smooth distributional assumptions enables us to use more refined online learning algorithms (a continuous version of multiplicative weight update algorithm that uses the geometric structure of utility functions), but this property has not been studied for menus of lotteries or two-part tariffs. One of our contributions is establishing this property for menus of two-part tariffs and obtaining more efficient algorithms. However, surprisingly we show this property does not work for lotteries. In the limited buyer type settings, we utilize the knowledge of the potential buyer types and interdependence of utilities across menus and provide algorithms with improved running time.

**Open Directions.** An open question is whether there is a generalization capturing the techniques applied to both menus of lotteries and two-part tariffs. It is unclear whether such a generalization capturing both problems exists. The key difficulty we face in generalizing the techniques we used for menus of two-part tariffs and lotteries stems from the difference in the structure of the utility functions, specifically, the shape of discontinuity hyperplanes. As shown in Theorem 33, we provide evidence of the failure of the dispersion technique for lotteries; however, this technique works for two-part tariffs (Propositions 16 and 39). Furthermore, the two mechanisms needed different discretization methods.

## 6 Acknowledgement

The authors would like to thank Avrim Blum, Misha Khodak, Rattana Pukdee, Dravyansh Sharma, and anonymous reviewers for helpful feedback and comments. This material is based on work supported in part by the National Science Foundation under grant CCF-1910321 and a Simons Investigator Award.

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

# A  Missing Proofs of Section 3

## A.1  Online Learning

### A.1.1  Online Learning Under Adversarial Inputs

**Full Information**

**Proposition 35** ((Auer et al., 1995), Theorem 3.2). *For any sequence of valuations $\bar{v}$,*

$$\mathrm{Rev}_{\mathrm{WM}}\left(\bar{v}\right) \geq \mathrm{OPT}_X\left(\bar{v}\right) - \frac{\beta}{2}\mathrm{OPT}_X\left(\bar{v}\right) - \frac{H\ln n}{\beta},$$

*where $X = m_1, \ldots, m_n$ are the set of experts (two-part tariff menus), $\mathrm{Rev}_{\mathrm{WM}}(\bar{v})$ is the expected revenue outcome of Algorithm 2, and $\mathrm{OPT}_X\left(\bar{v}\right)$ is the revenue of the optimal menu in $X$.*

**Theorem 7.** *In the full information case for length-$\ell$ menus of two-part tariffs, running Algorithm 2 over discretized set of menus specified in Theorem 1 for $\alpha = \beta = 1/\sqrt{T}$ has regret bounded by $\tilde{O}\left(\ell(K + H\ln H)\sqrt{T}\right)$, and running time $O(T\ell K \min\{H^{2\ell}T^\ell, 2^{H^2 T}\})$.*

*Proof.* Let $n$ be the number of menus resulting from the discretization procedure in Section 3.1. Let $\boldsymbol{v}_i$ be the valuation of the buyer at step $i$, and $\bar{v}$ be the vector of valuation of all buyers in rounds 1 through $T$. We denote $\mathrm{Rev}_{M'}()$ as the maximum revenue obtained in the set of menus resulting from the discretization procedure, $\mathrm{OPT}()$ as the optimal revenue, and $\mathrm{Rev}_{\mathrm{WM}}()$ as the revenue obtained from the weighted majority algorithm discussed above on the set of outcome menus of the discretization procedure. Then,

$$n = (H/\alpha)^{2\ell},$$

$$\mathrm{Rev}_{\mathrm{WM}}\left(\bar{v}\right) \geq \mathrm{Rev}(M')\left(\bar{v}\right) - \frac{\beta}{2}\mathrm{Rev}(M')\left(\bar{v}\right) - \frac{H\ln n}{\beta},$$

$$\mathrm{Rev}_{M'}\left(\bar{v}\right) = \sum_{i=1}^{T}\mathrm{Rev}_{M'}\left(\boldsymbol{v}_i\right),$$

$$\mathrm{Rev}_{M'}\left(\boldsymbol{v}_i\right) \geq \mathrm{OPT}\left(\boldsymbol{v}_i\right) - 2K\ell\alpha;$$

where the first expression is a result of the discretization procedure, the second expression uses Proposition 35, the third expands the revenue over $T$ terms, and the last uses Theorem 1. Rearranging the terms, we have:

$$\mathrm{Rev}_{M'}\left(\boldsymbol{v}_i\right) \geq \mathrm{OPT}\left(\boldsymbol{v}_i\right) - 2K\ell\alpha$$
$$\mathrm{Rev}_{M'}\left(\bar{v}\right) \geq \mathrm{OPT}\left(\bar{v}\right) - 2K\ell\alpha T$$
$$\mathrm{Rev}_{\mathrm{WM}}\left(\bar{v}\right) \geq \mathrm{OPT}\left(\bar{v}\right) - 2K\ell\alpha T - \frac{\beta HT}{2} - \frac{H\ln n}{\beta}$$
$$\mathrm{Rev}_{\mathrm{WM}}\left(\bar{v}\right) \geq \mathrm{OPT}\left(\bar{v}\right) - 2K\ell\alpha T - \frac{\beta HT}{2} - \frac{2H\ell\left(\ln\left(H/\alpha\right)\right)}{\beta}$$

We set variables $\alpha$ and $\beta$ to minimize the exponent of $T$ in the regret. By substituting $n$, the regret is upper bounded by

$$2K\ell\alpha T + \frac{\beta HT}{2} + \frac{2H\ell\left(\ln H - \ln\alpha\right)}{\beta}.$$

By setting $\alpha = \beta = \frac{1}{\sqrt{T}}$, The regret will be $\tilde{O}\left(\ell(K + H\ln H)\sqrt{T}\right)$. Based on the parameters chosen, the number of menus is $O(\min\{H^{2\ell}T^\ell, 2^{H^2 T}\})$. The algorithm needs to maintain the weights for these menus and update them based on the revenue at each time step. The revenue of each menu can be calculated in $O(K\ell)$ given the buyer's valuation, resulting in the stated running time. The running time in each round is the number of menus times the time to calculate the revenue for each menu. $\qquad\square$

**Partial Information**

**Proposition 36** ((Auer et al., 1995), Theorem 4.1). *For any sequence of valuations $\bar{v}$,*

$$\mathrm{Rev}_{\mathrm{Exp3}}(\bar{v}) \geq \mathrm{OPT}_X - \left(\gamma + \frac{\beta}{2}\right)\mathrm{OPT}_X - \frac{Hn\ln n}{\beta\gamma},$$

*where $X = m_1, \ldots, m_n$ are the set of experts (two-part tariff menus), $\mathrm{Rev}_{\mathrm{Exp3}}(\bar{v})$ is the expected revenue outcome of Algorithm 3, and $\mathrm{OPT}_X(\bar{v})$ is the revenue of the optimal menu in $X$.*

**Theorem 8.** *In the partial information case for length-$\ell$ menus of two-part tariffs, running Algorithm 3 over discretized set of menus in Theorem 1 for $\alpha = T^{-1/(2(1+\ell))}$, $\beta = \gamma = T^{-1/(4(1+\ell))}$ has regret bound $\tilde{O}\left(T^{1-\frac{1}{2(1+\ell)}}\ell(K + H^{2\ell+1})\right)$, and running time $O(T\min\{\min\{H^{2\ell}T^\ell, 2^{H^2T}\}, 2^{H^2T}\})$.*

*Proof.* The proof follows the same logic as that of Theorem 7. We denote $\mathrm{Rev}_{\mathrm{Exp3}}()$ as the revenue obtained from the Exp3 algorithm described above on the set of outcome menus of the discretization procedure. Similar to the proof of Theorem 7, in what follows $n$ denotes the number of menus resulting from the discretization procedure in Section 3.1. $\boldsymbol{v}_i$ is the valuation of the buyer at step $i$, and $\bar{v}$ is the sequence of valuation of all buyers in rounds 1 through $T$. $\mathrm{Rev}_{M'}()$ is the maximum revenue obtained in the set of menus resulting from the discretization procedure and $\mathrm{OPT}()$ is the optimal revenue.

$$n = (H/\alpha)^{2\ell},$$

$$\mathrm{Rev}_{\mathrm{Exp3}}(\bar{v}) \geq \mathrm{Rev}(M')(\bar{v}) - \left(\gamma + \frac{\beta}{2}\right)\mathrm{Rev}(M')(\bar{v}) - \frac{Hn\ln n}{\beta\gamma},$$

$$\mathrm{Rev}_{M'}(\bar{v}) = \sum_{i=1}^T \mathrm{Rev}_{M'}(\boldsymbol{v}_i),$$

$$\mathrm{Rev}_{M'}(\boldsymbol{v}_i) \geq \mathrm{OPT}(\boldsymbol{v}_i) - 2K\ell\alpha;$$

where the first expression is a result of Theorem 1, the second expression uses Proposition 36, the third expands the revenue over T terms, and the last uses Theorem 1. Rearranging the terms gives:

$$\mathrm{Rev}_{M'}(\bar{v}) \geq \mathrm{OPT}(\bar{v}) - 2K\ell\alpha T$$

$$\mathrm{Rev}_{\mathrm{Exp3}}(\bar{v}) \geq \mathrm{OPT}(\bar{v}) - 2K\ell\alpha T - \left(\gamma + \frac{\beta}{2}\right)HT - \frac{Hn\ln n}{\beta\gamma}$$

$$\mathrm{Rev}_{\mathrm{Exp3}}(\bar{v}) \geq \mathrm{OPT}(\bar{v}) - 2K\ell\alpha T - \left(\gamma + \frac{\beta}{2}\right)HT - \frac{2H(H/\alpha)^{2\ell}\ell(\ln H - \ln\alpha)}{\beta\gamma}$$

We set variables $\alpha$ and $\beta$ as a function of $T$ to minimize the exponent of $T$ in the regret. By setting $\alpha = T^{-1/(2(1+\ell))}$, $\beta = \gamma = T^{-1/(4(1+\ell))}$, the regret is $O\left(T^{1-\frac{1}{2(1+\ell)}}\ln(T)\ell(K + H^{2\ell+1}\ln H)\right)$. The algorithm involves maintaining weights for all the menus in the discretized set at each time step, therefore the running time at each time step is proportional to the number of the menus that are derived based on parameter $\alpha$. $\qquad\square$

### A.1.2 Online Learning Under Smooth Distributions

**Full Information**

For completeness we include previously established algorithms for the full information setting, under dispersion condition, adapted to our setting.

**Overview of Algorithms 4 and 6, related to Theorem 10.** Algorithm 4 (Balcan et al., 2018b) is an efficient algorithm for online learning in the full-information setting under smoothed distributional assumptions that uses Algorithm 6 (Balcan et al., 2018b) as a subroutine. The algorithm considers the cumulative revenue function up until the time $t-1$ over the parameter space, $\sum_0^{t-1} u_s$, and samples the

---

**Algorithm 6:** Multi-dimensional sampling algorithm ((Balcan et al., 2018b), Algorithm 2)

---

**Input:** Function $g$, partition with regions $\mathcal{P}_1, \ldots, \mathcal{P}_n$, approximation parameter $\eta$, confidence parameter $\zeta$.
  1: Define $\alpha = \beta = \eta/3$.
  2: Let $h(\boldsymbol{\rho}) = \exp(g(\boldsymbol{\rho}))$ and $h_i(\boldsymbol{\rho}) = \mathbb{I}\{\boldsymbol{\rho} \in \mathcal{P}_i\}h(\boldsymbol{\rho})$ be $h$ restricted to $\mathcal{P}_i$.
  3: For each $i \in [n]$, let $\hat{Z}_i = \mathcal{A}_{\text{integrate}}(h_i, \alpha, \zeta/(2n))$.
  4: Choose random partition index $I = i$ with probability $\hat{Z}_i / \sum_j \hat{Z}_j$.
  5: Let $\hat{\boldsymbol{\rho}}$ be the sample output by $\mathcal{A}_{\text{sample}}(h_I, \beta, \zeta/2)$.
**Output:** $\boldsymbol{\rho}$

---

menu to be presented at time $t$ approximately proportional to an exponential function of its cumulative revenue, i.e., $e^{g(\boldsymbol{\rho}_t)}$, where $g = \lambda \sum_0^{t-1} u_s$. In order to have an efficient implementation for sampling menu $\boldsymbol{\rho}_t$ approximately from distribution $\mu$ with density $f_\mu(\boldsymbol{\rho}) \propto e^{g(\boldsymbol{\rho}_t)}$, techniques from high-dimensional geometry are used in Algorithm 6. This algorithm is used when $g$ is piecewise concave (in our case, linear), and each piece is a convex set (in our case, convex polytopes where each buyer already in the sequence selects a fixed tariff index and the number of units) as shown in Lemma 13. Let $\mathcal{P}_1, \ldots, \mathcal{P}_n$ be the partition of $\mathcal{C}$ until time $t$. The algorithm first picks $\mathcal{P}_i$ with probability proportional to the integral of $f_\mu$ on that region and then outputs a sample from the conditional distribution of menus in $\mathcal{P}_i$. The algorithm assumes access to two procedures for approximate integration and sampling, namely $\mathcal{A}_{\text{integrate}}(h, \alpha, \zeta)$ and $\mathcal{A}_{\text{sample}}(h, \beta, \zeta)$. $\mathcal{A}_{\text{integrate}}(h_i, \alpha, \zeta)$ is a polynomial running-time procedure that takes the approximate integral of any logconcave function $h_i$ restricted to region $\mathcal{P}_i$ with accuracy parameter $\alpha$ and failure probability $\zeta$. $\mathcal{A}_{\text{sample}}(h_i, \beta, \zeta)$ is a polynomial procedure that approximately samples a menu with probability distribution according to $h_i$ in the region $\mathcal{P}_i$ with accuracy parameter $\beta$ and failure probability $\zeta$.

**Definition 37** ($\mathcal{A}_{\text{integrate}}(h, \alpha, \zeta)$ and $\mathcal{A}_{\text{sample}}(h, \beta, \zeta)$ (Balcan et al., 2018b)). *For any logconcave function $h : \mathbb{R}^d \to \mathbb{R}$, any accuracy parameter $\alpha > 0$, and any failure probability $\zeta > 0$, $\mathcal{A}_{\text{integrate}}(h, \alpha, \zeta)$ outputs a number $Z$ that with probability at least $1 - \zeta$ satisfies $e^{-\alpha} \int h \leq Z \leq e^\alpha \int h$. For any logconcave function $h : \mathbb{R}^d \to \mathbb{R}$, any accuracy parameter $\beta > 0$, and any failure probability $\zeta > 0$, $\mathcal{A}_{\text{sample}}(h, \beta, \zeta)$ outputs a sample $X$ drawn from a distribution $\hat{u}_h$ that with probability at least $1 - \zeta$, $D_\infty(\mu, \hat{\mu}) \leq \beta$, where $D_\infty(\mu, \hat{\mu})$ is the relative (multiplicative) distance between probability measures $\mu$ and $\hat{\mu}$. Formally, $D_\infty(\mu, \hat{\mu}) = \sup_{\boldsymbol{\rho}} |\log \frac{d\mu}{d\hat{\mu}}|$, where $\frac{d\mu}{d\hat{\mu}}$ denotes the Radon-Nikodym derivative.*

Similar to Balcan et al. (2018b), we use the implementation of $\mathcal{A}_{\text{integrate}}$ by Lovász and Vempala (2006) and $\mathcal{A}_{\text{sample}}$ by Bassily et al. (2014), Algorithm 6. These implementations satisfy the conditions in Definition 37. The first runs in time $\text{poly}(d, \frac{1}{\alpha}, \log \frac{1}{\zeta}, \log \frac{R}{r})$, where the domain of function $h$ is a subset of a ball of radius $R$ and its level set of probability mass $1/8$ is a superset of a ball with radius $r$. The second succeeds with probability 1 and runs in time $\text{poly}(d, L, \frac{1}{\beta}, \log \frac{R}{r})$.

**Theorem 10.** *Let $u_1, \ldots, u_T : \mathcal{C} \to [0, H]$ be the revenue functions of two-part tariff menus such that $u_t(\boldsymbol{\rho})$ denotes the revenue of a mechanism associated with menu parameters $\boldsymbol{\rho}$ for the buyer arriving at time $t$. Let the samples of buyers' values be drawn from $\mathcal{S} \sim \mathcal{D}^{(1)} \times \cdots \times \mathcal{D}^{(T)}$. Suppose $v(k) \in [0, H]$ for any number of units $k \in [K]$. Also, suppose that for each distribution $\mathcal{D}^{(t)}$, and every pair of number of units $k$ and $k'$, $v(k)$ and $v(k')$ have a $\kappa$-bounded joint distribution. An efficient implementation of the exponentially weighted forecaster with $\lambda = \sqrt{2\ell \ln(2H^2\kappa\sqrt{T})/T}/H$ (Algorithm 4) has expected regret bounded by $\tilde{O}((H\ell^2 K^2\sqrt{\log \kappa} + 1/(H\kappa))\sqrt{T})$ and runs in time $\tilde{O}((T+1)^{poly(\ell,K)}poly(\ell, \sqrt{T}) + KT\sqrt{T})$.*

*Proof.* Proposition 16 determines the dispersion for two-part tariff menus with probability $1 - \zeta$. Theorem 1 in Balcan et al. (2018b) relates dispersion to a regret bound for full information online learning algorithms. It states if a sequence of piecewise $L$-Lipschitz functions in $d$ dimensions is $(w, k)$-dispersed, there is an exponentially weighted forecaster with expected regret $O(H(\sqrt{Td \log R/w} + k) + TLw)$. Since dispersion holds with probability $1 - \zeta$, the final regret bound is $O((1 - \zeta)(H(\sqrt{Td \log R/w} + k) + TLw)) + \zeta H$.

Substituting $w$ and $k$ by dispersion found in Proposition 16 gives:

$$O\left(H\left(\sqrt{2T\ell\log(2H^2\kappa T^{1-\alpha})}+\ell^2 K^2 T^\alpha\sqrt{\ln\frac{\ell K}{\zeta}}\right)+\frac{T^\alpha}{2H\kappa}+\zeta HT\right).$$

For all rounds, $t \in [T]$, the sum of utilities is linear over at most $(T+1)^{\ell^2 K^2}$ pieces, and all the pieces are convex. In this case, we may use Algorithm 6 as a subroutine to Algorithm 4 for a more efficient but approximate implementation. Setting dispersion parameters $\zeta = 1/\sqrt{T}$ and $\alpha = 0.5$ and approximation parameters $\eta = \zeta = 1/\sqrt{T}$ and using Theorem 1 in Balcan et al. (2018b), gives the statement's regret bound and running time. $\qquad\square$

**Bandit Setting**

The bandit-setting algorithm considers a grid over the parameter space, whose granularity depends on the dispersion parameters, and runs the Exp3 algorithm over menus corresponding to the grid.

**Theorem 11.** *Let $u_1, \ldots, u_T : \mathcal{C} \to [0, H]$ be the revenue functions of two-part tariff menus such that $u_t(\boldsymbol{\rho})$ denotes the revenue of a mechanism associated with menu parameters $\boldsymbol{\rho}$ for the buyer arriving at time $t$. Let the samples of buyers' values be drawn from $\mathcal{S} \sim \mathcal{D}^{(1)} \times \cdots \times \mathcal{D}^{(T)}$. Suppose $v(k) \in [0, H]$ for any number of units $k \in [K]$. Also, suppose that for each distribution $\mathcal{D}^{(t)}$, and every pair of number of units $k$ and $k'$, $v(k)$ and $v(k')$ have a $\kappa$-bounded joint distribution. There is a bandit-feedback online optimization algorithm with expected regret $\tilde{O}\left(T^{(2\ell+1)/(2\ell+2)}\left(H^2 K\sqrt{\ell}\kappa^{d/2}\sqrt{\log\kappa}\right) + 1/H\kappa + H\ell^2 K^2\right)$. The per-round running time is $O(H^{4\ell}\kappa^{2\ell}T^\ell)$.*

*Proof.* Proposition 39 determines dispersion for two-part tariff menus with probability $1 - \zeta$. Theorem 3 in Balcan et al. (2018b) relates dispersion to a regret bound for the bandit setting. It states if a sequence of piecewise $L$-Lipschitz functions that are $(w, k)$-dispersed and when the parameter space is contained in a ball of radius $R$, running Exp3 algorithm has regret

$$O\left(H\sqrt{Td\left(\frac{3R}{w}\right)^d\log\frac{R}{w}}+TLw+Hk\right).$$

The per-round running time is $O((3R/w)^d)$. Note that dispersion holds only with probability $1 - \zeta$ and with probability $\zeta$, regret is bounded by $HT$. In our case, $L = K + 1$, $R = H$ and $d = 2\ell$. Substituting these terms along with $w$ and $k$, and setting $\alpha = 2\ell+1/2\ell+2$ and $\zeta = 1/\sqrt{T}$ gives the regret bound and running time in the theorem statement. $\qquad\square$

**Semi-Bandit Setting** For the semi-bandit setting, we need to invoke a more recent definition of dispersion.

**Definition 38** ((Balcan et al., 2020a), $\beta$-point-dispersion)**.** *The sequence of loss functions $l_1, l_2, \ldots$ is $\beta$-point-dispersed for the Lipschitz constant $L$ if for all $T$ and for all $\varepsilon \geq T^{-\beta}$, we have that, in expectation, the maximum number of functions among $l_1, \ldots, l_T$ that fail the $L$-Lipschitz condition for any pair of points at distance $\varepsilon$ in $\mathcal{C}$ is at most $\tilde{O}(\varepsilon T)$. That is, for all $T$ and for all $\varepsilon \geq T^{-\beta}$, we have $\mathbb{E}\left[\max_{\rho,\rho'}\left|\{t \in [T] : |l_t(\rho) - l_t(\rho')| > L\|\rho - \rho'\|_2\}\right|\right] = \tilde{O}(\varepsilon T)$. where the max is taken over all $\rho, \rho' \in \mathcal{C} : \|\rho - \rho'\|_2 \leq \varepsilon$.*

**Proposition 39.** *Suppose $l_t(\boldsymbol{\rho}) = H - u_t(\boldsymbol{\rho})$, where $u_t(\boldsymbol{\rho})$ is the revenue of the two-part tariff menu mechanism with prices $\boldsymbol{\rho}$ and buyer's values $\boldsymbol{v}_t$ at time $t$, where buyers' values are drawn from $\mathcal{D}^{(1)} \times \cdots \times \mathcal{D}^{(T)}$. If $\mathcal{D}^{(i)}$ are $\kappa$-bounded, where $\kappa = \tilde{o}(T)$, and $K$ and $\ell$, the maximum number of units and the number of tariffs, are polynomial in $T$, these loss functions are $\beta$-point-dispersed for $\beta = 1/2$.*

*Proof.* We use the following statement from Balcan and Sharma (2021), theorem 7.

**Proposition 40.** *(Balcan and Sharma, 2021) Let $l_1, \ldots, l_T : \mathbb{R}^d \to \mathbb{R}$ be independent piecewise $L$-Lipschitz functions, each having discontinuities specified by a collection of at most $K'$ algebraic hypersurfaces of bounded degree. Let $P$ denote the set of axis-aligned paths between pairs of points in $\mathbb{R}^d$, and for each $s \in P$ define $D(T, s) = |\{1 \leq t \leq T \mid l_t \text{ has a discontinuity along } s\}|$. Then we have $\mathbb{E}[\sup_{s \in P} D(T, s)] \leq \sup_{s \in P} \mathbb{E}[D(T, s)] + O(\sqrt{T\log(TK')})$.*

The number of hyperplanes, defined as $K'$ in the theorem, is at most $T\ell^2 K^2$ and $l_t$s are piecewise $(K+1)$-Lipschitz function (by Lemma 52); where $T$ is the number of buyers (rounds), $\ell$ is the number of tariffs, and $K$ is the maximum number of units. Note that, as shown in Lemma 13. The independence of $l_t$s comes from the assumptions of this setting, where the buyer valuations for each round are drawn independently.

Definition 38 counts the number of times (in $T$ time intervals) that the difference in utility of the pair violates the $L$-Lipschitz condition, and finds the worst pair for this property. Proposition 40, counts the number of times that in an axis-aligned path, the utility function has discontinuities. Therefore, $\sup_{s \in P} \mathbb{E}[D(T, s)] + O(\sqrt{T \log(TK')})$ is an upper bound on $\mathbb{E}\big[\max_{\rho,\rho'} \big|\{t \in [T] : |u_t(\rho) - u_t(\rho')| > L\|\rho - \rho'\|_2\}\big|\big]$. To find the dispersion we need to find $\sup_{s \in P} \mathbb{E}[D(T, s)]$.

Recall from the proof of Proposition 16 that the discontinuities can be partitioned into $\ell^2 K^2$ multisets of parallel hyperplanes, such that multiset $\mathcal{B}_{j,k,j',k'}$ corresponds to pairs of tariffs and the number of units $(j, k)$ and $(j', k')$. In addition, since we assume the buyers' valuations are in the range $[0, H]$ and are drawn from pairwise $\kappa$-bounded joint distributions, the offsets of the hyperplanes are independent draws from a $H\kappa$-bounded distribution. The number of multi-sets is $\ell^2 K^2$, and the size of each multi-set is $T$. The hyperplanes within each multi-set are well-dispersed. For a multi-set $\mathcal{B}_{j,k,j',k'}$, let $\Theta_{j,k,j',k'}$ be the multi-set of the hyperplanes' offsets. By assumption, the elements of $\Theta_{j,k,j',k'}$ are independently drawn from $H\kappa$-bounded distributions. Since the offsets are $H\kappa$-bounded, the probability that it falls in any interval of length $\varepsilon$ is $O(H\kappa\varepsilon)$. The expected number of hyperplanes crossed from each multiset in distance $\varepsilon$ along each axis is at most $H\kappa\varepsilon|\mathcal{B}_{j,k,j',k'}|$, and since there are $2\ell$ dimensions, the total expected number of crossings is $2\ell H\kappa\varepsilon|\mathcal{B}_{j,k,j',k'}|$. Using the upper bound on $|\mathcal{B}_{j,k,j',k'}|$, in total, for any pair of points at distance $\varepsilon$, $\sup_{s \in P} \mathbb{E}[D(T, s)] = O(\ell^3 K^2 H\kappa\varepsilon T)$. By Proposition 40, $\mathbb{E}[\sup_{s \in P} D(T, s)] \leq \sup_{s \in P} \mathbb{E}[D(T, s)] + O(\sqrt{T \log(TK\ell)})$, which in our case is upper bounded by: $O(\ell^3 K^2 H\kappa\varepsilon T + \sqrt{T \log(TK\ell)})$. For $\kappa = \tilde{o}(T)$, $K = O(\text{poly}(T))$ and $\ell = O(\text{poly}(T))$, $\mathbb{E}[\sup_{s \in P} D(T, s)] = \tilde{O}(\varepsilon T)$. Therefore, these loss functions are $\beta$-point dispersed for $\beta = 1/2$, satisfying the statement. $\qquad\square$

**Overview of Algorithm 7**  The generic algorithm for the semi-bandit case was previously developed in Balcan et al. (2020a). We adapt it to our setting and consider an efficient implementation using the approximate integration and sampling from Balcan et al. (2018b) discussed in Definition 37. The semi-bandit-setting algorithm is a continuous version of the Exp3-SET algorithm of Alon et al. (2017b). At each time step, the algorithm learns the revenue function (only) inside the region $\mathcal{P}^{(t)} \ni \boldsymbol{\rho}_t$ that the presented menu belongs to and updates the menu weights for the next round accordingly.

---

**Algorithm 7:** Semi-bandit two-part tariff under smoothed distributional assumptions (Adapted from (Balcan et al., 2020a), Algorithm 1 for two-part tariffs)

**Input:** Step size $\lambda \in [0, 1]$
1: Let $w_1(\boldsymbol{\rho}) = 1$ for all $\boldsymbol{\rho} \in \mathcal{C}$
2: **for** *buyer* $t = 1, \ldots, T$ **do**

    Let $p_t(\boldsymbol{\rho}) = \frac{w_t(\boldsymbol{\rho})}{W_t}$, where $W_t = \int_{\mathcal{C}} w_t(\boldsymbol{\rho}) \, d\boldsymbol{\rho}$;
    Sample $\boldsymbol{\rho}_t$ from $p_t$, present it to buyer $t$, observe the tariff index $j$ and the number of units $k$ selected by the buyer and region $\mathcal{P}^{(t)}$ for which the buyer takes this action; the revenue inside $\mathcal{P}^{(t)}$ is $u_t(\boldsymbol{\rho}) = \mathbb{I}\{k \geq 1\}(p_1^{(i)}(\boldsymbol{\rho}) + kp_2^{(i)}(\boldsymbol{\rho}))$ and the normalized loss is $l_t(\boldsymbol{\rho}) = \frac{H - u_t(\boldsymbol{\rho})}{H}$ for all $\boldsymbol{\rho} \in \mathcal{P}^{(t)}$;
    Let $\hat{l}_t(\boldsymbol{\rho}) = \frac{\mathbb{I}\{\boldsymbol{\rho} \in \mathcal{P}^{(t)}\}}{p_t(\mathcal{P}^{(t)})} l_t(\boldsymbol{\rho})$, where we define $p_t(\mathcal{P}^{(t)}) = \int_{\mathcal{P}^{(t)}} p_t(\boldsymbol{\rho}) \, d\boldsymbol{\rho}$;
    Let $w_{t+1}(\boldsymbol{\rho}) = w_t(\boldsymbol{\rho}) \exp(-\lambda \hat{l}_t(\boldsymbol{\rho}))$ for all $\boldsymbol{\rho}$.

---

**Theorem 12.** *Suppose the buyers' values are drawn from $\mathcal{D}^{(1)} \times \cdots \times \mathcal{D}^{(T)}$, where each $\mathcal{D}^{(t)}$ is $\kappa$-bounded for $\kappa = \tilde{o}(T)$. Then, running the continuous Exp3-SET algorithm (Algorithm 7) for menus of two-part tariffs under semi-bandit feedback has expected regret bounded by $\tilde{O}(H\sqrt{\ell T})$. An efficient implementation has the same regret bound and running time $\tilde{O}((T + 1)^{\text{poly}(\ell, K)} \text{poly}(\ell, \sqrt{T}) + KT\sqrt{T})$.*

*Proof.* For the regret bound, we invoke Theorem 2 of Balcan et al. (2020a), stating that if the loss functions are Lipschitz functions satisfying $\beta$-point-dispersion, running Algorithm 7 has expected regret bounded by

$\tilde{O}(\sqrt{dT} + T^{1-\beta})$, when the loss function is in $[0, 1]$. In our case, $d$, the number of dimensions is $2\ell$, the dispersion parameter $\beta = 1/2$, and the loss function is in $[0, H]$. This implies the regret bound.

Now, we discuss the running time of the algorithm. At each time $t$, using the buyer's valuation vector, the tariff $j$, and the number of units $k$ selected by the buyer, we can determine the region $\mathcal{P}^{(t)}$, where the buyer makes the same selection and whose utility function is linear by solving a linear program (the inequalities in Equation (2)). This computation is done in time $\text{poly}(\ell, K)$. Next, for the integration procedures inside the algorithm, we use the approximate version introduced in Definition 37, and for sampling, we use the efficient implementation demonstrated in Algorithm 6. In particular, we consider $\eta = \zeta = 1/(3\sqrt{T})$. For $\int_{\mathcal{C}} w_t(\boldsymbol{\rho}) \, d\boldsymbol{\rho}$, we use lines 1 through 3 of Algorithm 6 and take the sum of the integration outcomes of line 3, for $\eta' = \eta/4$ and $\zeta' = \zeta/T$. For $p_t(\mathcal{P}^{(t)}) = \int_{\mathcal{P}^{(t)}} p_t(\boldsymbol{\rho}) \, d\boldsymbol{\rho}$ we do the same, except that now we do the integration operations in line 3 only for the regions inside $\mathcal{P}^{(t)}$. For sampling $\boldsymbol{\rho}_t$ from $p_t$, we use the complete procedure Algorithm 6 that takes the regions with linear cumulative utility, $\lambda = \sqrt{2\ell \ln(2H^2\kappa\sqrt{T})/T}/H$, $g = \lambda \sum_{s=0}^{t-1} u_s$ and $\eta = \zeta = 1/(3\sqrt{T})$. Note that since the loss is only updated for $\mathcal{P}^{(t)}$, for any regions outside this part, we do not need to repeat the integration operations in Algorithm 6. This may result in potentially better running time for semi-bandit compared to full-information; however, we do not quantify the improvement. Using union bound, with probability at least $1 - 1/\sqrt{T}$, all the approximate integration and sampling operations performed in the algorithm succeed and the density function of the approximate distribution used for sampling is always within $(1 - \eta)$ fraction of the exact distribution. Using these parameters together with Theorem 1 in (Balcan et al., 2018b) conclude that the same regret bound is achievable from the approximate operations and give the running time in the statement. $\qquad\square$

### A.1.3 Limited Buyer Types

**Full Information Setting**

**Theorem 24.** *In the full information case for length-$\ell$ menus of two-part tariffs, when there are $V$ types of buyers, running Algorithm 2 over the set of menus corresponding to set $\mathcal{E}$ for $\beta = 1/\sqrt{T}$ has regret bounded by $\tilde{O}(H\ell\sqrt{T} \ln(V\ell K))$.*

*Proof.* We run the weighted majority algorithm Algorithm 2 with parameter $\beta = 1/\sqrt{T}$ on the set $\mathcal{E}$ as the set of menus (experts). The proof directly follows from Lemma 23 and Proposition 35. Let $n = |\mathcal{E}|$. Let $\boldsymbol{b}_i$ be the valuation of the buyer at step $i$, and $\bar{b}$ be the vector of valuation of all buyers in rounds 1 through $T$. We denote $\text{Rev}_{\mathcal{E}}()$ as the maximum revenue obtained in the set of $\mathcal{E}$, $\text{OPT}()$ as the optimal revenue, and $\text{Rev}_{\text{WM}}()$ as the revenue obtained from Algorithm 2 on the set of experts $X = \mathcal{E}$. Then,

$$n \leq (V\ell^2 K^2/4)^{2\ell},$$

$$\text{Rev}_{\text{WM}}\left(\bar{b}\right) \geq \text{Rev}(\mathcal{E})\left(\bar{b}\right) - \frac{\beta}{2}\text{Rev}(\mathcal{E})\left(\bar{b}\right) - \frac{H\ln n}{\beta},$$

$$\text{Rev}_{\mathcal{E}}\left(\bar{b}\right) = \sum_{i=1}^{T} \text{Rev}_{\mathcal{E}}\left(\boldsymbol{b}_i\right),$$

$$\text{Rev}_{\mathcal{E}}\left(\boldsymbol{b}_i\right) \geq \text{OPT}\left(\boldsymbol{b}_i\right) - 2K\varepsilon;$$

where the first expression uses the size of $\mathcal{E}$ in Lemma 22, the second expression uses Proposition 35, the third expands the revenue over T terms, and the last uses Lemma 23. Rearranging the terms, we have:

$$\text{Rev}_{\mathcal{E}}\left(\boldsymbol{b}_i\right) \geq \text{OPT}\left(\boldsymbol{b}_i\right) - 2K\varepsilon$$

$$\text{Rev}_{\mathcal{E}}\left(\bar{b}\right) \geq \text{OPT}\left(\bar{b}\right) - 2K\varepsilon T$$

$$\text{Rev}_{\text{WM}}\left(\bar{b}\right) \geq \text{OPT}\left(\bar{b}\right) - 2K\varepsilon T - \frac{\beta HT}{2} - \frac{H\ln n}{\beta}$$

$$\text{Rev}_{\text{WM}}\left(\bar{b}\right) \geq \text{OPT}\left(\bar{b}\right) - 2K\varepsilon T - \frac{\beta HT}{2} - \frac{2\ell H\left(\ln\left(V\ell K\right)\right)}{\beta}$$

We set variables $\varepsilon$ and $\beta$ to minimize the exponent of $T$ in the regret. By setting $\beta = \frac{1}{\sqrt{T}}$ and $\varepsilon = 1/(K\sqrt{T})$, The regret will be $O(H\ell\sqrt{T}\ln{(V\ell K)})$.  $\square$

**Partial Information Setting**   We first show how to estimate the utility of any menu by only using the response of the buyer to a limited number of menus. In doing so, we take advantage of the interdependence of the buyers' responses for different menus to obtain estimates for unused menus. In particular, using *barycentric spanner* concept from Awerbuch and Kleinberg (2008), we devise a basis for the menus such that observing buyers' responses to them is sufficient for estimating the revenue of other menus.

Let $\mathcal{I}$ be a set of length-$V$ indicator vectors, such that for each feasible mapping $\mu$ and option to select $(j,k)$, which is the tariff index and the number of units, there is a vector in $\mathcal{I}$. This vector indicates the (maximal) set of buyer types that select this option in mapping $\mu$. As an example, if in mapping $\mu$, $\{v_2, v_3\}$ is the exact set of valuation types that select the same option $(j,k)$, vector $(0,1,1,0,\dots)$ belongs to $\mathcal{I}$. For $I \in \mathcal{I}$, $\mu_I$ and $(j,k)_I$ denote the corresponding mapping and option to $I$, respectively. Similarly, $I_{\mu,(j,k)}$ is the vector in $\mathcal{I}$, corresponding to mapping $\mu$ and option $(j,k)$. Using principles from linear algebra, since the vectors are $V$-dimensional, there is a set of at most $V$ vectors in $\mathcal{I}$ such that any other vector in $\mathcal{I}$ is a linear combination of the vectors in this set. Awerbuch and Kleinberg make this property stronger and show that there is a set of $V$ vectors in $\mathcal{I}$, called the *barycentric spanner* or *spanner* for short, we denote it by $\mathcal{S}$, such that any member of $\mathcal{I}$ can be written as a linear combination of vectors in $\mathcal{S}$ with coefficients in $[-1,1]$.

**Lemma 41.** *There exists set $\mathcal{S}$ in $\mathcal{I}$ such that, for all $I \in \mathcal{I}$, there exists coefficients $\lambda_1, \dots, \lambda_V \in [-1,1]$, so that $I = \sum_{j=1}^{V} \lambda_i s_j$.*

*Proof.* The statement is a direct corollary of Awerbuch and Kleinberg (2008) Proposition 2.2.  $\square$

Here is the main idea on how to find estimates for the utility of all the menus by only presenting the menus corresponding to the spanner $\mathcal{S}$ to the buyers. First, similar to Balcan et al. (2015), we define function $f_\tau(\cdot)$ for the vectors in $\mathcal{I}$ that will be instrumental in computing the utility for all the menus based on the spanner. Recall that each vector $I$ in $\mathcal{I}$ corresponds to a mapping $\mu_I$ and an option $(j,k)_I$. Let $f_\tau(I)$ be the number of times during a time block $\tau$ that given a menu in $\mathcal{P}_\mu$ the arriving buyer selects option $(j,k)$. First, we show how the quantity of this function on inputs from the spanner is sufficient for finding the revenue of arbitrary menus and then show how to estimate it.

**Lemma 42.** *For each menu $\rho$ and any time block $\tau : t+1, \dots, t+\tau_\ell$, let $u_\tau(\rho)$ represent the average utility of $\rho$ for buyer types in $\tau$. Then,*

$$u_\tau(\rho) = \frac{1}{\ell_\tau} \sum_{(j,k) \in \mathcal{O}} \mathbb{I}\{k \geq 1\} \left(p_1^{(j)}(\rho) + k p_2^{(j)}(\rho)\right) \sum_{i=1}^{V} \lambda_i(I_{\mu_\rho,(j,k)}) f_\tau(s_i)$$

*Proof.* By definition, $u_\tau(\rho)$ is the average utility of menu $\rho$ for buyers arriving in $\tau$. Menu $\rho$, corresponds to a feasible mapping $\mu_\rho$. By definition, the buyers in time block $\tau$ select option $(j,k)$ equal to $f_\tau(I_{\mu_\rho,(j,k)})$ number of times. By Lemma 41, $I_{\mu_\rho,(j,k)}$ can be written as a linear combination of the vectors in the spanner. Furthermore, $f_\tau(.)$ is a linear function as it is equivalent to the dot product of a vector indicating the frequency, i.e., the number of arrivals, of each buyer type during $\tau$ and the function input. Therefore,

$$u_\tau(\rho) = \frac{1}{\ell_\tau} \sum_{(j,k) \in \mathcal{O}} \mathbb{I}\{k \geq 1\} \left(p_1^{(j)}(\rho) + k p_2^{(j)}(\rho)\right) f_\tau(I_{\mu_\rho,(j,k)})$$

$$= \frac{1}{\ell_\tau} \sum_{(j,k) \in \mathcal{O}} \mathbb{I}\{k \geq 1\} \left(p_1^{(j)}(\rho) + k p_2^{(j)}(\rho)\right) \sum_{i=1}^{V} \lambda_i(I_{\mu_\rho,(j,k)}) f_\tau(s_i).$$

$\square$

Let $\hat{f}_\tau(s_i)$ be the estimator to $f_\tau(s_i)/\ell_\tau$ for the spanner vectors. Let $\mu_{s_i}$ be the corresponding mapping to $s_i$. Recall that $f_\tau(s_i)$ is the number of times during $\tau$ that given a menu in $\mathcal{P}_{\mu_{s_i}}$, the arriving buyer, selects

option $(j, k)_{\boldsymbol{s}_i}$. In order to estimate this quantity we present a corresponding menu to $\boldsymbol{s}_i$, i.e., a menu in $\mathcal{P}_{\mu_{\boldsymbol{s}_i}}$, once uniformly at random during the time block $\tau$. If the buyer selects option $(j, k)_{\boldsymbol{s}_i}$, we let $\hat{f}_\tau(\boldsymbol{s}_i)$ equal to 1 and otherwise set it to 0. The next lemma shows that $\hat{f}_\tau(\boldsymbol{s}_i)$ has the same expected value and has range $[0, 1]$. Intuitively, the reason is that due to the uniform random selection of the time step, the estimator has the same expected value.

**Lemma 43** (Adapted from Balcan et al. (2015) Lemma 6.3)**.** *For any $\boldsymbol{s} \in \mathcal{S}$, $\mathbb{E}[\hat{f}_\tau(\boldsymbol{s})]\ell_\tau = f_\tau(\boldsymbol{s})$.*

*Proof.* Note that $\hat{f}_\tau(\boldsymbol{s}) = 1$ if and only if at the time step that menu $\boldsymbol{\rho_s}$ was presented, $(j, k)_{\boldsymbol{s}}$ was selected. Since $\boldsymbol{\rho_s}$ is presented once uniformly at random over the time steps and is independent of the sequence of buyers, the buyer presented with $\boldsymbol{\rho_s}$ is also picked uniformly at random over the time steps. Therefore, $\mathbb{E}[\hat{f}_\tau(\boldsymbol{s})]$ is the probability that a randomly chosen buyer from time block $\tau$ selects $(j, k)_{\boldsymbol{s}}$. □

Now, we prove that the expected value of the utility estimator for each menu is equal to the utility of that menu, i.e., the estimator is unbiased and, moreover, has a bounded range. The utility estimator is defined as follows, where $f_\tau(\boldsymbol{s}_i)/\ell_\tau$ in the utility formula is replaced by its estimator $\hat{f}_\tau(\boldsymbol{s}_i)$.

$$\hat{u}_\tau(\boldsymbol{\rho}) = \sum_{(j,k)\in\mathcal{O}} \mathbb{I}\{k \geq 1\} \left( p_1^{(j)}(\boldsymbol{\rho}) + k p_2^{(j)}(\boldsymbol{\rho}) \right) \sum_{i=1}^{V} \lambda_i(\boldsymbol{I}_{\mu_{\boldsymbol{\rho},(j,k)}}) \hat{f}_\tau(\boldsymbol{s}_i)$$

**Lemma 44.** *For any menu $\boldsymbol{\rho}$, $\mathbb{E}[\hat{u}_\tau(\boldsymbol{\rho})] = u_\tau(\boldsymbol{\rho})$ and $\hat{u}_\tau(\boldsymbol{\rho}) \in [-\ell KVH, \ell KVH]$.*

*Proof.* The proof of the equality of the expectation simply follows from $\hat{u}_\tau(\boldsymbol{\rho})$ and $u_\tau(\boldsymbol{\rho})$ definitions and Lemma 43. Now, we prove the range of the estimator. Since $S$ is a barycentric spanner, for any $\boldsymbol{I} \in \mathcal{I}$, $\lambda_i(\boldsymbol{I}) \in [-1, 1]$. Also, $\hat{f}_\tau(.)$ belongs to $\{0, 1\}$. Also, the utility of the buyer selecting each option in the menu, e.g., $p_1^{(j)}(\boldsymbol{\rho}) + k p_2^{(j)}(\boldsymbol{\rho})$, is always in $[0, H]$. Therefore, using the formula of the estimator, it is bounded by $H$ times the number of options times the number of buyer types. □

We use the algorithm below along with the weighted majority algorithm in the full-information (similar to Algorithm 2) that uses the utility (revenue) estimates. We use $\mathcal{E}$ as the set of experts (menus) and obtain distribution $q$ over set $\mathcal{E}$ as the weight vector.

**Overview of Algorithm 8** First, we provide a high-level structure of the algorithm and then discuss the details. The algorithm operates in time blocks, with each block consisting of exploitation and exploration time steps. The exploration time steps are selected uniformly at random within the block and are limited in number. In an exploitation step, the menu used is the output of the full information algorithm, employing the utility estimators from the previous time block. These menus are always the extreme points of the continuity regions, as discussed at the beginning of the section. During exploration time steps, the corresponding menu to a vector in the spanner is used. At the end of each time block, the algorithm refines the unbiased estimators of the utility of all extreme points using the information gathered in the exploration phases.

$Z$ is the number of time blocks, with each time block consisting of $T/Z$ time steps. The algorithm uniformly at random picks time steps $t_1, \ldots, t_V$ and their permutation $\pi$ in the current time block. Whenever the time step is equal to $t_i$, the algorithm runs an exploration step; otherwise, the algorithm runs an exploitation step. In the exploration step at time step $t_i$, a menu corresponding to $\boldsymbol{s}_i$, $\boldsymbol{\rho}_{\boldsymbol{s}_{\pi(i)}}$, is presented to the arriving buyer and the estimator $\hat{f}_\tau(\boldsymbol{s}_{\pi(i)})$ will be assigned as 1 if the buyer selects $(j, k)_{\boldsymbol{s}_{\pi(i)}}$ and will be assigned as 0, otherwise. At the end of the time block, we update the estimates of the revenue of the menus corresponding to the extreme points.

**Lemma 45.** *[(Balcan et al., 2015) Lemma 6.2] Let $M$ be the set of all actions. For any time block (set of consecutive time steps) $T'$ and action $j \in M$, let $c_{T'}(j)$ be the average loss of action $j$ over $T'$. Assume that $S \subseteq M$ is such that by sampling all actions in $S$, we can compute $\hat{c}_{T'}(j)$ for all $j \in M$ with the following properties: $\mathbb{E}[\hat{c}_{T'}(j)] = c_{T'}(j)$ and $\hat{c}_{T'}(j) \in [-\kappa, \kappa]$. Then there is an algorithm with a loss $L_{alg} \leq L_{min} + O\left(T^{\frac{2}{3}} |S|^{\frac{1}{3}} \kappa^{\frac{1}{3}} \log^{\frac{1}{3}}(|M|)\right)$, where $L_{min}$ is the loss of the best action in hindsight.*

---

**Algorithm 8:** Partial-Information Algorithm for Limited Buyer Types
(adapted from (Balcan et al., 2015) Algorithm 1)

---

**Input:** $V$ : the number of buyer types, $\mathcal{O}$ : the set of menu options $(|\mathcal{O}| = \ell(K+1))$

1: $Z \leftarrow (T^2|\mathcal{O}|^2 V \log(|\mathcal{O}|V))^{1/3}$             $\triangleright$ the number of time blocks

2: Create set $\mathcal{I} = \{I_{\mu,(j,k)}|$ for all options $(j,k)$ and feasible mappings $\mu\}$ such that the $i$th component of $I_{\mu,(j,k)}$ is 1 iff $\boldsymbol{v}_i$ selects $(j,k)$ in $\mu$ and is 0 otherwise.

3: Find a barycentric spanner $\mathcal{S} = \{\boldsymbol{s}_1, ..., \boldsymbol{s}_V\}$ for $\mathcal{I}$. For every $\boldsymbol{s} \in \mathcal{S}$, let $\mu_{\boldsymbol{s}}$ be the corresponding mapping, $(j,k)_{\boldsymbol{s}}$, the corresponding option, and $\boldsymbol{\rho}_{\boldsymbol{s}}$ a menu in $\mathcal{P}_{\mu_{\boldsymbol{s}}}$.

4: **for** *all* $\boldsymbol{I} \in \mathcal{I}$ **do**

     let $\boldsymbol{\lambda}(\boldsymbol{I})$ be the representation of $\boldsymbol{I}$ in spanner $\mathcal{S}$. That is $\sum_{i=1}^{V} \lambda_i(\boldsymbol{I})\boldsymbol{s}_i = \boldsymbol{I}$.

5: Let $q_1$ be the uniform distribution over $\mathcal{E}$.          $\triangleright$ initial weight vector over menus in $\mathcal{E}$

6: **for** $\tau = 1, ..., Z$ **do**                                         $\triangleright$ time blocks

     Choose a random permutation $\pi$ over $[V]$ and $t_1, \ldots, t_V$ from $[T/Z]$.;

     **for** $t = (\tau - 1)(T/Z) + 1, ..., \tau(T/Z),$ **do**          $\triangleright$ time steps in a time block

         **if** $t = t_i$ *for some* $i \in [V]$, **then**          $\triangleright$ exploration time step

             $\boldsymbol{\rho}_t \leftarrow \boldsymbol{\rho}_{\boldsymbol{s}_{\pi(j)}};$

             If $(j,k)_{\boldsymbol{s}_{\pi(j)}}$ is selected, then $\hat{f}_\tau(\boldsymbol{s}_{\pi(j)}) \leftarrow 1$, otherwise $\hat{f}_\tau(\boldsymbol{s}_{\pi(j)}) \leftarrow 0;$

         **else**                                    $\triangleright$ exploitation time step

             draw $\boldsymbol{\rho}_t$ at random from distribution $q_\tau;$

     **for** *all* $\boldsymbol{\rho} \in \mathcal{E}$, *for* $\mu$ *such that* $\boldsymbol{\rho} \in \mathcal{P}_\mu$, **do**

         $\hat{u}_\tau(\boldsymbol{\rho}) = \sum_{(j,k) \in \mathcal{O}} \mathbb{I}\{k \geq 1\} \left(p_1^{(j)}(\boldsymbol{\rho}) + k p_2^{(j)}(\boldsymbol{\rho})\right) \sum_{i=1}^{V} \lambda_i(\boldsymbol{I}_{\mu_{\boldsymbol{\rho}},(j,k)}) \hat{f}_\tau(\boldsymbol{s}_i).;$

     Call Algorithm 2 for experts $\mathcal{E}$ and $(\hat{u}_\tau)$ as their revenue function;

     And receive $q_{\tau+1}$ as a distribution over all mixed strategies in $\mathcal{E}$.

---

We are now ready to prove the main result of this section.

**Theorem 25.** *In the partial information (bandit) case for length-$\ell$ menus of two-part tariffs, when there are $V$ different types of buyers, there is an algorithm with regret bound of $\tilde{O}(T^{2/3}\ell(HKV)^{1/3}\log^{1/3}(V\ell K))$.*

*Proof.* In Lemma 45, $|S|$ is the number of dimensions (barycentric spanner set), $\kappa$ is the maximum revenue times the number of buyer types times the number of their options (entries in the menu), $|M|$ is the number of extreme points. In our case, $|S| = 2\ell$, $\kappa = H\ell KV$, and $|M| \leq (V\ell^2 K^2/4)^{2\ell}$. By Lemma 44, the expected value of the estimated utility is equal to the exact value of utility with range $[-H\ell KV, H\ell KV]$.

Using Lemma 45, the regret for menus of two-part tariffs is bounded by

$$O(T^{2/3}\ell^{1/3}(H\ell KV)^{1/3}\ell^{1/3}\log^{1/3}(V\ell K)) \in O(T^{2/3}\ell(HKV)^{1/3}\log^{1/3}(V\ell K)).$$

$\square$

The following quantifies the regret of simply running the Exp3 algorithm on the set of extreme points.

**Proposition 46.** *In the partial information case for length-$\ell$ menus of two-part tariffs when there are $V$ buyer types, running Algorithm 3 over menus corresponding to $\mathcal{E}$ for $\beta = \gamma = T^{-1/3}$ has regret bound $O\left(T^{2/3}\ell H(V\ell^2 K^2/4)^{2\ell}\ln(V\ell K)\right)$.*

*Proof.* The proof is similar to that of Theorem 11. We denote $\text{Rev}_{\text{Exp3}}()$ as the revenue obtained from the Exp3 algorithm as presented in Algorithm 3 on the set of menus corresponding to $\mathcal{E}$. Let $n$ denote the number of such menus. $\boldsymbol{b}_i$ is the valuation of the buyer at step $i$, and $\bar{b}$ is the sequence of valuation of all buyers in rounds 1 through $T$. $\text{Rev}_{\mathcal{E}}()$ is the maximum revenue obtained in the set $\mathcal{E}$ and OPT() is the

optimal revenue.

$$n \le (V\ell^2 K^2/4)^{2\ell},$$

$$\text{Rev}_{\text{Exp3}}\left(\bar{b}\right) \ge \text{Rev}(\mathcal{E})\left(\bar{b}\right) - \left(\gamma + \frac{\beta}{2}\right)\text{Rev}(\mathcal{E})\left(\bar{b}\right) - \frac{Hn\ln n}{\beta\gamma},$$

$$\text{Rev}_{\mathcal{E}}\left(\bar{b}\right) = \sum_{i=1}^{T}\text{Rev}_{\mathcal{E}}\left(\boldsymbol{b}_i\right),$$

$$\text{Rev}_{\mathcal{E}}\left(\boldsymbol{b}_i\right) \ge \text{OPT}\left(\boldsymbol{b}_i\right) - 2K\varepsilon;$$

where the first expression uses the size of $\mathcal{E}$ in Lemma 22, the second expression uses Proposition 36, the third expands the revenue over T terms, and the last uses Lemma 23. Rearranging the terms, we have:

$$\text{Rev}_{\mathcal{E}}\left(\boldsymbol{b}_i\right) \ge \text{OPT}\left(\boldsymbol{b}_i\right) - 2K\varepsilon$$

$$\text{Rev}_{\mathcal{E}}\left(\bar{b}\right) \ge \text{OPT}\left(\bar{b}\right) - 2K\varepsilon T$$

$$\text{Rev}_{\text{Exp3}}\left(\bar{b}\right) \ge \text{OPT}\left(\bar{b}\right) - 2K\varepsilon T - \left(\gamma + \frac{\beta}{2}\right)HT - \frac{Hn\ln n}{\beta\gamma}$$

$$\text{Rev}_{\text{Exp3}}\left(\bar{b}\right) \ge \text{OPT}\left(\bar{b}\right) - 2K\varepsilon T - \left(\gamma + \frac{\beta}{2}\right)HT - \frac{2\ell H(V\ell^2 K^2/4)^{2\ell}\left(\ln\left(V\ell K\right)\right)}{\beta\gamma}$$

We set variables $\varepsilon$ in $\mathcal{E}$ and $\beta = \gamma$ as a function of $T$ to minimize the exponent of $T$ in the regret. By setting $\beta = \gamma = T^{-1/3}$ and $\varepsilon = T^{-1/2}$, the regret is $O\left(T^{2/3}\ell H(V\ell^2 K^2/4)^{2\ell}\ln\left(V\ell K\right)\right)$. $\qquad\square$

**Remark.** The standard technique for the partial information algorithm of running the Exp3 algorithm on the extreme points leads to a regret bound that is exponential in the size of the menu as stated in Proposition 46; however, Algorithm 8 has regret bound polynomial in the size of the menus. Therefore, the new technique results in a significant improvement.

## A.2 Distributional Learning

**Theorem 26.** *In the distributional setting, for length-$\ell$ menus of two-part tariffs, there exists a learning algorithm with sample complexity $\frac{H^2}{2\varepsilon^2}(2\ell\ln\left(\frac{2KH\ell}{\varepsilon}\right) + \ln\left(2/\delta\right))$, and running time $\frac{H^2}{2\varepsilon^2}\left(2\ell\ln\left(\frac{2KH\ell}{\varepsilon}\right) + \ln\left(2/\delta\right)\right)K\ell\left(\frac{2HK\ell}{\varepsilon}\right)^{2\ell}$.*

*Proof.* We need to find the number of samples such that with probability $1 - \delta$, the difference between the expected revenue of our algorithm and the optimal revenue is at most $\varepsilon$. Note that since our algorithm uses discretization of possible menus, we face two types of errors: the discretization error, and the usual empirical error in a PAC learning setting. We find the sample complexity and discretization parameters such that the total error is bounded by $\varepsilon$.

The possible number of menus after discretization using parameter $N$ is computed by the following formula.

$$|\mathcal{H}| = (H/\alpha)^{2\ell}.$$

Using uniform convergence in the PAC learning setting, the sample complexity for empirical error $\varepsilon'$ is as follows.

$$|S| \ge \frac{H^2}{2\varepsilon'^2}\left(\ln|\mathcal{H}| + \ln\left(2/\delta\right)\right).$$

Replacing $\ln\mathcal{H}$ we have,

$$|S| \ge \frac{H^2}{2\varepsilon'^2}\left(2\ell\ln\left(H/\alpha\right) + \ln\left(2/\delta\right)\right).$$

Also, the revenue loss compared to the optimum for arbitrary buyer $i$ with valuation $\boldsymbol{v}_i$ is:

$$\text{Rev}_{M'}\left(\boldsymbol{v}_i\right) \ge \text{OPT}\left(\boldsymbol{v}_i\right) - 2K\ell\alpha.$$

The total error (from discretization and empirical error), when the empirical error is set to $\varepsilon'$, is

$$2K\ell\alpha + \varepsilon'.$$

By setting $2K\ell\alpha = \varepsilon'$, we have

$$\alpha = \frac{\varepsilon'}{2K\ell},$$

Replacing $\alpha$ gives the following sample complexity:

$$|S| \geq \frac{H^2}{2\varepsilon'^2}\left(2\ell\ln\left(H/\alpha\right) + \ln\left(2/\delta\right)\right)$$
$$\geq \frac{H^2}{2\varepsilon'^2}\left(2\ell\ln\left(2K\ell H/\varepsilon'\right) + \ln\left(2/\delta\right)\right)$$

which by replacing $\varepsilon'$ with $\varepsilon/2$ results in $\varepsilon$ total error.

The computational complexity of finding the empirical optimal menu for $|S|$ buyers and menu of size $\ell$ is:

$$O(|S|K\ell|\mathcal{H}|) = |S|K\ell\left(\frac{2HK\ell}{\varepsilon}\right)^{2\ell}.$$

This implies the efficiency of the algorithm. $\qquad\square$

**Lemma 47.** *The running time of distributional learning algorithm for two-part tariffs in (Balcan et al., 2020b) is at least*

$$\left(c\left(\frac{H}{\varepsilon}\right)^2\left(18\ell\log\left(8^2K^2\ell^3\right) + \log\frac{1}{\delta}\right)\right)^{2\ell+1}K^{4\ell+2}(2\ell)^{2+1/18}.$$

*Proof.* The algorithm involves computing $N^{2\ell}K^{4\ell}$ regions, where $N$ is $c(H/\varepsilon)^2(18\ell\log\left(8K^2\ell^3\right) + \log\frac{1}{\delta})$, and solving a linear program for each region with $2\ell$ variables and $NK^2$ constraints, which takes $\tilde{O}((2\ell)^{2+1/18}NK^2)$. $\qquad\square$

**Comparison with previous results.** The sample complexity using the pseudo-dimension method of (Balcan et al., 2018c) is $O(H^2/\varepsilon^2(\ell\log\left(K\ell\right) + \log\left(1/\delta\right)))$ and the best previously-known running time (Balcan et al., 2022b) is $O\left(R^2(2\ell)^{2\ell+1}KH^2/\varepsilon^2(\ell\log\left(K\ell\right) + \log\left(1/\delta\right))\right)$, where $R$ the number of discontinuity regions is bounded by $O([H^2/\varepsilon^2(\ell\log\left(K\ell\right) + \log\left(1/\delta\right))]^3K)$, resulting in the worst case running time of $O\left(\left(H^2/\varepsilon^2(\ell\log\left(K\ell\right) + \log\left(1/\delta\right))\right)^{2\ell+1}K^{4\ell+2}(2\ell)^{2+1/18}\right)$ due to (Balcan et al., 2020b; 2022b) (See Lemma 47).

## B   Missing Proofs of Section 4

### B.1   Online Learning

Similar to the section on two-part tariffs, using the outcome of the discretization summarized in Theorem 27, we show a reduction to a finite number of experts and run standard learning algorithms (weighted majority and Exp3) over the menus in the discretized set.

#### B.1.1   Full Information

In the full information setting, the seller sees the revenue generated for all the possible menus. To design an online algorithm in this case, we use a variant of the weighted majority algorithm by (Auer et al., 1995). The experts in our case are the discretized menus from the previous section, denoted in the algorithm by set $X = m_1, \ldots, m_n$. Furthermore, $\boldsymbol{v}_t$ is the valuation of the buyer are time $t$ and $\text{Rev}_k(\boldsymbol{v}_1, \ldots, \boldsymbol{v}_t)$ is the cumulative revenue of menu $m_k$ for the buyers until time step $t$.

Similar to two-part tariffs, we use Algorithm 2 for the full information case. The only difference is that since the maximum revenue in lotteries is $mH$, as opposed to two-part tariffs where it is $H$, in the algorithm we need to replace $H$ with $mH$.

**Proposition 48** ((Auer et al., 1995), Theorem 3.2). *For any sequence of valuations $\bar{v}$,*

$$\text{Rev}_{\text{WM}}(\bar{v}) \geq \left(1 - \frac{\beta}{2}\right) \text{OPT}_X(\bar{v}) - \frac{mH \ln n}{\beta},$$

*where $X = m_1, \ldots, m_n$ are the set of experts (lottery menus), $\text{Rev}_{\text{WM}}(\bar{v})$ is the expected revenue outcome of Algorithm 2 where $H$ is replaced with $mH$, and $\text{OPT}_X(\bar{v})$ is the revenue of the optimal menu in $X$.*

**Theorem 28.** *In the full information case for length-$\ell$ menus of lotteries, running Algorithm 2 over the discretized set of menus specified in Theorem 27 for $\alpha = T^{-1}$, $\beta = T^{-0.5}$, $K = T^{0.5}$, and $\delta = T^{-0.5}$ has regret $\tilde{O}(m^2 H \ell \sqrt{T})$.*

*Proof.* Let $n$ be the number of menus resulting from Algorithm 5. Let $\boldsymbol{v}_i$ be the valuation of the buyer at step $i$, and $\bar{v}$ be the vector of valuation of all buyers in rounds 1 through $T$. We denote $\text{Rev}_{M'}()$ as the maximum revenue obtained in the set of menus resulting from Algorithm 5, $\text{OPT}()$ as the optimal revenue, and $\text{Rev}_{\text{WM}}()$ as the revenue obtained from the weighted majority algorithm discussed above on the set of outcome menus of Algorithm 5. We have

$$n = (1/\alpha^{\ell m + \ell})(\ln(Hm/\alpha))^{lm},$$

$$\text{Rev}_{\text{WM}}(\bar{v}) \geq \text{Rev}_{M'}(\bar{v}) - \frac{\beta}{2}\text{Rev}(M')(\bar{v}) - \frac{mH \ln n}{\beta},$$

$$\text{Rev}_{M'}(\bar{v}) = \sum_{i=1}^{T} \text{Rev}_{M'}(\boldsymbol{v}_i),$$

$$\text{Rev}_{M'}(\boldsymbol{v}_i) \geq \text{OPT}(\boldsymbol{v}_i)(1-\delta)(1-\alpha)^K - (2K+1)\alpha - mH(1-\delta)^K;$$

where the first expression is a result of Algorithm 5, the second expression uses Proposition 48, the third expands the revenue over $T$ terms, and the last uses Theorem 27. Rearranging the terms, we have:

$$\begin{aligned} \text{Rev}_{M'}(\boldsymbol{v}_i) &\geq \text{OPT}(\boldsymbol{v}_i)(1-\delta)(1-\alpha)^K - (2K+1)\alpha - mH(1-\delta)^K \\ &\geq \text{OPT}(\boldsymbol{v}_i) - \text{OPT}(\boldsymbol{v}_i)\left(1 - (1-\delta)(1-\alpha)^K\right) - (2K+1)\alpha - mH(1-\delta)^K \\ &\geq \text{OPT}(\boldsymbol{v}_i) - mH\left(1 - (1-\delta)(1-\alpha)^K\right) - (2K+1)\alpha - mH(1-\delta)^K \\ \text{Rev}_{M'}(\bar{v}) &\geq \text{OPT}(\bar{v}) - mHT\left(1 - (1-\delta)(1-\alpha)^K\right) - T(2K+1)\alpha - mHT(1-\delta)^K \\ \text{Rev}_{\text{WM}}(\bar{v}) &\geq \text{OPT}(\bar{v}) - mHT\left(1 - (1-\delta)(1-\alpha)^K\right) - T(2K+1)\alpha \\ &\quad - mHT(1-\delta)^K - \frac{\beta mHT}{2} - \frac{mH \ln n}{\beta} \end{aligned}$$

We set variables $K$, $\alpha$, $\delta$, and $\beta$ as a function of $T$ to minimize the exponent of $T$ in the regret. The regret is upper bounded by

$$mHT\left(1 - (1-\delta)(1-\alpha)^K\right) + T(2K+1)\alpha + mHT(1-\delta)^K + \frac{\beta mHT}{2} + \frac{mH \ln n}{\beta},$$

$$\leq mHT\left(1 - (1-\delta)(1-\alpha)^K\right) + T(2K+1)\alpha + mHT(1-\delta)^K + \frac{\beta mHT}{2} + \frac{mHO\left(\ell m \ln(Hm/\alpha)\right)}{\beta};$$

where the inequality is followed by upper bounding $n$. By setting $\alpha = T^{-1}$, $\beta = T^{-0.5}$, $K = T^{0.5}$, and $\delta = T^{-0.5}$ the regret is bounded by $\tilde{O}(m^2 H \ell \sqrt{T})$. $\qquad\square$

**Theorem 29.** *In the full information case for arbitrary length menus of lotteries, running Algorithm 2 on menus specified in Theorem 27 for $\alpha = T^{-1/(2m+2)}$, $\beta = T^{-1/(m+1)}$, $K = T^{1/(m+1)}$, and $\delta = T^{-1/(m+1)}$ has regret $\tilde{O}(mHT^{1-1/(2m+4)} \ln^m(mHT))$.*

*Proof.* The proof follows the same argument as Theorem 28. The only difference in the parameters is $n$, the number of experts, which in this case is $n = 2^{(1/\alpha^{m+1})(\ln(Hm/\alpha))^m}$. We set variables $K$, $\alpha$, $\delta$, and $\beta$ as a function of $T$ to minimize the exponent of $T$ in the regret. The regret is upper bounded by the formula below after substituting $n$

$$mHT\left(1 - (1-\delta)(1-\alpha)^K\right) + T(2K+1)\alpha + mHT(1-\delta)^K$$
$$+ \frac{\beta mHT}{2} + \frac{mH(1/\alpha^{m+1})(\ln(Hm/\alpha))^m ln2}{\beta}$$

By setting $\alpha = T^{-1/(2m+2)}$, $\beta = T^{-1/(m+1)}$, $K = T^{1/(m+1)}$, and $\delta = T^{-1/(m+1)}$, the regret is bounded by $\tilde{O}(mHT^{1-1/(2m+4)} \ln^m(mHT))$. □

### B.1.2   Bandit Setting

In the partial information setting, the seller does not see the outcome for all the possible menus and only observes the outcome of the menu used (the lottery chosen by the buyer). Similar to the two-part tariffs results, to design an online algorithm in this case, we use a version of the Exp3 algorithm in (Auer et al., 1995). This variant of the Exp3 algorithm contains the weighted majority algorithm (Algorithm 2) a subroutine. At each step, we mix the probability distribution $\pi$, used by the weighted majority algorithm, with the uniform distribution to obtain a modified probability distribution $\bar{\pi}$, which is then used to select a menu from our discretized set. Following the lottery chosen by buyer $t$, we use the price paid (the gain from the chosen menu) to formulate a simulated gain vector, which is then used to update the weights maintained by the weighted majority algorithm.

Similar to two-part tariffs, we use Algorithm 3 for the bandit case. The only difference is that since the maximum revenue in lotteries is $mH$, as opposed to two-part tariffs where it is $H$, in the algorithm we need to replace $H$ with $mH$.

**Proposition 49** ((Auer et al., 1995), Theorem 4.1). *For any sequence of valuations $\bar{v}$,*

$$\text{Rev}_{\text{Exp3}}(\bar{v}) \geq \text{OPT}_X - \left(\gamma + \frac{\beta}{2}\right)\text{OPT}_X - \frac{mHn\ln n}{\beta\gamma},$$

*where $X = m_1, \ldots, m_n$ are the set of experts (lottery menus), $\text{Rev}_{\text{Exp3}}(\bar{v})$ is the expected revenue outcome of Algorithm 3 where $H$ is replaced with $mH$, and $\text{OPT}_X(\bar{v})$ is the revenue of the optimal menu in $X$.*

**Theorem 30.** *In the partial information case for length-$\ell$ menus of lotteries, running Algorithm 3 over discretized set of menus in Theorem 27 for $\alpha = T^{-1/(\ell m+2)}$, $\beta = \gamma = T^{-1/(4\ell m+8)}$, $K = T^{1/(2\ell m+4)}$, and $\delta = T^{-1/(2\ell m+4)}$ has regret $\tilde{O}(m^2 H\ell T^{1-1/(2\ell m+4)} \ln^{\ell m+1}(mHT))$.*

*Proof.* The proof follows the same logic as that of Theorem 28. We denote $\text{Rev}_{\text{Exp3}}()$ as the revenue obtained from the Exp3 algorithm described above on the set of outcome menus of Algorithm 5. Similar to the proof of Theorem 28, in what follows $n$ denotes the number of menus resulting from the procedure Algorithm 5. $\boldsymbol{v}_i$ is the valuation of the buyer at step $i$, and $\bar{v}$ is the vector of valuation of all buyers in rounds 1 through $T$. $\text{Rev}_{M'}()$ is the maximum revenue obtained in the set of menus resulting from Algorithm 5 and $\text{OPT}()$ as the optimal revenue.

$$n = (1/\alpha^{\ell m+\ell})(\ln(Hm/\alpha))^{lm},$$
$$\text{Rev}_{\text{Exp3}}(\bar{v}) \geq \text{Rev}_{M'} - \left(\gamma + \frac{\beta}{2}\right)\text{Rev}_{M'} - \frac{mHn\ln n}{\beta\gamma},$$
$$\text{Rev}_{M'}(\bar{v}) = \sum_{i=1}^{T} \text{Rev}_{M'}(\boldsymbol{v}_i),$$
$$\text{Rev}_{M'}(\boldsymbol{v}_i) \geq \text{OPT}(\boldsymbol{v}_i)(1-\delta)(1-\alpha)^K - (2K+1)\alpha - mH(1-\delta)^K;$$

where the first expression is a result of Algorithm 5, the second expression uses Proposition 49, the third expands the revenue over $T$ terms, and the last uses Theorem 27. Rearranging the terms, we have:

$$\text{Rev}_{\text{Exp3}}(\bar{v}) \geq \text{Rev}_{M'}(\bar{v}) - \left(\gamma + \frac{\beta}{2}\right)\text{Rev}_{M'}(\bar{v}) - \frac{mHn\ln n}{\beta\gamma}$$

$$\geq \text{Rev}_{M'}(\bar{v}) - \left(\gamma + \frac{\beta}{2}\right)mHT - \frac{mHn\ln n}{\beta\gamma}$$

$$\geq \text{OPT}(\bar{v}) - mHT\left(1 - (1-\delta)(1-\alpha)^K\right) - T(2K+1)\alpha - mHT(1-\delta)^K$$

$$- \left(\gamma + \frac{\beta}{2}\right)mHT - \frac{mHn\ln n}{\beta\gamma}$$

We set variables $K$, $\alpha$, $\delta$, $\beta$, and $\gamma$ as a function of $T$ to minimize the exponent of $T$ in the regret. After substituting $n$, the regret is upper bounded by

$$mHT\left(1 - (1-\delta)(1-\alpha)^K\right) + T(2K+1)\alpha + mHT(1-\delta)^K + \left(\gamma + \frac{\beta}{2}\right)mHT$$

$$+ \frac{2\ell m^2 H(1/\alpha^{\ell m+\ell})\left(\ln\left(Hm/\alpha\right)\right)^{\ell m+1}}{\beta\gamma}$$

By setting $\alpha = T^{-1/(\ell m+2)}$, $\beta = \gamma = T^{-1/(4\ell m+8)}$, $K = T^{1/(2\ell m+4)}$, and $\delta = T^{-1/(2\ell m+4)}$, the regret is bounded by $\tilde{O}(m^2 H\ell T^{1-1/(2\ell m+4)}\ln^{\ell m+1}(mHT))$. $\qquad\square$

## B.2 Limited Buyer Types

The ideas for designing a specific algorithm specific to the limited buyer types in the menus of lotteries are similar to those for menus of two-part tariffs. There are a few changes that we overview here.

One of the main differences is the menu options $\mathcal{O}$. Unlike two-part tariffs that given a menu, the buyer needed to select a tariff and number of units that maximized the buyer's utility; for menus of lotteries, the options are exactly aligned with menu entries, and $|\mathcal{O}| = \ell + 1$ for length-$\ell$ lotteries. The mechanism designer's utility (revenue) given menu $\boldsymbol{\rho}$ is equal to $p^{(j)}(\boldsymbol{\rho})$ if the buyer selects entry $j$. The buyer selects entry $j$, if this entry results in higher utility than any other entry in menu $\boldsymbol{\rho}$. These inequalities identify regions $\mathcal{P}_\mu$, where the buyer's utility maximizing option is aligned with $\mu$.

**Definition 50** (menu option for menus of lotteries, $\mathcal{O}$). *Index $j$ such that $0 \leq j \leq \ell$ indicating a lottery index in the menu is a menu option. We denote the set of all menu options as $\mathcal{O}$. This set identifies all potential actions of a buyer when presented with a menu.*

**Definition 51** (mapping $\mu$, feasible mappings, $\mathcal{P}_\mu$). *A mapping $\mu$ is a function from buyer types, $\boldsymbol{v}_1, \ldots, \boldsymbol{v}_V$ to menu options $j = 0, 1, \ldots, \ell$, where $j$ is the lottery index assigned to the buyer type. Mapping $\mu$ is feasible if there is a menu corresponding to the mapping, i.e., a menu that if presented to the buyers, each buyer selects their corresponding option in the mapping as their utility maximizing option. $\mathcal{P}_\mu$ denotes the region of the parameter space corresponding to $\mu$, i.e., the set of menus inducing mapping $\mu$.*

**Lemma 52.** *For each feasible mapping $\mu$, as defined in Definition 51, $\mathcal{P}_\mu$ is a convex polytope with hyperplane boundaries.*

*Proof.* For a fixed buyer type $\boldsymbol{i}$ and option $j = 0, \ldots, \ell$, let $\mathcal{P}_j^{(i)}$ be the set of all parameter vectors $\boldsymbol{\rho}$ corresponding to the length-$\ell$ menus that buyer type $i$ selects option $j$. The buyer selects option $j$ for menu $\boldsymbol{\rho}$ if this option produces more utility for the buyer than any other option. Formally,

$$\sum_{k=1}^m v(\boldsymbol{e}_k)\phi^{(j)}[k](\boldsymbol{\rho}) - p^{(j)}(\boldsymbol{\rho}) \geq \sum_{k=1}^m v(\boldsymbol{e}_k)\phi^{(j')}[k](\boldsymbol{\rho}) - p^{(j')}(\boldsymbol{\rho}); \quad \forall j'.$$

The above inequalities identify a convex polytope of parameter vectors (menus $\boldsymbol{\rho}$) with hyperplane boundaries. $\mathcal{P}_\mu$ is the intersection of $\mathcal{P}_{\mu(i)}^{(i)}$ for $i = 1, \ldots, V$. Therefore, $\mathcal{P}_\mu$ is also a convex region with hyperplane boundaries. $\qquad\square$

**Lemma 53.** *For each feasible mapping $\mu$ and any sequence of buyer valuations $\bar{b}$ the cumulative utility, $\sum_i u(\boldsymbol{b}_i, \boldsymbol{\rho})$, is linear in $\mathcal{P}_\mu$.*

*Proof.* We show that for any buyer valuation $\boldsymbol{v}_i$ in the sequence, $u(\boldsymbol{v}_i, \rho)$ is linear in the region. Proving this claim is sufficient for concluding the statement. Let $j = \mu(\boldsymbol{v}_i)$, i.e., $j$ is the lottery index that buyer valuation $\boldsymbol{v}_i$ selects under $\mu$. Therefore, the utility for this buyer for menu $\boldsymbol{\rho} \in \mathcal{P}_\mu$ is $\sum_{k=1}^m v(\boldsymbol{e}_k)\phi^{(j)}[k](\boldsymbol{\rho}) - p^{(j)}(\boldsymbol{\rho})$. Note that $\phi^{(j)}[k](\boldsymbol{\rho})$ is a coordinate of $\boldsymbol{\rho}$ and therefore, has a linear dependence on $\boldsymbol{\rho}$. Therefore, since the option that each buyer valuation selects is fixed inside $\mathcal{P}_\mu$, the utility is also linear. $\square$

**Lemma 54.** *The number of extreme points for menus of lotteries, $|\mathcal{E}|$, is at most $(V\ell^2)^{m(\ell+1)}$.*

*Proof.* Length-$\ell$ menus of lotteries occupy a $\ell(m+1)$-dimensional parameter space. In each $d$-dimensional space, an extreme point is the intersection of $d$ linearly independent hyperplanes. The total number of hyperplanes defining the regions is $\mathcal{H} = V\binom{\ell}{2}$, where for each buyer type compares the utility of two menu entries. Out of these hyperplanes, we need $\ell(m+1)$ of them to intersect for an extreme point. Therefore, the number of extreme points is at most $\binom{\mathcal{H}}{\ell(m+1)}$, implying the statement. $\square$

The following lemma bounds the loss in utility where the set of menus is limited to the extreme points $\mathcal{E}$. The proof is similar to Balcan et al. (2015); however, the loss depends on the problem-specific utility functions.

**Lemma 55.** *Let $\mathcal{E}$ be as defined in Definition 21, then for any sequence of buyer valuations $\bar{b} = \boldsymbol{b}_1, \ldots, \boldsymbol{b}_T$, and $\boldsymbol{\rho}^*$ as the optimal menu in the hindsight:*

$$max_{\boldsymbol{\rho} \in \mathcal{E}} \sum_{t=1}^T u(\boldsymbol{b}_t, \boldsymbol{\rho}) \geq \sum_{t=1}^T u(\boldsymbol{b}_t, \boldsymbol{\rho}^*) - \varepsilon T.$$

*Proof.* The proof is similar to that of Lemma 23. The only difference is in step (vi) which computes the loss in revenue between menus that are at $\varepsilon$ $L1$ distance. In menus of lotteries, this distance implies a price difference of at most $\varepsilon$ in any of the lotteries in the menu, and therefore causes $\varepsilon$ total loss per time step. $\square$

**Full Information Setting**

**Theorem 31.** *In the full information case for length-$\ell$ menus of lotteries, when there are $V$ types of buyers, there is an algorithm with regret bound of $O(m^2 H\ell\sqrt{T} \ln(V\ell))$.*

*Proof.* The proof follows the same logic as of theorem 24. We run the weighted majority algorithm (Algorithm 2, where $H$ is replaced by $mH$) with parameter $\beta = 1/\sqrt{T}$ on the set $\mathcal{E}$ as the set of menus (experts). The proof directly follows from Lemma 55 and Proposition 48. Let $n = |\mathcal{E}|$. Let $\boldsymbol{b}_i$ be the valuation of the buyer at step $i$, and $\bar{b}$ be the vector of valuation of all buyers in rounds 1 through $T$. We denote $\text{Rev}_\mathcal{E}()$ as the maximum revenue obtained in the set of $\mathcal{E}$, $\text{OPT}()$ as the optimal revenue, and $\text{Rev}_{\text{WM}}()$ as the revenue obtained from Algorithm 2 on the set of experts $X = \mathcal{E}$. Then,

$$n \leq (V\ell^2)^{m(\ell+1)},$$

$$\text{Rev}_{\text{WM}}\left(\bar{b}\right) \geq \text{Rev}(\mathcal{E})\left(\bar{b}\right) - \frac{\beta}{2}\text{Rev}(\mathcal{E})\left(\bar{b}\right) - \frac{mH \ln n}{\beta},$$

$$\text{Rev}_\mathcal{E}\left(\bar{b}\right) = \sum_{i=1}^T \text{Rev}_\mathcal{E}\left(\boldsymbol{b}_i\right),$$

$$\text{Rev}_\mathcal{E}\left(\boldsymbol{b}_i\right) \geq \text{OPT}\left(\boldsymbol{b}_i\right) - \varepsilon;$$

where the first expression uses the size of $\mathcal{E}$ in Lemma 54, the second expression uses Proposition 48, the third expands the revenue over T terms, and the last uses Lemma 55. Rearranging the terms, we have:

$$\text{Rev}_{\mathcal{E}}\left(\boldsymbol{b}_i\right) \geq \text{OPT}\left(\boldsymbol{b}_i\right) - \varepsilon$$

$$\text{Rev}_{\mathcal{E}}\left(\bar{b}\right) \geq \text{OPT}\left(\bar{b}\right) - \varepsilon T$$

$$\text{Rev}_{\text{WM}}\left(\bar{b}\right) \geq \text{OPT}\left(\bar{b}\right) - \varepsilon T - \frac{\beta m H T}{2} - \frac{mH \ln n}{\beta}$$

$$\text{Rev}_{\text{WM}}\left(\bar{b}\right) \geq \text{OPT}\left(\bar{b}\right) - \varepsilon T - \frac{\beta m H T}{2} - \frac{m^2(\ell+1)H\left(\ln\left(V\ell\right)\right)}{\beta}$$

We set variables $\varepsilon$ and $\beta$ to minimize the exponent of $T$ in the regret. By setting $\beta = \frac{1}{\sqrt{T}}$ and $\varepsilon = 1/(\sqrt{T})$, The regret will be $O(m^2 H \ell \sqrt{T} \ln\left(V\ell\right))$. □

**Partial Information (Bandit) Setting** In the partial information setting, the change in the menu options also affects the definition of set $\mathcal{I}$ that consists of indicator vectors over the buyer types that select the same menu entry $j$ in a mapping $\mu$. The changes that need to be made in Algorithm 8 to work for menus of lotteries include changing $|\mathcal{O}|$ to $\ell + 1$, using option (menu entry) $j$ instead of $(j, k)$, and changing utility from $\mathbb{I}\{k \geq 1\}\left(p_1^{(j)}(\boldsymbol{\rho}) + k p_2^{(j)}(\boldsymbol{\rho})\right)$ to $p^{(j)}(\boldsymbol{\rho})$. After making these changes, we can perform the modified algorithm to achieve a bounded regret.

**Lemma 56.** *For any menu $\boldsymbol{\rho}$, $\mathbb{E}[\hat{u}_\tau(\boldsymbol{\rho})] = u_\tau(\boldsymbol{\rho})$ and $\hat{u}_\tau(\boldsymbol{\rho}) \in [-mH(\ell+1)V, mH(\ell+1)V]$.*

*Proof.* The proof is similar to Lemma 56. The proof of the equality of the expectation simply follows from $\hat{u}_\tau(\boldsymbol{\rho})$ and $u_\tau(\boldsymbol{\rho})$ definitions and Lemma 43. Now, we prove the range of the estimator. Since $S$ is a barycentric spanner, for any $\boldsymbol{I} \in \mathcal{I}$, $\lambda_i(\boldsymbol{I}) \in [-1, 1]$. Also, $\hat{f}_\tau(.)$ belongs to $\{0, 1\}$. Additionally, the utility of the buyer selecting each option in the menu, e.g., $p^{(j)}(\boldsymbol{\rho})$, is always in $[0, mH]$. Therefore, using the formula of the estimator, it is bounded by $mH$ times the number of options times the number of buyer types. □

**Theorem 32.** *In the partial information (bandit) case for length-$\ell$ menus of lotteries, when there are $V$ different types of buyers, there is an algorithm with regret bound of $O(T^{2/3}(\ell m)^{4/3}(HV)^{1/3}\log^{1/3}(V\ell))$.*

*Proof.* The proof follows the same logic as of theorem 25. In Lemma 45, $|S|$ is the number of dimensions (barycentric spanner set), $\kappa$ is the maximum revenue times the number of buyer types times the number of their options (entries in the menu), $|M|$ is the number of extreme points. In our case, $|S| = \ell(m+1)$, $\kappa = mHV(\ell+1)$, and $|M| \leq (V\ell^2)^{m(\ell+1)}$. By Lemma 56, the expected value of the estimated utility is equal to the exact value of utility with range $[-mH(\ell+1)V, mH(\ell+1)V]$.

Using Lemma 45, the regret for menus of lotteries is bounded by

$$O(T^{2/3}(\ell m)^{4/3}(HV)^{1/3}\log^{1/3}(V\ell)).$$

□

## B.3 Distributional Learning

**Theorem 34.** *For length-$\ell$ menus of lotteries, there is a discretization-based distributional learning algorithm with sample complexity $\tilde{O}\left(m^2H^2/\varepsilon^2(\ell m + \ln\left(2/\delta\right))\right)$, and running time $\tilde{O}\left(\left(2m^2H^2/\varepsilon^2\right)^{\ell m+\ell+1}\ell(\ell m + \ln\left(2/\delta\right))\ln^{\ell m}\left(mH/\varepsilon\ln\left(mH/\varepsilon\right)\right)\right).$*

*Proof.* We need to find the number of samples such that with probability $1 - \delta$, the difference between the expected revenue of our algorithm and the optimal revenue is at most $\varepsilon$. Note that since our algorithm uses discretization of possible menus, we face two types of errors: the discretization error, and the usual empirical

error in a PAC learning setting. We find the sample complexity and discretization parameters such that the total error is bounded by $\varepsilon$.

The possible number of menus after discretization using Algorithm 5 with parameter $\alpha$ is computed by the following formula.

$$|\mathcal{H}| = (1/\alpha^{\ell m+\ell})\left(\ln\left(Hm/\alpha\right)\right)^{\ell m}$$

Using uniform convergence in the PAC learning setting, the sample complexity for empirical error $\varepsilon'$ is as follows.

$$|S| \geq \frac{m^2 H^2}{2\varepsilon'^2}\left(\ln|\mathcal{H}| + \ln\left(2/\delta\right)\right)$$

Replacing $\ln\mathcal{H}$ we have,

$$|S| \geq \frac{m^2 H^2}{2\varepsilon'^2}\left(\ell m(\ln(1/\alpha) + \ln\ln\left(mH\right) + \ln\left(2/\delta\right)\right)$$

Also, the revenue loss compared to the optimum for arbitrary buyer $i$ with valuation $\boldsymbol{v}_i$ when using Algorithm 5 with parameters $\alpha$, $K$, and $d$ (we use $d$ instead of $\delta$ in Algorithm 5 and reserve $\delta$ for $(\varepsilon, \delta)$-learning) is computed by the following formula.

$$\mathrm{Rev}_{M'}\left(\boldsymbol{v}_i\right) \geq \mathrm{OPT}(\boldsymbol{v}_i)(1-d)(1-\alpha)^K - (2K+1)\alpha - mH(1-d)^K$$

The total error (from discretization and empirical error), when the empirical error is set to $\varepsilon'$, is

$$mH[1 - (1-d)(1-\alpha)^K] + (2K+1)\alpha + mH(1-d)^K + \varepsilon'$$

By setting $d = \varepsilon'/(2mH)$, $K = 2mH/\varepsilon' \ln(mH/\varepsilon')$, and $\alpha = \varepsilon'/(2m^2 H^2 \ln(mH/\varepsilon'))$, the total mistake is less than $4\varepsilon'$.

Replacing these parameters and substituting $\varepsilon'$ with $\varepsilon/4$ to satify total error $\varepsilon$, we have the following sample complexity:

$$\begin{aligned}|S| &\geq \frac{m^2 H^2}{2\varepsilon'^2}\left(\ell m(\ln(1/\alpha) + \ln\ln\left(mH\right) + \ln\left(2/\delta\right)\right)\\ &= \tilde{O}\left(\frac{m^2 H^2}{\varepsilon^2}(\ell m + \ln\left(2/\delta\right))\right)\end{aligned}$$

Also, replacing the parameters we have:

$$|\mathcal{H}| = O\left(\left(\frac{2m^2 H^2}{\varepsilon^2}\right)^{\ell m+\ell}\ln^{\ell m}\left(mH/\varepsilon\ln\left(mH/\varepsilon\right)\right)\right)$$

The computational complexity of finding the empirical optimal menu for $|S|$ buyers and menu of size $\ell$ is:

$$|S|\ell|\mathcal{H}| = \tilde{O}\left(\left(\frac{2m^2 H^2}{\varepsilon^2}\right)^{\ell m+\ell+1}\ell(\ell m + \ln\left(2/\delta\right))\ln^{\ell m}\left(mH/\varepsilon\ln\left(mH/\varepsilon\right)\right)\right)$$

This implies the computational complexity of the algorithm. $\qquad\square$

**Theorem 57.** *For arbitrary-length menus of lotteries, there is a discretization-based distributional learning algorithm with sample complexity*

$$O\left(\frac{m^2 H^2}{\varepsilon^2}\left((32m^2 H^2/\varepsilon^2)^{m+1}\ln^m\left(mH/\varepsilon\ln(mH/\varepsilon)\right)\ln^{m+1}\left(mH/\varepsilon\right) + \ln\left(1/\delta\right)\right)\right),$$

*and running time*

$$O\left(2^{(32m^2 H^2/\varepsilon^2)^{m+1}\ln^m\left(mH/\varepsilon\ln(mH/\varepsilon)\right)\ln^{m+1}\left(mH/\varepsilon\right)}\right).$$

*Proof.* This proof follows the same line as the proof of Theorem 34. We need to find the number of samples such that with probability $1 - \delta$, the difference between the expected revenue of our algorithm and the optimal revenue is at most $\varepsilon$. Note that since our algorithm uses discretization of possible menus, we face two types of errors: the discretization error, and the usual empirical error in a PAC learning setting. We find the sample complexity and discretization parameters such that the total error is bounded by $\varepsilon$.

The possible number of menus after discretization using Algorithm 5 with parameter $\alpha$ is computed by the following formula.

$$|\mathcal{H}| = O\left(2^{(1/\alpha^{m+1})(\ln(Hm/\alpha))^m}\right)$$

Using uniform convergence in the PAC learning setting, the sample complexity for empirical error $\varepsilon'$ is as follows.

$$|S| \geq \frac{m^2 H^2}{2\varepsilon'^2}\left(\ln|\mathcal{H}| + \ln(2/\delta)\right)$$

Replacing $\ln \mathcal{H}$ we have,

$$|S| \geq \frac{m^2 H^2}{2\varepsilon'^2}\left(\frac{\ln^m(Hm/\alpha)}{\alpha^{m+1}} + \ln(2/\delta)\right)$$

Also, the revenue loss compared to the optimum for arbitrary buyer $i$ with valuation $\boldsymbol{v}_i$ when using Algorithm 5 with parameters $\alpha$, $K$, and $d$ (we use $d$ instead of $\delta$ in Algorithm 5 and reserve $\delta$ for $(\varepsilon, \delta)$-learning) is computed by the following formula.

$$\text{Rev}_{M'}(\boldsymbol{v}_i) \geq \text{OPT}(\boldsymbol{v}_i)(1-d)(1-\alpha)^K - (2K+1)\alpha - mH(1-d)^K$$

The total error (from discretization and empirical error) when the empirical error is set to $\varepsilon'$ is

$$mH[1 - (1-d)(1-\alpha)^K] + (2K+1)\alpha + mH(1-d)^K + \varepsilon'$$

By setting $d = \varepsilon'/(2mH)$, $K = 2mH/\varepsilon' \ln(mH/\varepsilon')$, and $\alpha = \varepsilon'/(2m^2 H^2 \ln(mH/\varepsilon'))$, the total mistake is less than $4\varepsilon'$.

Replacing these parameters and substituting $\varepsilon'$ with $\varepsilon/4$ to satify total error $\varepsilon$, we have the following sample complexity:

$$|S| \geq \frac{m^2 H^2}{2\varepsilon'^2}\left(\frac{\ln^m(mH/\alpha)}{\alpha^{m+1}} + \ln(2/\delta)\right)$$
$$= O\left(\frac{m^2 H^2}{\varepsilon^2}\left((32m^2 H^2/\varepsilon^2)^{m+1}\ln^m(mH/\varepsilon\ln(mH/\varepsilon))\ln^{m+1}(mH/\varepsilon) + \ln(1/\delta)\right)\right)$$

Also, replacing the parameters we have:

$$|\mathcal{H}| = O\left(2^{(1/\alpha^{m+1})\ln^m(Hm/\alpha)}\right)$$
$$= O\left(2^{(32m^2 H^2/\varepsilon^2)^{m+1}\ln^m(mH/\varepsilon\ln(mH/\varepsilon))\ln^{m+1}(mH/\varepsilon)}\right)$$

The computational complexity of finding the empirical optimal menu for $|S|$ buyers is the number of potential menus $|\mathcal{H}|$ times $|S|$ times the maximum size of a menu which is $O(\ln(\mathcal{H}))$. □

**Lemma 58.** *The sample complexity of length $\ell$ menus of lotteries using the techniques in (Balcan et al., 2018b) is bounded by*

$$c\left(\frac{H}{\varepsilon}\right)^2\left(9\ell(m+1)\log\left(4\ell(m+1)\left((\ell+1)^2 + m\ell\right)\right) + \log\frac{1}{\delta}\right).$$

*Proof.* Balcan et al. (2018c) introduce *delineability* as a condition to upper bound the pseudo-dimension and therefore, the sample complexity. They show the class of lotteries is $(\ell(m + 1), (\ell + 1)^2 + m\ell)$-delineable. Also, if $\mathcal{M}$ is a mechanism class that is $(d, t)$-delineable, then the pseudo dimension of $\mathcal{M}$ is at most $9d \log(4dt)$. Therefore, the pseudo-dimension for menus of lotteries is bounded by $9\ell(m + 1) \log \left(4\ell(m + 1) \left((\ell + 1)^2 + m\ell\right)\right)$. Furthermore, the sample complexity is at most $c(H/\varepsilon)^2 \left(\text{Pdim}(\mathcal{H}) + \log\left(1/\delta\right)\right)$, which by replacing pseudo dimension for this class of mechanism completes the proof. $\qquad\square$

