# OpenReview forum: "New Guarantees for Learning Revenue Maximizing Menus of Lotteries and Two-Part Tariffs"
_TMLR — Accepted by TMLR_

### Review · Reviewer_fukN · 2023-12-12

**Summary Of Contributions:**

The paper studies the learnability of two classes of mechanisms: menus of lotteries and two-part tariffs. The authors study both offline and online variants. In the former, it is assumed there are samples available from the buyer's evaluation. In the latter, the buyers arrive one at a time without assumptions on their evaluation functions. In both settings, the buyers are interested in learning approximate revenue maximizing mechanisms.

**Audience:**

Yes

**Claims And Evidence:**

Yes

**Requested Changes:**

-- I think the exposition needs to be improved. A more high-level idea of the techniques/challenges should be added.

-- A more clear discussion of comparisons to prior work should be added. This has been done mainly in the introduction but I think it would be valuable to have this at each section.

On the technical side, here are some questions I like the authors to address:

(1) Are there lower bounds known for any of the regret bounds?

(2) Some lower bounds have unusual dependencies in the regret terms (e.g., exponential on $\ell$ in Theorem 8). Are these just the artifacts of discretizing to an exponential number of experts and the use of-the-shelf algorithms? Can these be avoided?

(3) Some algorithms are not computationally efficient (e.g., Theorems 7 and 8). Are there hardness results known for these problems?

**Strengths And Weaknesses:**

Strengths:

-- The paper's results are very comprehensive. They study two rich mechanisms in both online and offline settings and provide theoretical guarantees in all of the settings.


Weakness:

-- The paper is pretty dense and hard to read at times.

-- The contributions are rather limited and at times it is not clear what's new and what can be carried over from previous work (especially all those by Balcan et al).

---

> ### Author Response · Authors · 2023-12-23
> **Response to reviewer fukN**
>
> Thank you for your thoughtful review and questions. We appreciate your feedback and are happy to incorporate it and edit the paper accordingly, specifically to improve the clarity of the contributions.
>
> Regarding the requested changes, we are happy to add high-level ideas of the techniques and challenges and expand the comparison to prior work in each section.
>
> We summarize below our contributions including additional detail on how it compares to prior work. We will add these to the paper for further clarification.
>
> ## *Contributions*
>
> There has been a lot of recent interest and progress in the area of revenue maximization through data-driven mechanism design. However, the results on online learning have been limited and specifically did not expand to mechanisms that go beyond selling separately, such as menus of two-part tariffs and lotteries. We provide both conceptual and technical novelties. For completeness, we provide a list summary of our contributions below. In the next revision of the paper, we will add this list in the introduction to clarify the novel conceptual and technical contributions as well as applications of previously developed ideas and techniques to menus of lotteries and two-part tariffs.
>
> ### *Summary of Contributions*:
>
> We study the learnability of menus of two-part tariffs and lotteries both in the online and distributional settings, provide a variety of results, and advance the state-of-the-art in several aspects.
>
> First, we overview the results related to **menus of two-part tariffs**.
> - By extracting structural properties, we develop a **novel discretization method** that identifies a finite set of menus that approximate the revenue of any arbitrary menu, including the optimum for any valuation. This allows the development of new no-regret online learning algorithms as well as improved distributional learning algorithms (see the two bullet points below).
> - We provide the **first** no-regret online learning algorithms under adversarial inputs, smooth distributional assumptions, and limited buyer type assumptions (under full information, bandit setting, and semi-bandit setting).
> - We also provide a novel distributional learning algorithm for menus of two-part tariffs. Our algorithm chooses several menus of two-part tariffs in a data-independent way (via data-independent discretization) and then selects the best of them based on data. This is much simpler than previous algorithms [BSV2018, BPS2020] for the same problem, yet it enjoys improved runtime guarantees in the worst-case scenario when the length of the menu is more than one.
> - For limited buyer types, we provide improved regret bounds for both the full-information and bandit settings. We show a reduction to online linear optimization, which allows us to obtain no-regret guarantees by presenting buyers with menus that correspond to a barycentric spanner.
>
> Next, we overview our results related to **menus of lotteries**.
> - We extend the discretization of menus of lotteries developed by Dughmi–Han–Nisan [2014]. Our extension is three-fold: we remove the lower bound assumption on value distribution, support additive valuations, and provide improved regret bounds and running times when the size of the menu is limited.
> - We provide the **first** no-regret online learning algorithms under adversarial inputs.
> - Compared to the previous distributional learning results for fixed-length menus [BSV2018], our algorithm requires similar sample complexity; however, it has an efficient implementation.
> - We provide evidence that menus of lotteries may not satisfy dispersion---a sufficient condition to provide a no-regret algorithm under smooth distributional assumption---without assuming extra structures about the optimal solution. Menus of lotteries are the **first** family of mechanisms where there is evidence or potential failure of the dispersion property.
> - For limited buyer types, we provide improved regret bounds for both the full-information and bandit settings. We show a reduction to online linear optimization, which allows us to obtain no-regret guarantees by presenting buyers with menus that correspond to a barycentric spanner.

---

> ### Author Response · Authors · 2023-12-23
> **Response to reviewer fukN, cont.**
>
> ## *Technical contributions compared to prior work*
>
> **Online Algorithms:** No online learning algorithm existed for menus of lotteries and two-part tariffs.
>
> **Discretization:** The establishment of data-driven discretization (and the subsequent online learning and distributional learning algorithms) is in contrast with previous findings. For other data-driven algorithm design problems, such as data-driven clustering and data-driven learning to branch that share a similar piecewise structure in the utility functions, it was proven that algorithms that use data-independent discretization could perform very poorly [BNVW2017,BDSV2018]. Thus, by contrast, our work shows the power of data-independent discretization for data-driven mechanism design and algorithm design more generally.
>
> **Structural properties of utility functions:** For menus of lotteries and two-part tariffs, it has been shown in [BSV2018] that based on the values observed from users until time $t$, the parameter space is partitioned into convex regions with hyperplane boundaries such that the utility inside each region satisfies Lipschitz continuity. We give a more refined characterization by showing that (1) the utility function inside each region is linear, and (2) the boundary hyperplanes constitute a multiset of parallel hyperplanes. Properties (1) and (2) are important for establishing dispersion and obtaining no-regret online learning algorithms under smooth distributional assumptions. Furthermore, property (1) is used in limited buyer types.
>
> **Distributional learning algorithms:** For both menus of lotteries and two-part tariffs, distributional learning results were presented before [BSV2018,BPS2020]. Our algorithms choose several menus in a data-independent way (via data-independent discretization) and then select the best of them based on the data (empirical risk minimization over a cover); however, the prior algorithms [BSV2018,BPS2020] optimize over the infinite space based on the sampled data (empirical risk minimization over the entire space utilizing geometric structures of utility functions). In the context of two-part tariffs, our algorithm is much simpler than prior ones for the same problem, yet it enjoys improved worst-case runtime guarantees compared to them [BSV2018,BPS2020] when the length of the menu is more than one (Theorem 26). In the context of lotteries, compared to the previous distributional learning results for fixed-length menus [BSV2018], our algorithm requires similar sample complexity; however, it has an efficient implementation (Theorems 34 and 57). For arbitrary-length menus, our algorithm provides similar sample complexity and running time compared to [DHN2014]; however, it works for a slightly more general setting.
>
> **Limited buyer types:** [BBHP2015] study a security games setting, in which at each time step, the defender has a mixed strategy (a probability distribution) for protecting the attack targets. Knowing this mixed strategy, the attacker selects a target to attack, which maximizes the attacker’s utility (depending on the attacker’s type). [BBHP2015] provides no-regret algorithms under limited attackers. Our work is inspired by [BBHP2015], and the general structure of the algorithm is similar. Although the problem settings are quite different, and it is not apparent whether similar ideas could work in both settings, we are able to adapt the ideas to provide no-regret algorithms for menus of two-part tariffs and lotteries. This adaptation requires establishing new properties and definitions for our problem settings. We have given proper attributions to the previous paper throughout the section.
>
> **Satisfying dispersion property and online learning under smooth distributional assumption:** In the context of menus of lotteries, we show the first evidence since 2018 for the failure of the dispersion property, which is quite surprising. This property has been shown to hold for many different algorithm and mechanism design problems [BDV2018,BDP2020,BS2021,BKST2022]. In the context of menus of two-part tariffs, the proof of the dispersion property is inspired by [BDV2018]. The algorithms for full-information, semi-bandit, and bandit settings were previously developed in a general format [BDV2018,BDP2020] for any problem setting satisfying dispersion property. We adapt those algorithms to our settings in algorithms 4, 6 and 7.

---

> ### Author Response · Authors · 2023-12-23
> **Response to reviewer fukN, cont.**
>
> ## *Answers to technical questions*
>
> (1) Are there lower bounds known for any of the regret bounds?
>
> There are lower bounds with respect to the dependence on $T$. The dependence on $T$ (square root) is tight in the full information case [Ed. Nisan, Roughgarden Tardos, Vazirani, Algorithmic Game Theory, Theorem 4.8]. In the bandit setting, Kleinberg–Slivkins–Upfal [2008] provide a lower bound for general globally Lipschitz functions. Our dependence on $T$ matches this lower bound even though the utility functions in our case are only piecewise Lipschitz (not globally Lipschitz).  Finding a lower bound for the dependence on the other parameters is a nice open direction. We will add this discussion to the next revision of the paper.
>
> (2) Some lower bounds have unusual dependencies in the regret terms (e.g., exponential on $\ell$ in Theorem 8). Are these just the artifacts of discretizing to an exponential number of experts and the use of-the-shelf algorithms? Can these be avoided?
>
> Yes, the exponential dependencies in $\ell$ in the regret terms are the artifacts of the discretization and using standard algorithms.
>
> The question of whether such dependencies can be avoided is precisely what motivated us to study more structured settings such as the smooth distributional assumptions and the limited buyer type setting. The exponential dependence matches a lower bound by Kleinberg–Slivkins–Upfal [2008] for (globally) Lipschitz functions. This dependence appears both in our discretization-based and dispersion-based algorithms.  However, in the limited buyers type case where we utilize the knowledge of the potential buyer types and interdependence of utilities across experts, the dependence is improved. We will add this discussion to the next revision of the paper.
>
> (3) Some algorithms are not computationally efficient (e.g., Theorems 7 and 8). Are there hardness results known for these problems?
>
> We are not aware of hardness results for these problems; however, even in the simpler problem of distributional learning, the previous results were not computationally efficient, and we improve upon those.  At this point, there are no better known algorithms in the worst case. It is a nice open direction to understand the computational complexity and explore the existence of more efficient algorithms.
>
> We have also taken steps to explore the possibility of more efficient algorithms by adding extra structure, e.g., smooth distributional assumption or limited buyer type assumption, to our settings. Establishing the ‘dispersion’ property under smooth distributional assumptions enables us to use more refined online learning algorithms (a continuous version of multiplicative weight update algorithm that uses the geometric structure of utility functions), but this property was not studied before for menus of lotteries or two-part tariffs. One of our contributions is establishing this property for menus of two-part tariffs and obtaining more efficient algorithms. However, surprisingly we show this property does not work for lotteries. In the limited buyer type settings, we utilize the knowledge of the potential buyer types and interdependence of utilities across menus and provide algorithms with improved running times.

---

### Review · Reviewer_B31f · 2023-12-22

**Summary Of Contributions:**

Summary: This papers studies the problem of learning the revenue-maximizing mechanism of multi-item single buyer auction in various learning settings. The  authors consider learning two classes of direct mechanisms, one is two-part tariffs and the other is lotteries. This submission provides new algorithms for learning revenue-maximizing mechanisms for both classes of mechanisms that achieve better performance in many learning settings including online learning (adversarial/smoothed distribution) and distributional learning.

Model: In the setting of two-part tariffs, there are multiple copies of the same item, the buyer has a utility function u(k) where k is the number of items he gets. The menu of a two-part tariff is a list of pairs (p1_i, p2_i), and the buyer is asked to choose a menu line (p1_i, p2_i) together with a number k denoting the number of items. Then the price of each menu line (p1_i, p2_i) is given by  p1_i + k * p2_i. The buyer with utility function u(k) picks a menu line i and number k that maximize u(k) -  (p1_i + k * p2_i).

The second setting this paper considers is the most general one — menus of lotteries. In this setting the seller is selling multiple items, and the (additive) buyer has values of each item. Each menu line consists of a price p and probabilities of giving each item to the buyer. Given a menu, the buyer picks a menu line that maximizes his expected total value minus the price.

In traditional Bayesian mechanism design, a prior distribution of types of buyers is known, the problem is to find the revenue-maximizing mechanism for a specific distribution. However, in the learning model this paper considers, the seller has limited information about the buyers and the problem is to learn the revenue-maximizing mechanism given data in various learning settings.


Contribution: The authors give algorithms in many different settings of learning that improve existing guarantees in the literature. Specifically, for two-part tariffs, in online learning settings (full-information and bandit), the authors provide the first no-regret learning algorithms under adversarial and smoothed distributional assumptions. For lotteries, the authors give the first no-regret learning algorithm in the online setting with adversarial inputs. As for the distributional learning setting, the authors give algorithms for both two-part tariffs and lotteries that improve the running times compared to existing works in the literature in certain regimes.

In addition, the authors observe that the learning problem of optimal lotteries may not satisfy a property called dispersion. As a consequence, this problem cannot be solved by a no-regret learning algorithm that works for problems satisfied dispersion in the literature under smooth distributional assumption.

The paper also studies the special case where the number of types of buyers is small. In this case, the authors design algorithms that improve the regret bound for both full-information and bandit setting.

Techniques: The key technical barrier of applying classic learning algorithms is that the space of mechanisms is infinite. To overcome it, the main idea underpins most learning algorithms in the paper is discretization of space of mechanisms. The authors propose a novel data-independent discretization scheme for two-part tariffs and extends a discretization scheme first proposed by Dughmi et al. (2014) for lotteries. After discretizing the space of mechanisms, the rest part is just to apply the respective classic algorithm in each learning setting (e.g. EXP3 for partial info online learning,  weighted majority for full-info, and empirical risk minimization for distributional learning).

**Audience:**

Yes

**Broader Impact Concerns:**

None.

**Claims And Evidence:**

Yes

**Requested Changes:**

The authors claim that they extend algorithms by Dughmi et al. but they did not compare them in detail. Therefore, I suggest the authors to make a clear discussion on similarities and differences between their discretization scheme for lotteries and the one in Dughmi et al. (2014).

**Strengths And Weaknesses:**

Strengths:
1. The paper is well-written especially the discussion of related works.
2. This paper considers an important problem and improves learning guarantees in many different settings.

Weaknesses:
1. I feel the technical contribution is a bit incremental. If I understand correctly, the only novel ideas are discretization of two-part tariffs and limited buyer types.

---

> ### Author Response · Authors · 2023-12-25
> **Response to Reviewer B31f**
>
> Thank you for your thoughtful review. We appreciate your feedback and are happy to incorporate it and edit the paper accordingly.
>
> Our discretization is completely novel for **menus of two-part tariffs**. Recent work on this topic did not even try this approach [BSV2018,BPS2020], and as we discuss below and in the paper, this leads to improved results.
>
> As per the requested changes, we will add a clear discussion on similarities and differences between our discretion scheme and Dughmi–Han–Nisan [2014] for **menus of lotteries**. In summary (and as mentioned briefly at the beginning of page 3 in the submission), our extension is three-fold:
> - We remove the lower bound assumption on value distribution. [DHN 2014] assume values are between [1, H]. We extend the discretization scheme to work when there is no lower bound on value distributions, i.e., values are in [0, H].
> - Supporting additive valuations. The original discretization in [DHN 2014] works for unit-demand valuations.
> - We also modify the algorithm to support limited-length menus. As a consequence, we are able to provide improved regret bounds and running times when the size of the menu is limited.
>
> These extensions are done by small modifications to [DHN 2014]’s algorithm and they expand the scope of the application of the scheme. We will clarify these in Section 4.1 and proof of Theorem 27.
>
> We believe our paper offers further technical contributions other than discretization. Some of them are summarized below:
> - Distributional learning algorithms: For both menus of lotteries and two-part tariffs, distributional learning results were presented before [BSV2018,BPS2020]. Our discretization-based techniques lead to improvements over the previously best known algorithms. Our algorithms choose several menus in a data-independent way (via data-independent discretization) and then select the best of them based on the data (empirical risk minimization over a cover); however, the prior algorithms [BSV2018,BPS2020] optimize over the infinite space based on the sampled data (empirical risk minimization over the entire space utilizing the geometric structure of utility functions). In the context of two-part tariffs, our algorithm is much simpler than prior ones for the same problem, yet it enjoys improved worst-case runtime guarantees compared to them [BSV2018,BPS2020] when the length of the menu is more than one (Theorem 26). In the context of lotteries, compared to the previous distributional learning results for fixed-length menus [BSV2018], our algorithm requires similar sample complexity; however, it has an efficient implementation (Theorems 34 and 57). For arbitrary-length menus, our algorithm provides similar sample complexity and running time compared to [DHN2014]; however, it works for a slightly more general setting.
> - In the limited buyer type setting, we show a reduction to online linear optimization, which allows us to obtain no-regret guarantees by presenting buyers with menus that correspond to a barycentric spanner. Our reduction is inspired by [BBHP2015]; however, we show how to make it work in this very different context as well. The problem settings are quite different, and it is not immediately apparent whether similar ideas could work in both settings. We are able to adapt the ideas to provide no-regret algorithms for menus of two-part tariffs and lotteries. This adaptation requires establishing new properties and definitions for our problem settings
> - Establishing dispersion property for menus of two-part tariffs. Dispersion property has been introduced by [BDV2018,BDP2020] and known to hold for various algorithm and mechanism design problems. However, this was not studied for menus of two-part tariffs.
> - The **first** evidence of failure of dispersion property. In the context of menus of lotteries, we show the first evidence since 2018 for the failure of the dispersion property, which is quite surprising.
>
> We will update the paper to more clearly spell out our contributions.

---

### Review · Reviewer_Pkib · 2024-01-05

**Summary Of Contributions:**

This work investigates the learnability of two classes of economic mechanisms, namely menus of lotteries and two-part tariffs, at the intersection of learning theory and computational economics. The authors concentrate on learning high-revenue mechanisms of this type from buyer valuation data in both distributional settings, where they have access to buyers' valuation samples ahead of time, and the more difficult and less-studied online settings, where buyers arrive one-at-a-time and no distributional assumption about their values is made. The study presented the first online learning algorithms for lotteries and two-part tariffs with regret-bound guarantees. They are able to provide a reduction to online learning over a finite number of parameters. Furthermore, they demonstrate a reduction to online linear optimization in the case of few customers, allowing them to get no-regret assurances by presenting purchasers with menus that match to a barycentric spanner.

**Audience:**

Yes

**Broader Impact Concerns:**

This paper constitutes a valuable contribution intersecting two critical domains: learning theory and mechanism design. It provides new guarantees for learning revenue-maximizing menus of lotteries and two-part tariffs. The framework has the potential to be applied to a more general class of problems.

**Claims And Evidence:**

Yes

**Requested Changes:**

Major Comments:

1. Theorem 1 demonstrates that for a given menu of two-part tariffs $M$ and a parameter $0<\alpha<1$, Algorithm 1 generates a revised menu $M'$ with a revenue at least as high as that of $M$ minus $2K\alpha l$, where $2K\alpha l$ may be considerably large. Notably, achieving a smaller margin necessitates the selection of a smaller $\alpha$. However, a notable challenge emerges: the algorithm's implementability becomes increasingly questionable as the value of $\alpha$ decreases due to the exponential growth in space requirements. This issue underscores a critical consideration regarding the practicality and feasibility of deploying the algorithm in scenarios where minimizing the revenue gap is crucial.

2. In Theorem 7, when $\alpha$ is set to $1/\sqrt{T}$, the calculated running time of Algorithm 1 is established as $\min{(HT)^{2l}, 2^{H^2T^2}}$. Notably, the total running time should inherently exceed that of a single round, suggesting a higher runtime than $\min\{(HT)^{2l}, 2^{H^2T^2}\}$. However, the theorem's concluding statement presents a time complexity of $O(TlK\min{H^{2l}T^l,2^{H^2T}})$, which raises concerns as it could potentially be less than $T^{2l}$. Similar concerns arise regarding the analysis in Theorem 8. These inconsistencies prompt questions regarding the accuracy of the derived time complexities and their alignment with the expected runtime behavior of the algorithm across varying settings.

3. Theorem 8 establishes the regret bound for the partial information scenario concerning length-$l$ menus of two-part tariffs as $\tilde{O}(T^{1-1/(2(1+l))l(K+H^{2l+1}})$. This regret bound scales in the order of $T^{1-1/(2(1+l))}$. However, a critical consideration arises concerning the tightness of this derived regret bound. It is important to provide additional analysis or evidence to ascertain whether this bound is indeed tight. Demonstrating the tightness of the regret bound is pivotal for understanding the algorithm's performance limits and ensuring a comprehensive understanding of its behavior under varying conditions.

4. The paper presents Theorem 10, demonstrating regret analysis under smooth distributions for the full information setting, while Theorem 11 elucidates regret in the bandit setting. However, an enhanced exposition regarding the technical contributions compared to prior works is needed. From the technical perspective, the proof of Theorem 10 builds upon Theorem 32 of Balcan et al. (2018b) and Theorem 1 in Balcan et al. (2018b); the proof of Theorem 11 builds upon Theorem 3 in Balcan et al. (2018b) and Theorem 7 in Balcan and Sharma 2021. Highlighting the distinct technical nuances and innovations specific to this paper within these established frameworks would accentuate its unique contributions.

Moreover, it's notable that the order of $T$ in the regret under smooth distributions for both full information (Theorem 12) and bandit settings (Theorem 11) coincides with the regret derived under adversarial inputs (Theorem 7). Considering the purportedly advantageous properties of $\kappa$-bounded distributions, there is an opportunity to explore whether Theorems 11 and 12 achieve the optimal regret bounds.

Additionally, incorporating discussions after theorems and technical results would significantly aid readers in better grasping and contrasting the differences between these theorems. These discussions could shed light on the distinctive aspects of the algorithms or approaches employed, facilitating a clearer understanding and comparison of their performance across various settings.

5. The absence of numerical experiments in this paper is notable. Integrating numerical experiments would significantly enhance the paper's comprehensiveness by providing empirical evidence on two crucial fronts:

Algorithm Implementability: Numerical experiments can offer insights into the algorithm's practical implementation, particularly in terms of its running time. Demonstrating the algorithm's computational efficiency through empirical evaluations would fortify its applicability in real-world scenarios.

Algorithm Performance Comparison: Additionally, numerical experiments allow for a comparative assessment of the algorithm's performance against existing state-of-the-art algorithms. Such comparisons provide valuable insights into the algorithm's efficacy, offering a nuanced understanding of its strengths and weaknesses relative to other approaches.

Introducing numerical experiments could bridge the gap between theoretical propositions and practical applicability, offering a more holistic view of the algorithm's capabilities and positioning within the domain.

Other Minor Comments:

1. Lemma Placement: Consider relocating certain results, such as Lemma 3-5, to the appendix. These lemmas might be considered self-evident or straightforward, allowing the main text to prioritize more pivotal and substantive results.

2. Algorithm 3 Notation Clarification: In Algorithm 3, the statement "For the selected menu $k^*$, set $g_{k^*}(t)$ to be the price paid by buyer $t$" requires clarification. Specifically, verifying if $g_{k^*}(t)$ simply equals $p_1^{j}+kp_2^{j}$, where $j$ denotes the selected menu might aid reader comprehension. Additionally, the notation usage, where $k$ represents quantity in the context while $k^*$ signifies the selected menu, appears confusing. Clarifying this notation discrepancy could enhance readability and understanding for readers.

**Strengths And Weaknesses:**

The paper is well-written overall and makes good contributions to the field. However, several critical points need addressing. Please see my major and minor comments below.

---

> ### Author Response · Authors · 2024-01-09
> **Response to Reviewer Pkib**
>
> Thank you for your thoughtful review. We appreciate your feedback and are happy to incorporate it and edit the paper accordingly.
>
> Please see our responses to the questions below:
>
> Major comments:
>
> 1. Your analysis indicating that the loss of revenue depends on $\alpha$ and small $\alpha$ affecting the performance of learning algorithms is an accurate reflection. However, we do not consider the dependence on $\alpha$ as a downside. The rounding procedure is designed parametrically so we can flexibly set the parameters to tune for the desired objective, e.g., precision, regret bounds, and sample complexity.
>
> 2. We do not believe there is an inconsistency in running times. We elaborate below; please let us know if this elevates your concern.
>
> The rounding scheme (Algorithm 1) does not need to be run as a subroutine; in fact, it does not need to be run at all. Correctness of the rounding scheme (Algorithm 1) as in the proof of Theorem 1 implies that the set of menus whose prices, i.e., $p_1^{(i)}, p_2^{(i)}$, are multiples of $\alpha$ constitute (an almost) revenue-preserving set of menus. This set of menus for any fixed $\alpha$ (e.g., $1/\sqrt{T}$ in the full information case) is fixed and constitutes a grid in the parameter space.
>
> We will update the statement of Theorem 1 and add the following remark: *Our rounding scheme (Algorithm 1) is only described for the purpose of the proof to argue that the multiples of alpha provide (an almost) revenue-preserving set of menus. Algorithmically, all we have to do is to enumerate the multiples of $\alpha$.*
>
> Theorems 7 and 8 specify the running times of our learning algorithms (Algorithm 2 and Algorithm 3). These algorithms receive the discretized set of menus as the input initially. In each round, the algorithms choose a menu to be used and update the weights for all the menus.
>
> Calculation of the running time as noted in the proof of Theorem 7 (end of page 28): The learning algorithms need to maintain the weights for the set of discretized menus and update them based on the revenue at each round. In the full information case, where we observe the value of the buyer, the revenue of each menu can be calculated in $O(K \ell)$ given the buyer’s valuation. The running time in each round is the number of menus times the time to calculate the revenue for each menu. In the bandit case, the running time in each round is the same order as the number of menus.
>
> 3. Our dependence on T matches a lower bound by Kleinberg–Slivkins–Upfal [2008] for general globally Lipschitz functions even though the utility functions in our case are only piecewise Lipschitz (not globally Lipschitz). The subtle thing is this does not immediately imply a lower bound for our case since, on one hand, learning piecewise Lipschitz functions is harder than globally Lipschitz ones, and on the other hand, in our case, the utility functions have more structure beyond Lipschitzness in each piece. It is a nontrivial open question if it is really tight for our case and we will leave it as an open question in the paper.

---

> > ### Author Response · Authors · 2024-01-09
> > **Response to Reviewer Pkib, cont.**
> >
> > 4. We will add a discussion highlighting the distinct technical innovations of our paper compared to prior work for Theorems 10 and 11. In summary, the major novelty is establishing dispersion. After that, we use previously developed results, i.e., regret bounds for dispersed setting, from prior work (e.g., Theorem 1 in [BSV2018] for full information and Theorem 3 in [BSV2018] for bandit setting). To prove dispersion, we need to prove structural characterization. It has been shown in [BSV2018] that based on the values observed from users until time t, the parameter space is partitioned into convex regions with hyperplane boundaries such that the utility inside each region satisfies Lipschitz continuity. We give a more refined characterization by showing that (1) the utility function inside each region is linear, and (2) the boundary hyperplanes constitute a multiset of parallel hyperplanes. Properties (1) and (2) are important for establishing dispersion and obtaining no-regret online learning algorithms under smooth distributional assumptions, as in Theorems 10 and 11.
> >
> > Comparison between the performance between adversarial and smooth settings as provided in page 14 of the submission: *Although the discretization-based algorithms work under adversarial inputs and are more general, they provide similar regret bounds and even improved running times in some cases. In the full information case, the dependence on the regret bound in parameter $T$ is similar in both algorithms. In running time, the discretization-based algorithm suffers worse dependence in $H$, but enjoys better dependence in $T$ and $K$ (the maximum number of units) compared to the dispersion-based algorithm. In the bandit setting, similarly, the regret bounds are similar in their dependence on $T$, while the running-time comparison depends on the value of $\kappa$ (maximum density under smoothness assumption) such that lower-density distributions may result in better running times.*
> >
> > 5. Providing experimental, although well-motivated, as you mentioned, is beyond the scope of the theoretical foundations that we aimed for in our paper.
> >
> > Minor Comments:
> >
> > 1. We are open to moving the lemmas to the appendix to streamline our presentation.
> >
> > 2. Thank you for pointing it out. We will add the formula and modify the notation in the algorithm.

---

### Decision · Action_Editor_rvzE · 2024-03-01

**Recommendation:** Accept as is

**Comment:**

All reviewers agreed that the paper presents new results about learning revenue maximizing menus that go beyond what is presented in the literature.  The authors diligently addressed the reviewers' questions and requests for clarification.  While there were some reservations about how significant the results were, everyone agreed that the paper is suitable for TMLR according to the guidelines.

**Audience:**

This paper fits into a sizeable body of work on learning in auctions.  It would certainly be interesting to any members of the AGT and perhaps to the broader audience of researchers working on generalization bounds.

**Claims And Evidence:**

Yes, all claims are convincingly supported.  This is a theoretical paper and no significant issues with correctness were raised.  There were no significant questions about whether the results are significantly different from prior work.